# Cellular hierarchy insights reveal leukemic stem-like cells and early death risk in acute promyelocytic leukemia

Wen Jin [1,2,4], Yuting Dai [1,4], Li Chen [1], Honghu Zhu [3], Fangyi Dong[1], Hongming Zhu[1], Guoyu Meng [1], Junmin Li[1], Saijuan Chen [1], Zhu Chen[1,2] ✉, Hai Fang [1] ✉ & Kankan Wang [1,2] ✉

Acute promyelocytic leukemia (APL) represents a paradigm for targeted differentiation therapy, with a minority of patients experiencing treatment failure and even early death. We here report a comprehensive single-cell analysis of 16 APL patients, uncovering cellular compositions and their impact on all-trans retinoic acid (ATRA) response in vivo and early death. We unveil a cellular differentiation hierarchy within APL blasts, rooted in leukemic stem-like cells. The oncogenic PML/RARα fusion protein exerts branch-specific regulation in the APL trajectory, including stem-like cells. APL cohort analysis establishes an association of leukemic stemness with elevated white blood cell counts and FLT3-ITD mutations. Furthermore, we construct an APL-specific stemness score, which proves effective in assessing early death risk. Finally, we show that ATRA induces differentiation of primitive blasts and patients with early death exhibit distinct stemness-associated transcriptional programs. Our work provides a thorough survey of APL cellular hierarchies, offering insights into cellular dynamics during targeted therapy.

The landscape of cancer-targeted therapies has shifted from molecular target identification to cellular heterogeneity characterization and targeting[1,2]. The direct targeting of molecular changes driving tumorigenesis has proven to enhance therapeutic efficacy in cancer, as exemplified by the success of molecular targeted therapy in acute promyelocytic leukemia (APL)[3]. APL is characterized by its driver oncogenic fusion protein (PML/RARα), which plays a crucial role in initiating APL leukemogenesis, as supported by substantial evidence, including insights from murine models[4,5]. Notably, two drugs, all-trans retinoic acid (ATRA) and arsenic trioxide (ATO), have already achieved remarkable therapeutic outcomes by directly targeting this oncogenic fusion protein[6]. We[7–11] and others[12,13] have long been focusing on molecular mechanisms to illustrate how ATRA and ATO directly target the stability of PML/RARα to reverse transcriptional deregulation, affecting the proliferation and differentiation of APL cells. It is well-established that intratumoral heterogeneity at the cellular level is a critical factor in leukemogenesis, disease progression, and therapy response[14]. Advances in single-cell genomics technologies have uncovered the cellular heterogeneity of chronic myeloid leukemia and lung cancer[1,15], demonstrating the potential for exploring heterogeneous features to improve cancer therapy success rates. In the context of APL, it has now become imperative to explore whether leukemic cells driven by the same oncogenic driver, PML/RARα, exhibit diverse cellular states; if so, to what extent the cellular composition and transcriptional heterogeneity might impact the outcomes of targeted therapy in APL.

[1]Shanghai Institute of Hematology, State Key Laboratory of Medical Genomics, National Research Center for Translational Medicine at Shanghai, Ruijin Hospital, Shanghai Jiao Tong University School of Medicine, Shanghai 200025, China. [2]Sino-French Research Center for Life Sciences and Genomics, Ruijin Hospital, Shanghai Jiao Tong University School of Medicine, Shanghai 200025, China. [3]Department of Hematology, Beijing Chao-Yang Hospital, Capital Medical University, Beijing 100020, China. [4]These authors contributed equally: Wen Jin, Yuting Dai. ✉e-mail: zchen@stn.sh.cn; fh12355@rjh.com.cn; kankanwang@shsmu.edu.cn

Long-term therapeutic efficacy in cancer seems to be attributable to targeting a rare cell population with stemness potentials[16–18], such as leukemic stem cells (LSCs) in APL, although APL LSCs are far from clear, particularly in terms of cell-of-origin and their relationship with genetic events. Studies in mice have suggested that APL might arise from myeloid-committed progenitors, including committed myeloid progenitors (CMPs) and granulocyte-monocyte progenitors (GMPs)[19–21]. Alternatively, APL LSCs might arise from more primitive progenitors that are earlier than CMPs, as PML/RARα has been reported to be detectable in both CD34+CD38+ and CD34+CD38− populations isolated from APL patients[22]. To reconcile these seemingly conflicting observations, it is imperative to gain a comprehensive understanding of APL cellular compositions, with a particular emphasis on the rare LSCs.

Furthermore, extensive investigations into the molecular mechanisms underlying effective targeted therapies in APL have primarily relied on in vitro analyses or in vivo murine models[3,7,8]. These studies have sought to elucidate the effectiveness of ATRA and/or ATO in reversing the aberrant transcriptional regulatory activity of PML/RARα. Additionally, functional analyses of cell behavior have showcased the efficient induction of terminal granulocyte differentiation[3,7,8]. However, the genuine in vivo responses of APL cells to ATRA, particularly its impact on different cellular compartments, remain ill-defined. This limitation restricts our ability to explain the effectiveness of ATRA in treating APL patients with varying prognoses.

In this work, we use the single-cell RNA sequencing (scRNA-seq) technology to comprehensively dissect the cellular heterogeneity in APL and its potential impact on ATRA therapy in vivo and early death, as outlined in Fig. 1a. We generate a single-cell transcriptome resource on the malignant APL blasts from 16 newly diagnosed APL patients. Subsequently, we conduct a series of data analyses to gain deeper insights into APL cellular heterogeneity and its association with genomic and clinical characteristics of APL patients. The resources and findings presented in this study hold significant implications in four aspects. Firstly, our resource enables the characterization of intratumoral heterogeneity with multiple branches, including a small subpopulation of APL stem-like cells at the root of the differentiation trajectories. Secondly, at the single-cell level, we show that the stemness characteristics of APL stem-like cells are determined by PML/RARα target genes and can be further enhanced in the presence of FLT3-ITD. Thirdly, deconvolution analysis conducted on a large cohort of 323 APL patients reveals a significant association of higher APL stem-like cell proportions with elevated white blood cell (WBC) counts and the presence of FLT3-ITD. We also successfully construct an APL-specific stemness score, which effectively assesses prognosis, especially the risk of early death in APL patients. Lastly, our single-cell level investigations into the in vivo effects of ATRA confirm that ATRA directly targets APL primitive blasts, leading to their differentiation and maturation.

## Results
### Study design and analysis overview
We performed single-cell transcriptome analysis on bone marrow (BM) samples collected from 16 newly diagnosed APL patients (Fig. 1a; also see "Methods" section for patient selection and Supplementary Data 1 for detailed clinical characteristics and sample information). This cohort included four patients who experienced early death, which is defined as death occurring within 30 days from diagnosis. In this endeavor, we generated a total of 136,497 cells by combining 16 separate scRNA-seq datasets of APL BM samples at disease onset, forming the foundation for a comprehensive understanding of APL cellular composition. In parallel, we reanalyzed 23 separate scRNA-seq data of normal BM samples from healthy individuals (Gene Expression Omnibus with accession IDs of GSE120221 and GSE130116) to construct normal hematopoietic cell populations (totaling 102,792 cells) for use

as controls. We adjusted for batch effects and performed an integrated analysis of both APL and normal cell populations to characterize the malignant APL blasts and intratumoral heterogeneity (detailed in "Methods" section).

Based on the APL blasts characterized in this study, we designed a series of analyses and validations as follows: (i) we constructed the cellular architecture and differentiation trajectory of APL blasts, with a specific focus on identifying APL stem-like cells; (ii) we validated the expression of the PML/RARα fusion gene and FLT3-ITD using targeted scRNA-seq (scTarget) in two de novo APL patients; (iii) we determined the association between cellular compositions and clinical presentations, including the incidence of early death, by conducting deconvolution analysis on a large cohort of 323 APL patients (including 22 patients with early death); (iv) we explored the impact of ATRA therapy on changes in cellular compositions for three patients, both at disease onset and on Day 2 after ATRA treatment; and (v) we performed deconvolution analysis on RNA-seq data from 10 newly diagnosed APL patients before and after ATRA treatment to establish the association of cellular compositions, especially APL stem-like cells, with ATRA responses in vivo.

### Single-cell characterization of APL blasts
To unravel the cellular diversity within APL, we first performed analysis by comparing BM samples collected from APL patients with those from healthy individuals. Using 23 normal BM samples, we established the baseline of cellular diversity, which revealed seven major cell populations, consistent with previously published findings[23,24]. These populations included hematopoietic stem/progenitor cells (HSPCs), GMPs, monocytes (Mono), dendritic cells (DCs), B cells, T/Natural Killer (NK) cells, and erythroid (Ery) cells (detailed in "Methods" section and Supplementary Fig. 1).

Next, we employed UMAP to project 136,497 cells from 16 newly diagnosed APL patients, along with normal hematopoietic cell populations identified above, onto a two-dimensional space (Fig. 1b). While lymphoid and erythroid populations from APL patients formed clusters that corresponded to cell types also annotated by normal hematopoietic cells, APL cells were predominantly grouped into a distinct cluster. The accuracy of APL blast identification was confirmed by the exclusive presence of PML/RARα expression, as observed in two APL patient samples using PML/RARα-targeted scRNA-seq (Fig. 1b, right panel, and Supplementary Fig. 2). This cluster exhibited a high expression level of the gene MPO, which encodes a widely used diagnostic marker for APL[25], and also differed from the normal spectrum of hematopoietic cell populations (such as HSPCs and myeloid cell populations; Fig. 1b, c). The established markers of APL cells were highly expressed in this cluster, as highlighted by genes activated by PML/RARα-associated super-enhancers[8] (e.g., STAB1, CITED2, CCND2, and GFI1) (Fig. 1d). This finding further reinforced our previous findings, highlighting the role of PML/RARα in determining the identity of APL blasts through super-enhancer regulation[8]. Moreover, this cluster exhibited significantly elevated expression levels of GMP-specific genes, such as azurophilic granule genes (MPO, AZU1, ELANE, and CTSG) (Fig. 1d), supporting the notion that APL blasts may be blocked at the GMP stage[14,23].

We proceeded to compare the functional and regulatory characteristics of the APL blasts with those of the normal GMP cluster. Firstly, we conducted gene ontology (GO) enrichment analysis on differentially expressed genes (DEGs) calculated using Seurat[26]. The analysis revealed that significantly upregulated genes (adjusted P-value < 0.05) in APL blasts were of functional relevance to several key processes, including HSC self-renewal/differentiation (RUNX1, MYC, and JAG1), histone modification (EP300) and DNA methylation (DNMT3A and MBD1), cell cycle arrest and cell growth (CDK6, CCNA1, and WT1), as well as the response to endoplasmic reticulum stress and unfolded protein (XBP1, ATF6, and USP14) (Fig. 1e and Supplementary

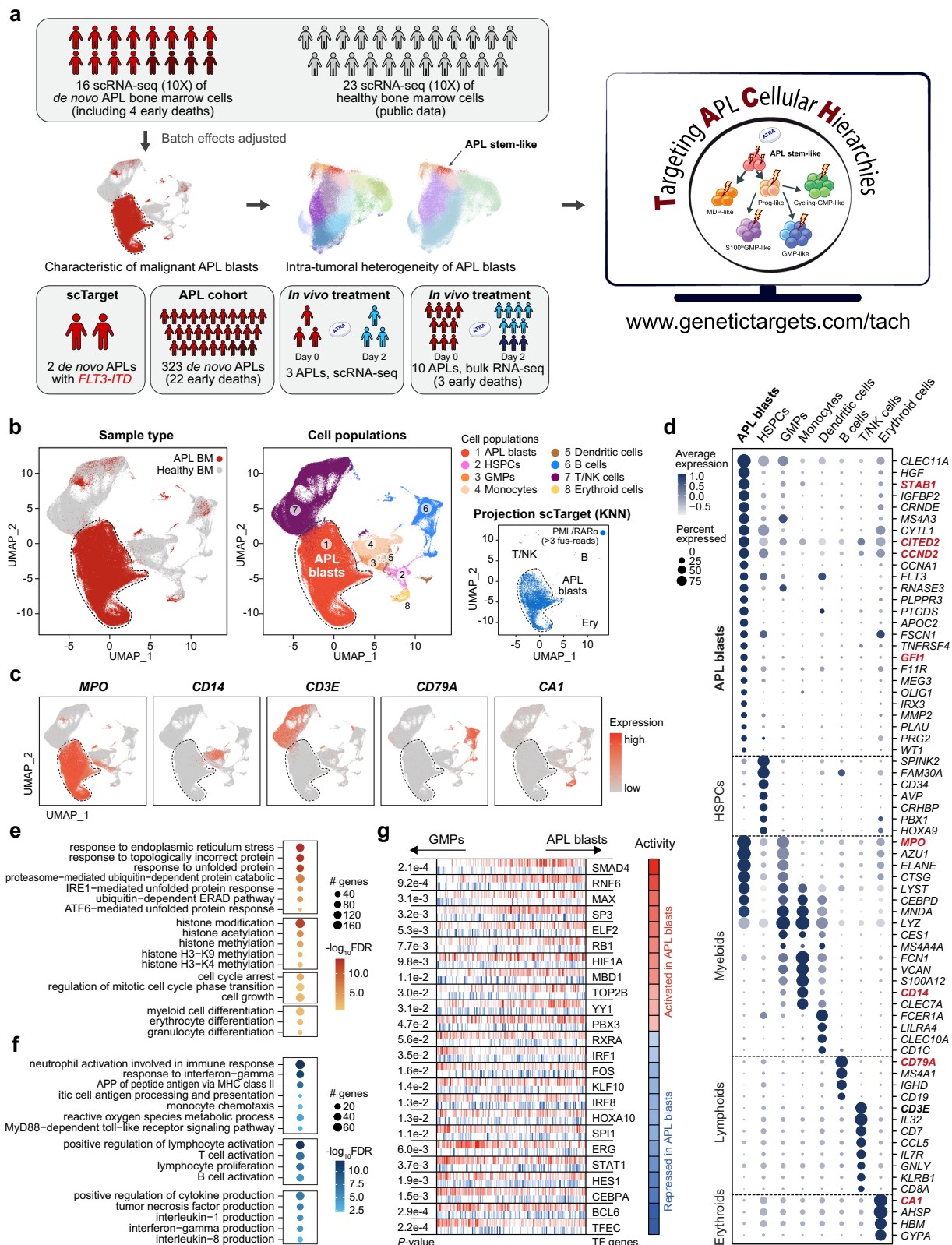

**Fig. 3a).** In contrast, genes that were significantly downregulated (adjusted *P*-value < 0.05) in APL blasts were enriched for immune response-related functions, including antigen processing and presentation (*PSMB9*, *PSMB10*, and *CTSS*), MHC class II protein complex (*CD74*, *HLA-DPA1*, and *HLA-DRA*), regulation of cytokine production (*CLEC7A*, *CCR2*, and *CCL3*), and response to interferon-gamma (*IRF8*, *IRF5*, and *IFI30*) (Fig. 1f and Supplementary Fig. 3b). These findings

confirmed the disruption of antigen presentation in APL cells[27]. Secondly, we used VIPER[28] to infer the transcription factor (TF) activity differentially between the two clusters. Our analysis suggested that the hematopoietic TFs and cofactors (such as SPI1, ERG, FOS, and RXRA) were repressed in APL blasts (Fig. 1g), supporting the differentiation blockade observed in APL blasts. Also repressed in APL blasts were the mediators of interferon (IFN) signaling (e.g., STAT1, IRF8, and IRF1).

**Fig. 1 | Characterization of APL blasts through an integrative analysis of scRNA-seq data from APL and normal bone marrow (BM) cells. a** Overview of the experimental strategy. **b** UMAP plots of APL and normal BM cells (*n* = 239,332 cells), with color-coding indicating sample types (left panel), inferred cell populations (middle panel), and PML/RARα-positive cells detected by scTarget in two APL samples (right panel). Cells detected with more than three PML/RARα fusion reads are illustrated. HSPCs hematopoietic stem/progenitor cells, GMPs granulocyte-monocyte progenitors, NK Natural Killer, Ery erythroid. **c** UMAP plots with each cell (*n* = 239,332 cells) colored according to their normalized expression of *MPO, CD14, CD3E, CD79A,* and *CA1*, respectively. **d** Normalized expression level and expression percentage of cell type-specific genes in eight cell populations in APL and normal BM cells. **e, f** Gene Ontology (GO) enrichment analysis showing significantly enriched biological process terms for upregulated genes (**e**) and downregulated genes

(**f**) in APL blasts compared with GMPs. **g** Inferred activated (red) and repressed (blue) TFs in APL blasts compared to normal GMPs. The central two-row graph illustrates the distribution of activated targets (depicted in red) and repressed targets (depicted in blue) of different TFs, with positions ranked according to the differential expression between APL blasts and normal GMPs (leftmost: most downregulated in APL blasts, rightmost: most upregulated in APL blasts). The regulatory model was based on the ARACNe-inferred interactome, provided in the build-in function of the VIPER R package. The *P*-value is shown on the left of the column, and the inferred differential activity level is shown on the right. The *P*-values were calculated using the msviper function in the VIPER R package. Two-sided *P*-values were calculated. APP antigen processing and presentation; MHC major histocompatibility complex.

Conversely, oncogenic TFs (e.g., RB1, HIF1A, and MAX), epigenetic regulators (e.g., SMAD4, MBD1, YY1, and SP3), and cell proliferation-associated TFs (e.g., RNF6, ELF2, and TOP2B) were activated in APL blasts compared to normal GMPs (Fig. 1g). Consistent with these findings, cell cycle analysis showed that the APL cluster significantly accumulated in the S or G2/M phases compared with normal myeloid cells, further indicating the highly proliferative state of APL blasts (Supplementary Fig. 4).

## Characterizing the intratumoral heterogeneity of APL blasts reveals a complex cell-state transition trajectory with leukemic stem-like cells sitting at the top

Next, we determined the cellular architecture and differentiation trajectory within the characterized APL blasts. Through unsupervised clustering and UMAP analysis, we identified 18 clusters, each characterized by distinct expression patterns of known marker genes[23] (Fig. 2a and Supplementary Data 2). Among these clusters, the 12 APL clusters (C1-C12) accounted for 82.9% of the cells and, as expected, exhibited high expression of GMP-specific genes[23], particularly those associated with azurophilic granules, such as *ELANE, CTSG,* and *AZU1* (Fig. 2b, right panel). Notably, these GMP-like clusters displayed significant heterogeneity. For example, the three major clusters C1-C3 exclusively expressed GMP-specific genes, whereas clusters C6-C10 showed high expression of cell proliferation-related genes (*TOP2A, MKI67,* and *PCNA*)[29], and clusters C11-C12 were marked by the highly expressed S100 family genes (*S100A8/A9/A10*).

Of striking interest, clusters C14-C16 exhibited stemness-like characteristics with high expression of marker genes specific to early HSPCs and/or LSCs (such as *CD200*[30], *CD44*[31], *CD99*[32], *CD2*[33], and *FAM30A*[34]) (Fig. 2b), master stemness-related TF genes (such as *SOX4*[35] and *MYC*[36]), as well as APL characteristic genes (such as *MPO*) (Supplementary Fig. 5). Comparatively, C15 had the highest expression levels of stemness- and progenitor-specific genes compared to C14 and C16. In addition, C14 also expressed CD38, while C16 highly expressed marker genes associated with monocyte-DC progenitors (MDP)[37], such as *CSF1R/CD115* and *FLT3/CD135*. Collectively, the cells in C15 appeared to resemble the most primitive cells among APL blasts, possibly representing leukemic stem-like cells.

Next, we performed RNA velocity analysis to verify the differentiation trajectory from the leukemic stem-like cell cluster (C15) to the GMP-like cell clusters. The 18 APL clusters were reorganized into six branches (Supplementary Data 3), with the stem-like cell cluster (C15) sitting at the root of differentiation trajectories, supported by both the velocity-based cell-state transition probabilities and the similarity of expression patterns (Fig. 2c, d). Likely going through the Prog-like branch (C14), the stem-like cell cluster (C15) gave rise to the three main branches of APL blasts: the GMP-like branch (C1-C5 and C13), the cycling GMP-like branch (C6-C10), and the S100^hi^GMP-like branch (C11 and C12) (Fig. 2c, d and Supplementary Fig. 6). Another trajectory starting from C15 led to the MDP-like cell cluster (C16), which subsequently differentiated into cells in the CD1C+ PrecDC-like cell cluster

(C17) and the CD14+ Promonocyte-like cell cluster (C18). This trajectory supports the current view that, instead of being derived from GMPs[37], MDPs might represent an earlier stage, possibly even earlier than the CMP stage. Furthermore, Monocle2 pseudotime ordering trajectory analysis also showed that the stem-like cell cluster was the starting point of APL blasts, giving rise to the S100^hi^GMP-like and GMP-like branches (Fig. 2e). This analysis also revealed that RUNX1, RUNX2, and interferon-related factors (i.e., STAT1 and FOS) might be involved in the GMP-like differentiation trajectory, while CEBP family members (CEBPA, CEBPB, and CEBPE), MAFB, JUNB, and JUND were likely associated with the lineage decision towards the S100^hi^GMP-like APL blasts (Fig. 2f).

Furthermore, we investigated the role of PML/RARα in the identified APL trajectory. PML/RARα-targeted scRNA-seq confirmed that PML/RARα was uniformly expressed across all six branches of APL blasts, with notable expression in the stem-like cell cluster (Fig. 2g, h and Supplementary Fig. 7a). Furthermore, we integrated PML/RARα chromatin occupancy data[31] obtained from CUT&Tag-seq (Cleavage Under Targets and Tagmentation sequencing) in an APL patient-derived cell line, NB4 (Supplementary Data 4 and Supplementary Fig. 7b). Notably, each branch possessed a considerable number of distinct PML/RARα targets (Fig. 2i and Supplementary Data 5), suggesting the presence of branch-specific expression patterns for PML/RARα targets across the APL trajectory. GO analysis provided insights into their functional significance, revealing that these targets were associated with distinct functional pathways (Supplementary Fig. 7c). For instance, the PML/RARα targets located within the APL stem-like cells were found to be predominantly involved in stem cell maintenance. Those within cycling GMP-like cells were mainly linked to E2F targets involved in the G2-M checkpoint. Those within GMP-like cells exhibited marked enrichment in ribosomal functions. These results collectively illuminate the impact of PML/RARα in shaping the intratumoral heterogeneity of APL cells.

## The characteristics of APL stem-like cells were determined by PML/RARα target genes and further enhanced by FLT3-ITD

To delve into the characteristics of the APL stem-like cells defined in our study, we identified APL-specific leukemic stemness genes by comparing the transcriptome data between the APL stem-like cell cluster (C15) in Fig. 2a and the HSPC cluster in Fig. 1b (Supplementary Data 6). We obtained the following findings.

First, PML/RARα targets were significantly enriched in these APL-specific leukemic stemness genes (Fig. 3a and Supplementary Fig. 8). Notably, well-known LSC marker genes[16,38–40], such as *FCGR2A, CD9, ITGA5, IL1RAP,* and *CD82*, were exclusively expressed in APL stem-like cells but not in HSPCs. They were also direct targets of PML/RARα (Fig. 3b and Supplementary Fig. 9). Moreover, well-known stemness/self-renewal-related PML/RARα targets[8], such as *HCK* and *GFI1*, were highly expressed in APL stem-like cells. Genes closely related to APL leukemogenesis, such as *FLT3* and *JAG1*[41,42], were not only targets of

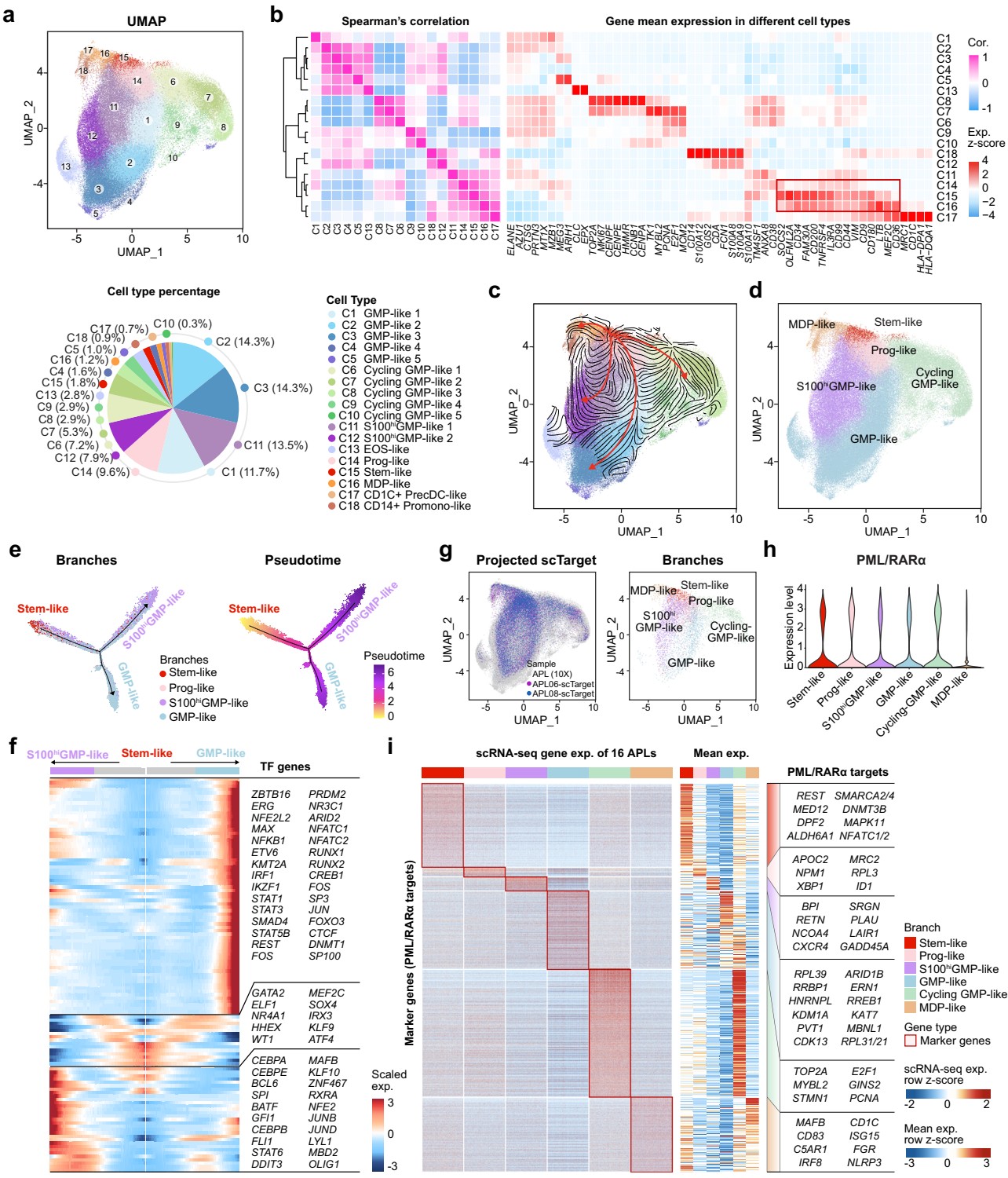

PML/RARα but also exhibited elevated expression in APL stem-like cells (Fig. 3b).

Second, we performed pathway enrichment analysis to reveal the involvement of the PML/RARα-dysregulated signaling pathways in APL stem-like cells. This analysis identified numerous LSC-associated pathways, including classical WNT, MAPK, VEGF, P53, and mTOR signalings[43,44] (Fig. 3c), all crucial for maintaining the LSC population. Furthermore, pathway crosstalk analysis based on these LSC-associated PML/RARα target genes indicated that they coordinated the regulation of APL stem-like cells (Supplementary Fig. 10 and Supplementary Data 7).

Third, we performed VIPER analysis to elucidate potential TFs involved in the PML/RARα-induced transcriptional network in APL stem-like cells (Fig. 3d). By comparing with normal HSPCs, we found that stemness-associated TFs, such as TGIF1 and HIF1A[45,46] were activated, suggesting their potential roles in regulating the stemness of APL cells. Additionally, we observed the activation of regulators associated with histone modification, including MBD2/3 (methylated-DNA binding proteins) and PRDM2 (H3K9 methyltransferase), in APL stem-like cells. These findings suggested their potential involvement in epigenetic control of self-renewal and quiescence of APL stem-like cells[47]. TFs linked to malignant transformation and stemness

**Fig. 2 | Characterization of intratumoral heterogeneity in APL blasts reveals a complex trajectory with multiple branches and a small subpopulation of APL stem-like cells. a** UMAP plot of APL blasts (upper panel; $n = 126,802$ cells). Eighteen clusters are labeled in different colors and numbers (lower panel). GMP granulocyte-monocyte progenitors, EOS eosinophils, Prog progenitors, MDP monocyte-DC progenitors, PrecDC pre-conventional dendritic cells, Promono promonocytes. **b** The left heatmap shows Spearman's correlation between the 18 APL clusters, calculated using the average expression profiles of the clusters. The right heatmap illustrates the expression levels of cell type-specific genes in each cluster. **c** Visualization of RNA velocity-based cell-state transitions of APL blasts. **d** UMAP plot of APL blasts with six branches, i.e., stem-like, Prog-like, S100hiGMP-like, GMP-like, cycling GMP-like, and MDP-like branches. **e** Pseudotime-ordered analysis of four major branches in APL blasts, including stem-like, Prog-like, S100hiGMP-like, and GMP-like branches. **f** Heatmap showing the dynamic changes in gene expression ($n = 116$ genes) along the pseudotime. Cell branches are labeled by colors (upper panel), including stem-like cells (center), S100hiGMP-like cells (left), and GMP-like (right). Characteristic transcription factors (TFs) are listed on the right. **g** UMAP plots of the targeted scRNA-seq (scTarget) data from two APL patients, with color coding for sample types (left panel) and branches (right panel). On the right panel, cells detected more than three PML/RARα fusion reads were illustrated. **h** The expression levels of PML/RARα in six branches of APL blasts. **i** Branch-specific expression patterns for PML/RARα targets across the APL trajectory. The left heatmap visualizes the single-cell expression of PML/RARα-regulated branch-specific marker genes across branches, with rows representing genes and columns for cells. To offer a clear and representative depiction of the branch-specific expression patterns for PML/RARα targets, we selected 1000 cells from each branch for interpretation. The right heatmap displays the mean gene expression ($n = 1758$ genes) across branches, accompanied by the annotations of representative marker genes on the right side. Cor. correlation, Exp. expression.

properties, such as OLIG2 and ARID3B[48,49], were also found to be active in APL stem-like cells. These findings highlighted the critical roles of PML/RARα in regulating APL stem-like cells at the single-cell level.

Fourth, we proceeded to associate the cellular architecture of APL blasts with the cooperating genetic alterations commonly found in APL, including FLT3 (FLT3-ITD and FLT3-TKD), WT1, and NRAS mutations. As illustrated in Fig. 3e, FLT3-ITD was significantly associated with a more primitive disease phenotype than other investigated mutations, suggesting that the presence of FLT3-ITD might play a significant role in enhancing the leukemic stemness. We further used two established leukemic stemness scorings[34,50], LSC17 and LSC6, to compare the stemness characteristics between APL stem-like cells with and without FLT3-ITD. The analysis revealed that the stem-like cells with FLT3-ITD received significantly higher scores than those without FLT3-ITD (Fig. 3f), supporting the notion that FLT3-ITD might enhance the stemness characteristics. Moreover, we performed FLT3-ITD-targeted scRNA-seq in two APL patients to verify its existence and, more importantly, to confirm its higher expression in APL stem-like cells (Fig. 3g, h).

## The predictive power of the APL stemness score in early death and therapy outcome in APL

In this section, we explored how APL stem-like cells defined by scRNA-seq can be utilized to predict the clinical obstacles in APL, more precisely, the occurrence of early death, risk stratification, and therapy outcome. We first established deconvolution-based prediction procedures, as graphically illustrated in Fig. 4a and detailed in the "Methods" section, and demonstrated their robustness and performance (Fig. 4b). At Step 1, we applied the CIBERSORTx algorithm[51] to scRNA-seq data, generating an APL-specific signature matrix that involved six APL blast cell populations (i.e., stem-like, Prog-like, GMP-like, cycling GMP-like, S100hiGMP-like, and MDP-like) and three non-leukemic cell types (i.e., T/NK, B, and erythroid cells). At Step 2, we prepared the transcripts per kilobase of the exon model per million mapped reads (TPM) matrix from bulk RNA-seq data. At Step 3, we employed support vector regression (SVR)[52] to deconvolute both the signature and TPM matrices, resulting in a coefficient matrix. The percentage of each cell type in the coefficient matrix was used to build a linear regression model for benchmarking. At Step 4, we constructed an 11-gene scoring model through LASSO to evaluate the stemness of APL blast cells from bulk RNA-seq data, where a higher score indicates a higher stemness. Additionally, we designed a leave-one-out test to demonstrate the robustness of the inferred APL stem-like cell proportions (median $R = 0.933$, 95% CI $= 0.922$–$0.945$, Fig. 4b). In other words, our deconvolution approach could accurately predict APL stem-like cells from bulk APL transcriptomes.

Next, employing our established deconvolution prediction procedures, we examined a large cohort comprising 323 APL patients[53] to explore the correlation between APL stem-like cells and the clinical characteristics of APL patients (Supplementary Data 8). Firstly, a higher proportion of APL stem-like cells was significantly associated with an elevated white blood cell (WBC) count ($P < 0.0001$) and a lower platelet count ($P = 4.0\text{e-}3$) (Fig. 5a). Notably, the APL stem-like cell type showed the strongest correlation with the WBC count, followed by GMP-like and cycling GMP-like cell types (Supplementary Figs. 11, 12 and Supplementary Data 9). This finding also emphasized the intertumoral heterogeneity among APL patients. Further analysis revealed that a higher proportion of stem-like cells was significantly associated with a higher percentage of APL blasts in BM cells ($R = 0.53$, $P < 0.0001$; Supplementary Fig. 13a) and a higher blast count in peripheral blood ($R = 0.35$, $P < 0.0001$; Supplementary Fig. 13b). This was also notably correlated with an increased WBC ($P < 0.0001$; Supplementary Fig. 14). These results indicated that APL patients with a higher percentage of APL stem-like cells in APL blasts might have an increased tendency for blasts to circulate in peripheral blood. Secondly, we examined the relationship between the percentage of APL stem-like cells and recurrent mutations in APL patients, including three common isoforms of PML/RARα (L-type, S-type, and V-type)[54], FLT3 mutations (ITD and TKD), and mutations involving WT1, NRAS, and ARID1A. Remarkably, a higher percentage of APL stem-like cells was significantly associated with the S-type PML/RARα transcript (S-type vs. L-type: $P < 0.0001$; S-type vs. V-type: $P = 7.3\text{e-}3$; Fig. 5a and Supplementary Fig. 15) and FLT3-ITD ($P < 0.0001$). In addition, FLT3-ITD was identified as the most significant co-occurrence event, supporting the importance of FLT3-ITD in enhancing the stemness activity of APL stem-like cells (Fig. 3f).

Given the close association of APL stem-like cells with potential unfavorable prognostic factors (including the high WBC count, S-type PML/RARα, and FLT3-ITD), we sought to develop a stemness scoring system tailored for APL blast cells. Employing the LASSO algorithm, we established an APL-specific stemness score based on the estimated cell proportions of APL stem-like cells, which was then utilized to quantify the stemness of leukemic cells in each patient (detailed in "Methods" section; Supplementary Fig. 16). We identified eleven genes (*SKAP2*, *IL1RAP*, *PLD1*, *HOPX*, *TRIM47*, *MAP2K1*, *TNFSF4*, *OLFML2A*, *P2RY14*, *NPTX2*, and *RALA*) to construct the APL stemness score, which showed a significant correlation with the proportion of APL stem-like cells (Pearson's correlation $= 0.802$; $P < 0.0001$). We then explored the relationship between the APL stemness score and prognosis, including overall survival (OS), event-free survival (EFS), and disease-free survival (DFS). Remarkably, a high APL stemness score was significantly associated with a poorer OS ($P = 5.7\text{e-}3$) and EFS ($P = 0.0342$), but not DFS ($P = 0.731$; see Fig. 5b, with the cutoff optimized using the R 'maxstat' algorithm). Univariate Cox analysis also revealed that patients with a higher APL stemness score had a poor prognosis, as reflected by OS ($P = 3.4\text{e-}4$; Fig. 5c) and EFS ($P = 4.6\text{e-}4$; Fig. 5d) using the optimized cutoff. Multivariate Cox analysis confirmed the APL stemness score as an independent prognostic factor for OS and EFS (Supplementary Fig. 17).

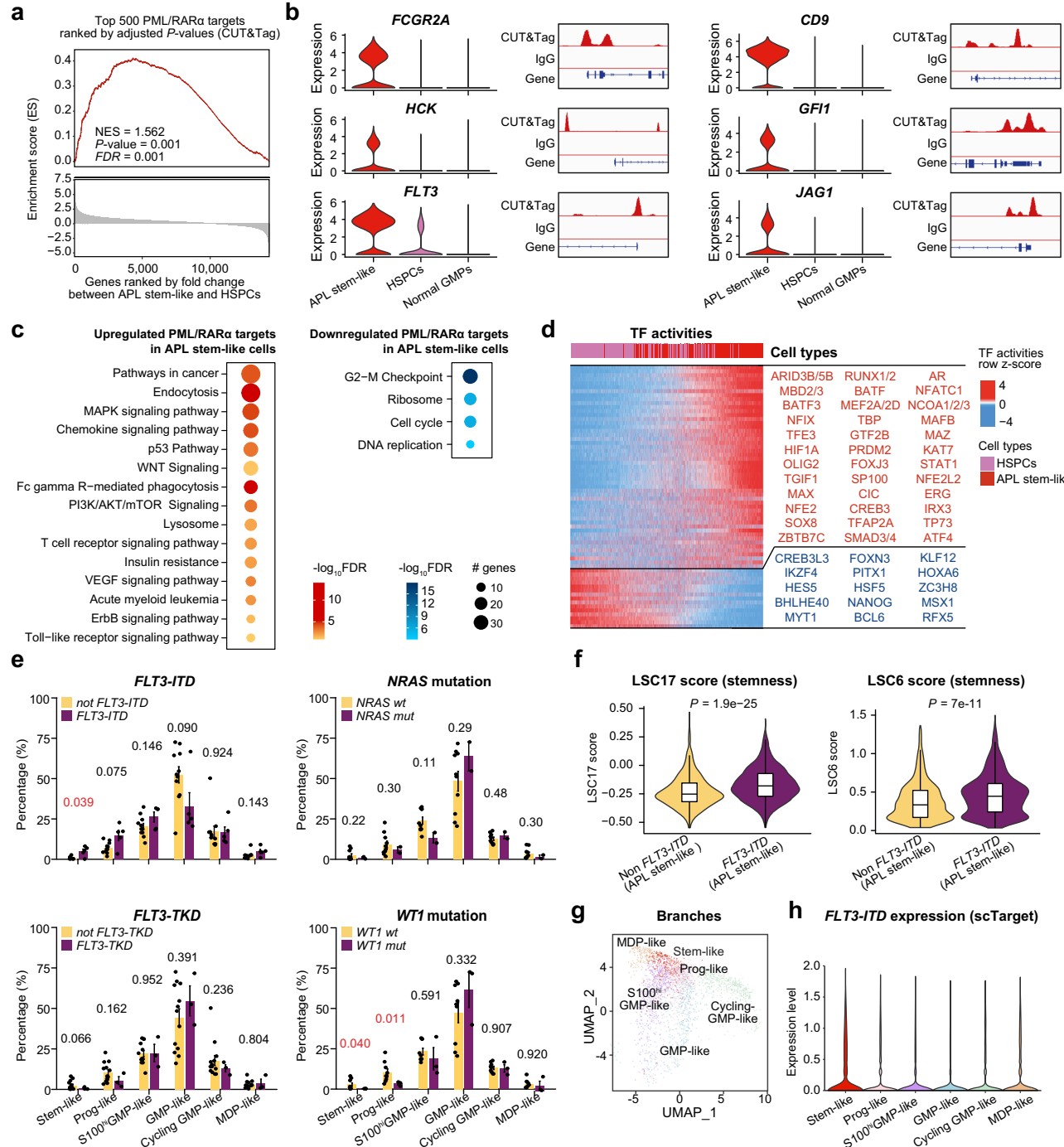

**Fig. 3 | The crucial role of PML/RARα and FLT3-ITD in regulating the properties of APL stem-like cells. a** Gene set enrichment analysis (GSEA) plot of top 500 PML/RARα targets. The gene set for GSEA analysis was defined based on the top 500 PML/RARα targets according to adjusted *P*-values derived from CUT&Tag. Genes were ranked by the fold change between APL stem-like cells and HSPCs at the mRNA level. NES normalized enrichment score. The *P*-value was calculated using GSEA. A two-sided *P*-value was calculated. **b** Violin plots illustrating representative genes highly expressed in APL stem-like cells compared with HSPCs. **c** GO enrichment analysis showing the KEGG pathways enriched in upregulated (left panel) and downregulated (right panel) PML/RARα targets in APL stem-like cells compared with HSPCs. **d** Master regulator analysis to explore activated (red) and repressed (blue) transcription factors (TFs) in APL stem-like cells compared with HSPCs. **e** Comparison of the percentages of each branch with and without indicated mutations. *n* = 16 patients with FLT3-ITD/TKD mutation information and *n* = 12 patients with NRAS/WT1 mutation information. Error bars in bar plots represent the

means ± SE. The *P*-values were calculated using Student's *t*-test and labeled in red when *P*-values < 0.05. Two-sided *P*-values were calculated. **f** Comparison of the LSC17 score (left panel) and the LSC6 score (right panel) of the stem-like cells in APL patients with or without FLT3-ITD. *n* = 2344 stem-like cells were used for visualization, excluding those with a score of 0 due to the absence of detected gene expression. In the boxplot, a black line within the box marks the median. The bottom and top of the box are located at the 25th and 75th percentiles, respectively. The bars represent values more than 1.5 times the interquartile range from the border of each box. The *P*-values were calculated using the Wilcoxon rank-sum test. Two-sided *P*-values were calculated. **g** Visualization of FLT3 expression through projection onto the UMAP of APL blasts using the scTarget data from two patients. Cells detected with more than three FLT3-ITD mutated reads were color-coded according to the different branches. **h** The expression levels of FLT3-ITD in the six branches of APL blasts, analyzed using data from FLT3-ITD-specific targeted scRNA-seq (scTarget).

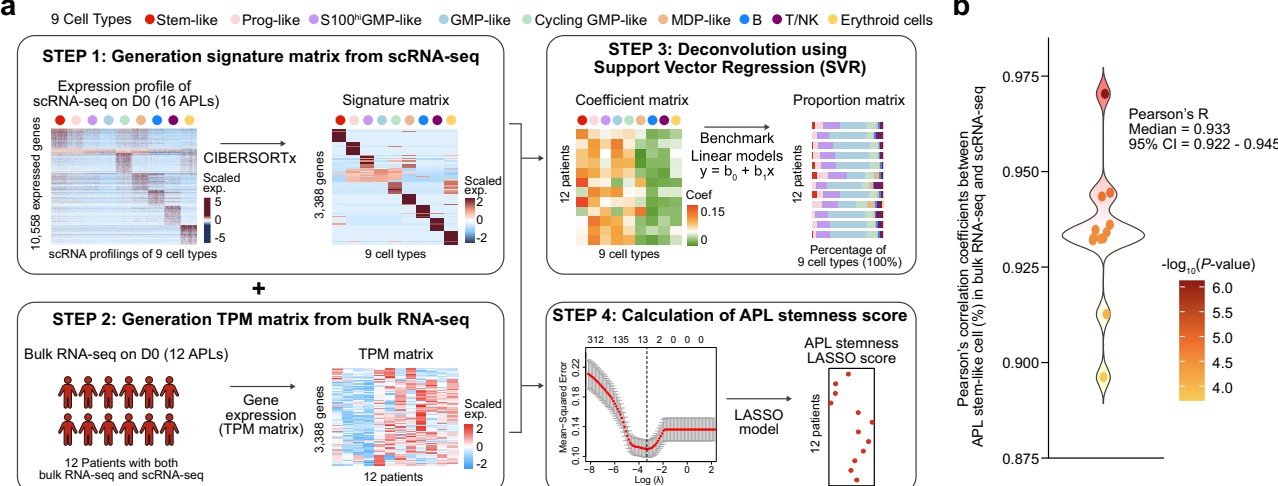

**Fig. 4 | Construction of an 11-gene APL stemness score. a** Schematic depicting the APL deconvolution approach and the generation of the APL stemness score using the 16 APL scRNA-seq data as the reference. This improved deconvolution approach is based on the support vector regression (SVR) algorithm: (1) generation of signatures from scRNA-seq populations, including the six APL blasts branches and T/NK cells, B cells, and erythroid cells; (2) calculating the TPM matrix of bulk RNA-seq of the 12 patients with matched scRNA-seq (detailed in "Methods" section); (3) using SVR to calculate the coefficients of each scRNA-seq population from bulk RNA-seq and performing linear regression to benchmark the percentage and the coefficient of each population; (4) generation of APL stemness signature genes, which were PML/RARα targets highly expressed in APL stem-like cells; and (5) the APL stemness score was calculated by the mean expression level of APL stemness signature genes. **b** Model performance of the deconvolution approach to predict the APL stem-like cell percentage from bulk RNA-seq of the 12 patients. The performance of the model is evaluated by Pearson's correlation coefficients between the observed APL stem-like cell percentage from scRNA-seq and the predicted APL stem-like cell percentage from the deconvolution approach. Leave-one-out (LOO) is used to evaluate the robustness of the model. The *P*-values were calculated using the Pearson's correlation. Two-sided *P*-values were calculated.

Notably, a higher APL stemness score was significantly associated with an increased risk of early death ($P = 8.3e\text{-}3$; Fig. 5b). This finding was consistent with our scRNA-seq data, revealing that compared to patients without early death, those patients with early death had a relatively higher proportion of stem-like cells, as well as elevated expression of stemness-associated genes, such as *FCGR2A*, *IL1RAP*, *MAP2K1*, and *KLF9* (Supplementary Fig. 18). Moreover, further analysis showed that our APL stemness score was an independent risk factor (HR = 5.627; 95% CI, 1.981–15.980; $P = 0.001$) with a superior predictive value for early death (Supplementary Fig. 19). These results collectively underscore the utility of the APL stemness score in assessing APL risk, including the risk of early death in APL.

## In vivo effect of ATRA on differentiation of primitive APL blasts and its influence on early death risk

Administrating ATRA as early as possible has been proven essential in reducing the early death rate in APL[48,49]. We then delved further into exploring the in vivo impact of ATRA treatment on APL cellular hierarchies, with a particular focus on the primitive blasts, as their stemness might influence treatment response. We performed scRNA-seq on BM samples collected from three patients (APL03, APL04, and APL05) after two days of ATRA therapy. At this time point, a notable increase in CD11b expression was observed (Supplementary Fig. 20a). Using the pre-defined six branches of APL blasts at diagnosis served as the reference, the cell types of APL cells on Day 2 after ATRA treatment were determined by employing the KNN algorithm in a merged dataset that included cells from both Day 0 and Day 2. We then investigated the in vivo effects of ATRA through changes in the abundance of cell groups and differential transcriptional regulation. We further applied the deconvolution method on bulk transcriptomes from RNA-seq performed on 10 APL patients (including three who suffered from early death) before and after ATRA treatment to explore the potential influence of leukemic stemness on the differential in vivo response to ATRA, which might be a contributing factor to early death. The stemness scores were indeed higher in patients who experienced early

death than those who achieved complete remission in bulk RNA-seq data of these 10 patients, consistent with our findings from the large cohort (Supplementary Fig. 20b).

First, comparison analysis of scRNA-seq data revealed that diagnostic APL samples were enriched in more primitive cells than post-treatment samples, which were relatively enriched in more mature progenitor cells (Fig. 6a). Especially on Day 2 after treatment, the stem-like cells were almost undetectable in the post-treatment samples (Fig. 6b). More precisely, ATRA treatment significantly increased the percentages of the more mature GMP-like cell type and decreased the percentages of three primitive cell types (stem-like, Prog-like, and S100^hi^GMP-like), with stem-like cells almost undetectable after the treatment, suggesting that ATRA had a notable impact on primitive APL cells, especially the stem-like cells (Fig. 6c, d). To confirm the ability of ATRA to induce the differentiation of APL primitive blasts towards more mature progenitor cells, we also employed the established score to quantify the matureness of leukemic cells[14], and observed a significant increase in this score in the 2-day post-treatment samples compared to samples at diagnosis (Fig. 6e), indicating that APL blasts became more mature upon ATRA treatment. Similar findings were also obtained by analyzing bulk transcriptomes of 7 APL patients who achieved complete remission before and after ATRA treatment. As illustrated in Fig. 6f, the proportion of APL stem-like cells and the stemness of leukemic cells were notably decreased following ATRA treatment, especially on Day 2.

Next, we looked at the expression changes of hematopoietic differentiation-related CD markers and TFs to show that ATRA could induce a stepwise differentiation, starting from APL stem-like cells towards more mature cells. Notably, we observed a significant down-regulation of APL stemness CD markers (such as *CD200*, *CD34*, *FCGR2A/CD32*, *CD9*, and *IL3RA/CD123*) and TFs (such as *HHEX*, *MYC*, *JAG1*, and *ERG*) in primitive cell types (Fig. 6g and Supplementary Data 10). Conversely, markers and TFs associated with mature hematopoietic cell lineages were upregulated following ATRA treatment. For example, *CD38*, *CD84*, *CEBPA*, and *ELF4* were upregulated in the

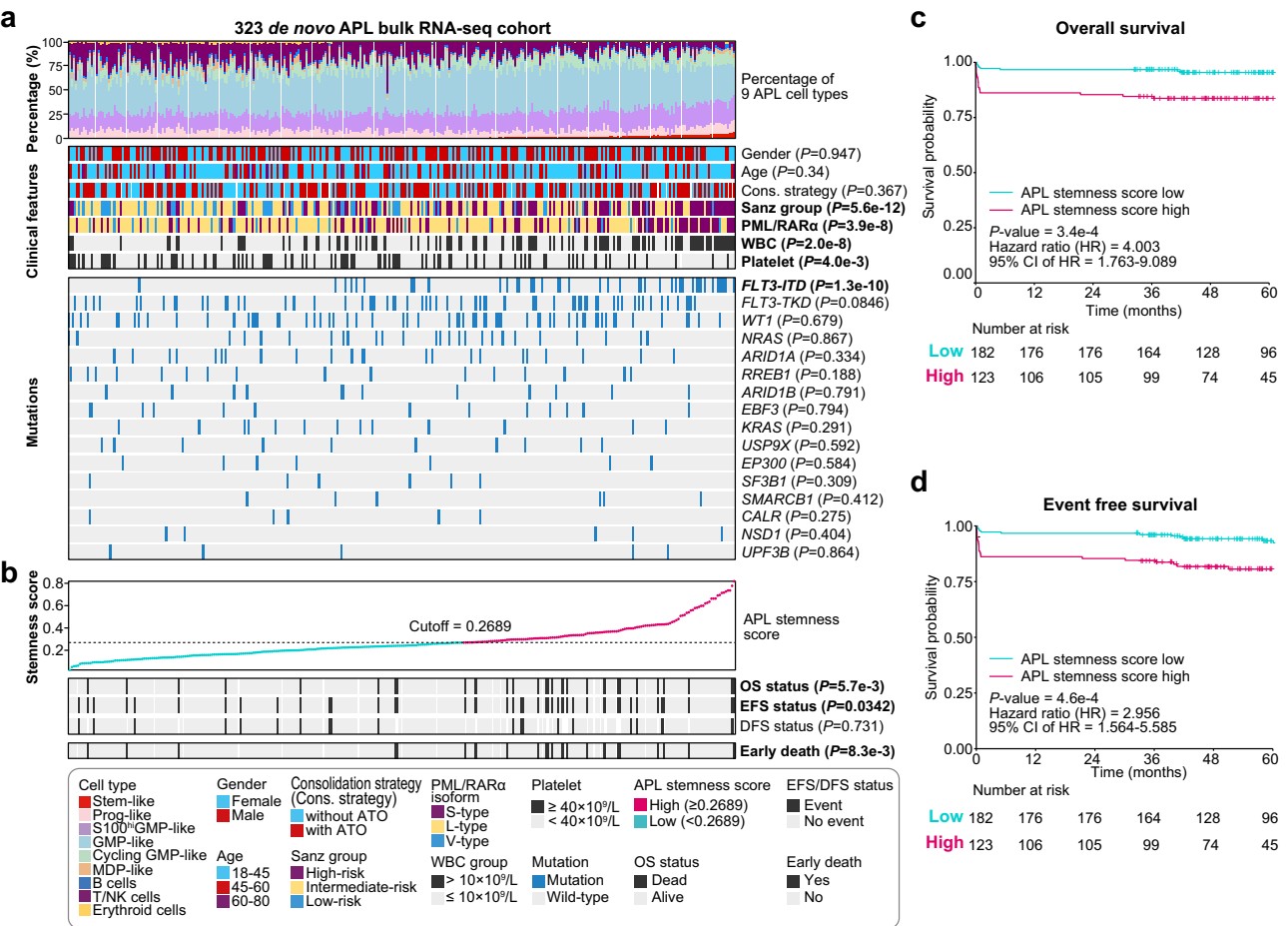

**Fig. 5 | Informativeness of the APL stemness score in predicting prognosis and early death in APL. a** Profiling of the predicted APL cell type percentage, clinical features, and gene mutations in the 323 APL patients. Columns represent individual APL patients, and the *P*-values were calculated using the Wilcoxon rank-sum test (statistical tests with two groups) or analysis of variance (ANOVA, with more than two groups) to illustrate the correlation between the APL stem-like cell percentage and clinical features/genetic alternations. Two-sided *P*-values were calculated. **b** Examination of the APL stemness score and its relationship to the prognosis of the 323 APL patients. The *P*-values were calculated using the

Wilcoxon rank-sum test to reveal the correlation between the APL stemness score and the prognosis of the 323 APL patients. * PML/RARα isoforms mainly encompass three typical types: long (L), short (S), and variant (V) types, respectively defined by the breakpoint of PML on intron 6, intron 3, and exon 6. OS overall survival, EFS event-free survival, DFS disease-free survival. **c**, **d** Kaplan–Meier estimates of overall survival (OS) (**c**) and event-free survival (EFS) (**d**) of APL patients (*n* = 305 patients with prognostic information) in the entire cohort stratified by the APL stemness score. The *P*-values were calculated using the log-rank test. Two-sided *P*-values were calculated.

Prog-like cells; *ITGAM/CD11b*, *C5AR1*, *TFEC*, and *MAFB* were upregulated in the S100hiGMP-like cells; and *CD24*, *CEACAM6*, *CEBPE*, and *CEBPB* upregulated in the GMP-like cells (Fig. 6h and Supplementary Data 10).

Of particular interest, we identified distinct transcriptional responses induced by ATRA in APL patients with and without early death. As illustrated in Fig. 6i, the consistent downregulation of several stemness-associated CD markers (*CD200* and *CD9*) and TFs (*HHEX* and *NFATC2*) was not observed in APL patients with early death (Supplementary Data 11). Similarly, the consistent upregulation of differentiation-related CD markers (*FUT4* and *IL1R1*) and TFs (*FOS* and *EGR1*) was also not observed in patients with early death (Fig. 6j and Supplementary Data 11). GO analysis revealed similar results: in addition to the induction of differentiation, the repression of stemness-associated pathways, such as the MAPK cascade, was also not observed in patients with early death after ATRA treatment (Supplementary Fig. 20c, d). Additionally, a significant decrease of the APL stemness scores after ATRA treatment was observed in patients who achieved complete remission, but not in patients who experienced early death (Supplementary Fig. 20b). Our findings suggest that ATRA treatment had a lesser impact on the stemness program in APL patients with early

death, potentially explaining the significant association we observed between the stemness activity of leukemic cells and early death, as revealed by our deconvolution analysis of transcriptomes from 323 APL patients (Fig. 5b).

## Discussion

Single-cell RNA sequencing (scRNA-seq) provides a powerful means to precisely identify complex cell type compositions, especially rare cell types. In this study, we employed scRNA-seq to comprehensively decipher APL heterogeneity and draw an overall picture of the APL hierarchy composition at the single-cell level. Our findings illuminated the pivotal role of PML/RARα in orchestrating the intratumoral heterogeneity of APL blasts, with a special emphasis on its influence on the stem-like cells identified in our study. Furthermore, we discovered that FLT3-ITD further enhanced the stemness attributes of these cells. By integrating scRNA-seq data with a large cohort of bulk RNA-seq data and in vivo ATRA treatment data, we uncovered the complex and multifaceted contributions of APL stem-like cells to early death in APL. This included a notable increase in cell proportion, elevated expression of stemness-associated genes, and the persistence of the stemness program post-ATRA treatment in patients with early death. More

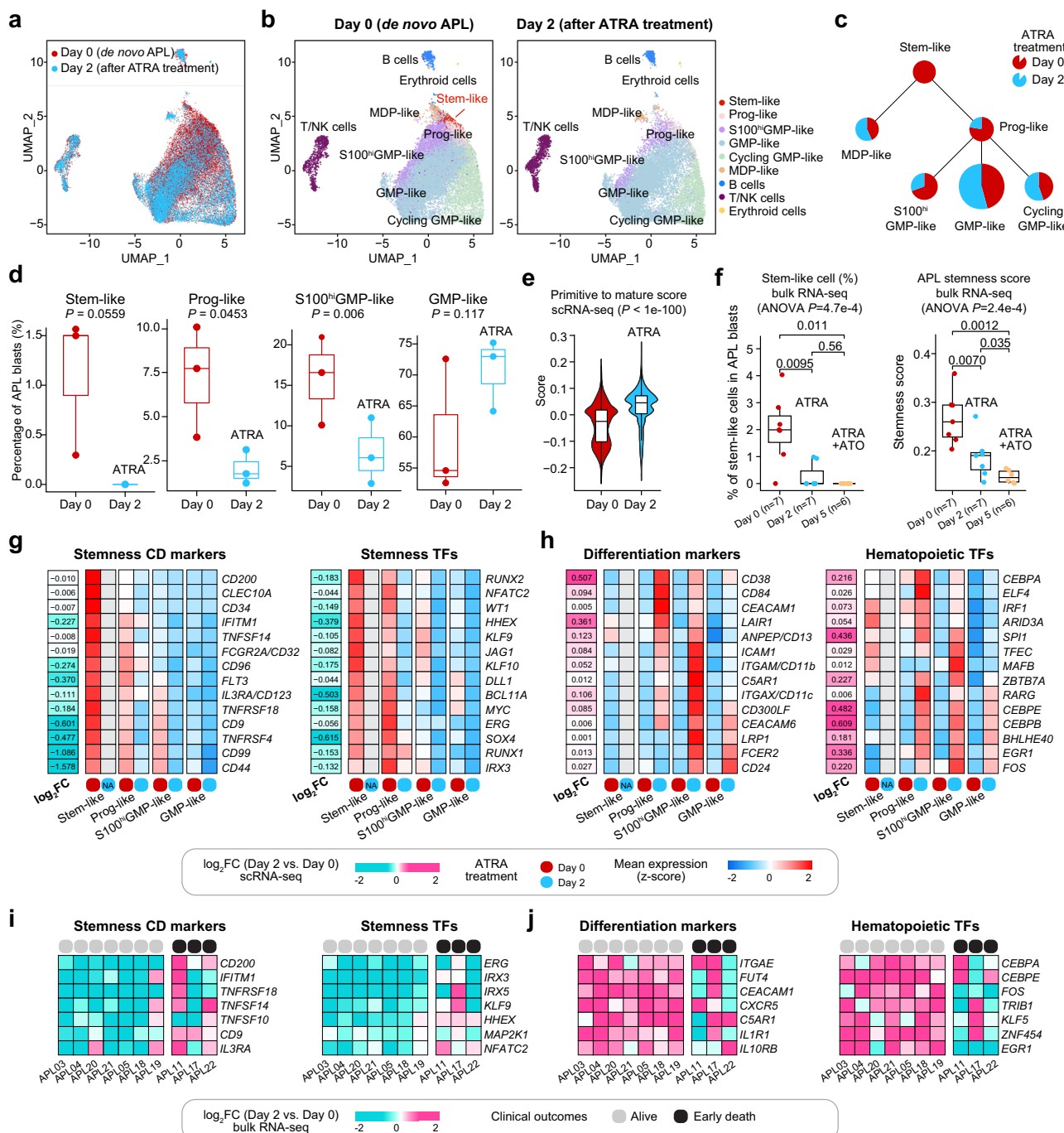

**Fig. 6 | In vivo analysis revealing direct targeting of APL primitive blasts (including stem-like cells) by ATRA and induction of their differentiation and maturation. a** UMAP plot showing the integration of APL BM cells ($n = 39,415$ cells) collected on Day 0 (red) and Day 2 after ATRA treatment (blue). **b** UMAP plots showing all defined cell populations of APL BM cells ($n = 39,415$ cells) collected on Day 0 and Day 2 after ATRA treatment. **c** Comparison of the percentages of six branches before (red) and after two days of ATRA treatment (blue). **d** Comparison of the percentages of the stem-like, Prog-like, S100hiGMP-like, and GMP-like cells on Day 0 (red, $n = 3$ samples) and Day 2 after ATRA treatment (blue, $n = 3$ samples). The *P*-values were calculated using Student's *t*-test. A one-sided *P*-value was calculated. In the boxplot, a black line within the box marks the median. The bottom and top of the box are located at the 25th and 75th percentiles, respectively. **e** Comparison of the Primitive-to-Mature scores of APL blasts on Day 0 (red, $n = 13,966$ cells) and Day 2 after ATRA treatment (blue, $n = 10,852$ cells). **f** Comparison of APL stem-like cell percentages (left panel) and stemness scores (right panel) on Day 0 (in red, $n = 7$ samples), Day 2 (in blue, $n = 7$ samples treated with ATRA alone), and Day 5 (in orange, $n = 5$ samples treated with ATRA + ATO). Notably, within the boxplot shown

in (**e** and **f**), a black line marks the median, and the bottom and top of the box are located at the 25th and 75th percentiles, respectively. The bars represent values more than 1.5 times the interquartile range from the border of each box. The *P*-values were calculated using the Wilcoxon rank-sum test. Two-sided *P*-values were calculated. **g** Heatmap showing the normalized expression of stemness CD markers and TFs in APL stem-like cells, Prog-like, S100hiGMP-like, and GMP-like clusters on Day 0 (red) and Day 2 after ATRA therapy (blue). **h** Heatmap showing the normalized expression of differentiation markers and hematopoietic TFs in APL stem-like cells, Prog-like, S100hiGMP-like, and GMP-like clusters on Day 0 (red) and Day 2 (blue) after ATRA therapy. **i** Heatmap illustrating the log2(fold changes) (log2FC) of gene expression levels for stemness-associated CD markers and TFs between APL samples on Day 0 and Day 2 (Day 2 vs. Day 0) in different APL patients. **j** Heatmap illustrating the log2FC of gene expression levels for differentiation markers and hematopoietic TFs between APL samples on Day 0 and Day 2 (Day 2 vs. Day 0) in different APL patients. Gray represents patients alive after induction therapy, and black for patients with early death.

importantly, we developed an APL-specific stemness score that proved to be a robust independent risk factor with superior predictive power for poor prognosis and early death, surpassing traditional risk factors, such as WBC, FLT3-ITD, and the Sanz score.

Our results provide valuable single-cell insights into the identification of leukemic stem cells in APL. Our scRNA-seq analysis revealed a small portion of stem-like leukemic cells (C15) directly from APL patients, expressing stemness markers (CD200, CD99, and CD9) and PML/RARα, and sitting at the top of the differentiation trajectories in the APL hierarchy. APL LSCs have not been well defined, partly due to a lower density of the CD34 expression on APL cells compared to other forms of AML[55]. Interestingly, previous studies have reported variable CD34 expression in APL, with frequencies ranging from 20% to 31%[56–58] and occasionally as high as 43% when considering a low cutoff level[59]. In our study, we indeed observed CD34 expression in a subset of APL stem-like cells. Detailed inspection showed that these CD34-positive cells within APL stem-like cells predominantly originated from three patients who were CD34-positive based on their immunotyping results. This observation aligns with the known variability of CD34 expression in APL patients. Notably, LSCs can be found in both CD34-positive and CD34-negative populations[60]. In NPM1-mutated AML (commonly CD34-negative), LSCs were found in both CD34+ and CD34− cells, suggesting that their presence is not limited to CD34+ progenitor cells[60,61]. Furthermore, CD200 was identified as a novel LSC marker and highly expressed in both CD34+ and CD34− LSCs, including those with mutant NPM1[30], further validating our identification of leukemic stem-like cells in APL for its characteristic expression of CD200. Furthermore, APL stem-like cells co-expressed lymphoid-related genes, such as T-lineage-affiliated glycoprotein CD2, also supporting its more primitive status[33].

Based on the discovery and characterization of APL stem-like cells, we introduced the APL stemness score, which has demonstrated the ability to predict the prognosis and risk of APL, notably with regards to early death in APL patients. The concept of leukemic stemness as a clinical predictor has gained growing attention in the field of AML. Various measures have been developed to evaluate leukemic stemness in non-APL AML, including metrics like the LSC17 score[50] and the frequency of leukemic progenitor cells. However, applying these measures to APL has been challenging due to the limited understanding of APL stem cells. Our study marks the endeavor to characterize APL stem-like cells and establish a measure based on APL leukemic stemness. In comparison to other commonly used predictors, such as high WBC counts and FLT3-ITD status, the APL stemness score exhibited superior performance in predicting poor prognosis and early death. This superiority can be partially attributed to the central role of leukemic stemness in APL, as it significantly contributes to the development of the disease. Incorporating our stemness score with the examination of WBC levels and aggressive mutations offers a valuable addition to the current APL risk assessment. While the detection of WBC levels helps clinicians evaluate risk promptly, our stemness score may serve as a reliable measure for more intensive surveillance, which is both practical and of clinical relevance.

Early death remains a formidable obstacle to achieving favorable outcomes in APL patients[62], and considerable research efforts have been made to investigate its underlying mechanisms[53,63–65]. Our data highlight the multifaceted impacts of stem-like cells on early death in APL, encompassing a relatively higher proportion of stem-like cells ("quantity" aspect), elevated expression of stemness-associated genes, and persistence of the stemness program ("quality" aspect) post-ATRA program in patients who experienced early death compared to those who achieved complete remission. Several studies have linked stemness genes, such as *CD2* and *CD9*, with an increased risk of thrombosis and coagulopathy[66–68], the common causes of early death in APL. Moreover, many stemness genes encoding adhesion molecules, such as *CD99* and *CD9*, have been reported to play a role in leukocyte migration[69,70], potentially contributing to extramedullary infiltration. Further investigation is required to comprehensively understand these underlying mechanisms. We also acknowledge that early death is a complex event influenced by numerous factors beyond the stemness score, such as the tumor microenvironment, cooperating mutations, epigenetic modifications, and various clinical variables such as the patient performance status (ECOG-PS) and the treatment protocol. Our preliminary comparative analysis, using cohort data, suggested that the IL8-related pathway and abnormal metabolic processes might also play a role in influencing the survival outcome of patients with higher stemness scores (Supplementary Fig. 21). Our results highlighted the leukemic stemness as one of the critical factors for early death in APL, which shed deeper insights into the complicated mechanisms underlying early death in APL and provide promising targets for mitigating the risk of early death.

Our study has several limitations that warrant consideration. First, given the complex effects of ATRA on primitive stem cells[71], further investigations to validate and explore the impact of ATRA on APL stem-like cells we identified are required. Second, expanding the sample size, particularly for patients experiencing early death, is essential to robustly confirm our findings regarding the prognostic significance of leukemic stemness in early death.

In conclusion, our work provides valuable single-cell insights into APL leukemogenesis by comprehensively elucidating intratumoral heterogeneity, the cellular composition hierarchy in APL, and their differential responses to ATRA therapy in vivo, which may contribute to early death in APL patients. We have characterized a small subpopulation of APL stem-like cells at the single-cell level, closely correlated with FLT3-ITD and poor prognosis, laying the foundation for further exploration of cellular therapeutic targeting strategies. Future studies will be geared towards elucidating the biological mechanisms of APL stem-like cells to extend the success of APL-targeted therapy to other cancers.

## Methods

### Patient samples

Bone marrow (BM) samples were collected according to the Declaration of Helsinki at the initial diagnosis of APL. Patients were randomly recruited in the clinical setting and we also confirmed that the patients in each analysis were representative of the broader APL patient demographic. The selection/recruitment was also contingent upon the availability of high-quality samples, which is a prerequisite for the reliable scRNA-seq analysis that our study relies on. Written informed consent was obtained from the patient allowing for the publication of clinical information, and ethical approval was obtained from the Ethics Committees of Ruijin Hospital, Shanghai Jiao Tong University School of Medicine (2021/154).

### Single-cell sequencing alignment and data preprocessing

Single-cell RNA sequencing data (10X Genomics) from 23 healthy BM samples were obtained from the Gene Expression Omnibus (GEO) public database with accession IDs of GSE120221[72] and GSE130116[73]. Raw sequencing data of healthy BM cells were downloaded and converted into FASTQ format using SRA-toolkit (version 2.11.0). Our scRNA-seq data for the 16 APL BM samples were generated through the separate scRNA-seq experiments and were integrated to explore cellular heterogeneity in APL cells. To obtain gene expression matrices from both healthy and APL BM cells, we used cellranger (10X Genomics, default settings, version 6.0.2) to align the scRNA-seq data to the human GRCh38 reference (2020-A version). Both the cellranger software and the human reference were downloaded from the 10X Genomics website (https://www.10xgenomics.com). The gene expression matrices were then imported into the Seurat R package[26]. For quality control, cells expressing fewer than 300 genes, having

unique counts exceeding 30,000 or falling below 500, expressing mitochondrial RNA exceeding 10%, or identified as doublets using DropletUtils (https://bioconductor.org/packages/DropletUtils/) were excluded. As a result, a total of 102,792 cells from healthy BM cells and a total of 136,497 from APL BM cells were retained for downstream data analysis.

### Annotation of cell populations of healthy BM cells

We used Seurat to perform highly variable gene identification, dimensionality reduction, graph-based clustering, and differentially expressed genes (DEGs) calculation. For identifying cell populations of healthy BM cells, the top 3000 highly variable genes were selected using the "FindVariableFeatures" function and enrolled for principal component analysis (PCA). The R package Harmony was used for sample-wide batch effect adjustment. Uniform Manifold Approximation and Projection (UMAP) and t-distributed Stochastic Neighbourhood Embedding (tSNE) were performed for dimensionality reduction using cell embeddings generated by Harmony. UMAP was chosen for visualization to better interpret the hematopoietic lineage. We first performed cell annotation on normal BM cells. Unsupervised clustering was performed on the 102,792 cells from 23 normal BM samples using the default parameters of the "FindClusters" function. After unsupervised clustering, 26 clusters were classified. Marker genes for each cluster were calculated using the "FindAllMarkers" function and defined under the following criteria: $\log_2$(fold changes) $(\log_2\text{FC}) > 0.58$ (FC > 1.5), min.pct >0.1, and adjusted $P < 0.05$. Cells were annotated using both machine-learning-based software SingleR and high expression of canonical hematopoietic markers (i.e., *CD34* for hematopoietic stem/progenitor cells (HSPCs), *CD14* for monocytes, *CD1C* for dendritic cells, *CD3* for T cells, *CD79A* for B cells, and *CA1* for erythroid cells) in each cluster. Finally, a total of seven major cell populations (HSPCs, GMPs, monocytes, DCs, B cells, T/NK cells, and erythroid cells) from normal BM cells were identified.

### Characterizing malignant APL blasts through integration with healthy BM cells and batch effect correction using Harmony

To characterize malignant APL blasts, we performed the integration of the scRNA-seq data from both APL and normal BM cells. We first identified the top 3,000 highly variable genes across APL and healthy BM cells, and then adjusted batch effects using Harmony. UMAP was performed to visualize the cell distribution of healthy and APL BM cells before (Supplementary Fig. 22) and after batch effect adjustment (Fig. 1b). The classification of APL blasts was determined based on the following criterion: cells were grouped into a distinct cluster that was clearly separated from cell populations of healthy BM cells, as observed in the UMAP plot. This classification was further validated using the following methods: (i) targeted scRNA-seq analysis, (ii) examination of the expression of canonical APL markers (such as clinical immunophenotype markers like MPO) and other genes like azurophilic granule genes (*AZU1*, *ELANE*, and *CTSG*); and (iii) annotation as progenitor cells using SingleR.

### Differentially expressed genes analysis and transcription factors activities estimation of scRNA-seq

Differentially expressed genes (DEGs) between different cell populations were calculated using the "FindMarkers" function in Seurat with logfc.threshold set to 0.01 and min.pct set to 0.01. Significance level of DEGs was set to $\log_2\text{FC} > 0.26$ (FC > 1.2) and adjusted $P < 0.05$. Two methods, VIPER[28] and DoRothEA[74], were used to estimate the protein activities of transcription factors (TFs). The msviper function from VIPER package was used for master regulator inference analysis and visualization. $P < 0.05$ was set as the significance level to demonstrate the difference of TF activities between different cell types.

### Constructing single-cell trajectories of APL blast populations

The APL blasts were subjected to downstream analysis using Seurat. To ensure robust results and account for potential batch effects across samples, we applied the Harmony algorithm for batch correlation. Subsequently, UMAP was employed to visually demonstrate the effectiveness of our batch correction (Supplementary Fig. 23). Additionally, the results in Supplementary Fig. 23b, c demonstrated that factors such as age and gender did not affect our identified clusters of APL blasts. To estimate the potential dynamic process of cell differentiation in APL blasts, we performed RNA velocity analysis. Velocyto was used to run the RNA velocity analysis and generate the spliced, unspliced, and ambiguous fractions of each cell. A likelihood-based dynamical model was utilized to learn the transcriptional states and infer intratumoral differentiation of APL blasts using scVelo[75]. To gain additional insights into genes during differentiation, the pseudotime transitional trajectory of the four APL blast cell populations was utilized using the R package Monocle2 (version 2.24.0)[76]. To eliminate the effects of cell cycle blasts and better interpret intratumoral heterogeneity, cells that were APL stem-like, Prog-like, S100$^{\text{hi}}$GMP-like and GMP-like were enrolled to construct the trajectory. To reduce computation time, the cell number of each cell type was down-sampled to 2,000. The DDRTree method was used to perform dimensionality reduction based on the top 30 DEGs in each cell type. Based on the trajectory result, APL stem-like cells were defined as the initiating point of the trajectory. Differentially expressed TFs in S100$^{\text{hi}}$GMP-like and GMP-like branches were calculated using the BEAM subprogram in Monocle2. Q-value < 0.05 was used to filter TFs. The visualization was performed via the plot_genes_branched_heatmap function.

### Targeted single-cell RNA sequencing data analysis

To validate the expression level of PML/RARα and FLT3-ITD transcripts, two samples (APL06 and APL08) were performed for targeted single-cell RNA sequencing (scTarget; Singleron Biotechnologies) and bulk RNA-seq. For gene expression analysis and cell type annotation of scTarget, gene expression matrices of cells were generated using CeleScope (https://github.com/singleron-RD/CeleScope) with default parameters. We used the K-nearest neighbor (KNN)[23] algorithm to predict cell types, and the calculation steps were described as follows: (i) build a single-cell-based expression reference using the pre-annotated 16 APL blasts (10X Genomics); (ii) integrate the expression profiles of the 16 APL blasts and the two scTarget samples. Batch effects adjustment was performed across samples using Harmony; (iii) for each cell from scTarget, the top 50 nearest cells were calculated using the KNN algorithm by the R package BiocNeighbors; (iv) the annotations were based on the highest frequency of the annotated nearest cell types; (v) cells from scTarget were projected onto the UMAP of APL blasts for visualization. Next, for the identification of PML/RARα and FLT3-ITD transcripts, the calculation steps were described as follows: (i) build a reference that contained wild-type gene regions of *PML*, *RARA*, and *FLT3*, and the different isoforms of PML/RARα fusion transcripts; (ii) FLT3-ITD were called from bulk RNA-seq using RNAmut; (iii) preprocessing steps of scTarget data to discover fusions/mutations were performed using CeleScope; (iv) a 12-bp of nucleotides across the PML/RARα fusion point was used to scan the reads that mapped onto *PML* and *RARA* genes and identify the expression level of PML/RARα transcripts of each cell; (v) detection of FLT3-ITD according to the mutated nucleotides obtained from bulk RNA-seq; (vi) merge the expression level of PML/RARα and FLT3-ITD transcripts with gene expression data. There were, however, certain limitations in scTarget. For example, the presence of dropouts, a well-known challenge in single-cell analysis[77], implies that only a fraction of the transcriptome in each cell may have been captured. Therefore, there is room for improving the sensitivity and accuracy of scTarget in detecting targeted gene expression at the single-cell level.

### Bulk RNA sequencing, alignment, and data analysis

Raw FASTQ files of bulk RNA sequencing (RNA-seq) data were aligned to the human reference genome GRCh38 (release 38). The human reference genome and its annotation file were downloaded from the GENCODE database (https://www.gencodegenes.org/). Salmon (version 1.5.1) was used to generate the count and transcripts per kilobase of the exon model per million mapped reads (TPM) matrix. For calculating differentially expressed genes, the limma package was used with significance level settings of adjusted $P < 0.05$ and $\log_2 FC > 0.58$ or $< -0.58$. The R package pheatmap (https://CRAN.R-project.org/package=pheatmap) was used to perform and visualize hierarchical clustering in the heatmap plot.

### Deconvolution analysis and construction of the LASSO score model

Deconvolution analysis was performed to estimate APL cell populations from bulk RNA-seq data. The reference was constructed from scRNA-seq of the 16 de novo APLs. 10,000 cells were randomly selected from the total 136,497 cells of APL BM cells. The normalized expression profile was uploaded to CIBERSORTx (https://cibersortx.stanford.edu/) and the generation of the signature matrix of the six APL branches (stem-like, Prog-like, S100hiGMP-like, GMP-like, cycling GMP-like, and MDP-like) and three non-APL populations (B cells, T/NK cells, and erythroid cells) was performed. Then the signature matrix containing 3388 genes was generated and enrolled for deconvolution analysis. Support Vector Regression (SVR) was performed to estimate the coefficients of each cell type from the bulk TPM matrix using R package e1071 (https://CRAN.R-project.org/package=e1071). The coefficients could be used to reflect the abundance of cell types in bulk RNA-seq. In this study, we had data from 12 patients who had both scRNA-seq and matched bulk RNA-seq data available. To estimate the cell proportions, we employed the linear regression model to link the coefficients calculated from the bulk RNA-seq and the observed cell proportions from scRNA-seq. For evaluation, leave-one-out (LOO) cross-validation was performed using the 12 patients as the "ground truth" to assess the accuracy and reliability of the model. The model was then applied to the 323 de novo APL cohort (GSE172057)[53] to estimate cell proportions. For better application on bulk RNA-seq, we constructed a least absolute shrinkage and selection operator (LASSO) based score model to reveal the stemness score of APL. This model considered two factors: (i) the proportion of APL stem-like cells by using genes that were exclusively expressed in APL stem-like cells relative to other APL blast populations (quantity-driven); and (ii) the characteristics of APL stem-like cells by analyzing differentially expressed genes (DEGs) between APL stem-like cells and normal HSPCs (quality-driven). From this analysis, 898 significant DEGs were selected for use in the LASSO regression model. The optimal value of the penalty parameter λ was determined through 10-fold cross-validation, using the R package glmnet (https://CRAN.R-project.org/package=glmnet). The top 10% estimated proportions of APL stem-like were used to separate the samples. Finally, eleven genes were selected for constructing the APL stemness score model.

### Analysis of APL BM cells after ATRA treatment

For scRNA-seq analysis, cells from Day 0 and Day 2 (after ATRA treatment) were integrated using Seurat. Batch effects were adjusted using Harmony. In the realm of cell type annotation, the cell types identified in Day 0 samples were consistent with annotations from the original analysis of 16 de novo APL scRNA-seq datasets. Subsequently, for Day 2 samples, we annotated cell types based on the KNN algorithm, considering their top 50 nearest cells in Day 0. For bulk RNA-seq analysis, we leveraged deconvolution analysis to estimate the proportions of the 9 APL cell types, providing insights into the cellular composition within the samples.

### Functional enrichment analysis and pathway crosstalk analysis

Gene ontology (GO) enrichment and gene set enrichment analysis (GSEA) were performed using the clusterProfiler package. Gene sets used in GSEA were downloaded from MSigDB. We used single-sample GSEA (ssGSEA) to evaluate the expression level of pathways in each sample through the GSVA package. The public gene sets used in GSEA were downloaded from MSigDB, including Hallmark gene sets (H) and KEGG gene sets (C2). To perform GSEA analysis with PML/RARα targets generated from CUT&Tag (GSE195776)[78], we initiated NES evaluation by randomly selecting 500 PML/RARα targets with different random seeds spanning from 1 to 100 (Supplementary Fig. 24). Genes included in the GSEA analysis were ranked based on their $\log_2 FC$ between APL stem-like cells and normal HSPCs. The ultimate gene set of PML/RARα targets was defined according to the top 500 PML/RARα targets, determined by adjusted $P$-values derived from CUT&Tag. We employed OpenXGR[79] to identify pathway crosstalk through integrative analysis of differential expression data (PML/RARα targets expressed in APL stem-like cells compared with HSPCs) and KEGG-derived pathway interaction data. Enrichr[80] was also conducted for functional enrichment analysis of differentially expressed genes. The differentially expressed pathways depicted were the most relevant to hematological malignancies or stemness.

### Statistics and reproducibility

Statistical analysis was performed using R version 4.2.1. The detailed descriptions of the statistical tests used were provided in the legends of the corresponding figures. APL patient samples were randomly recruited in the clinical setting without sample size calculated. All analyses are reproducible using codes available through the project-dedicated website 'TACH' (http://www.genetictargets.com/tach).

### Reporting summary

Further information on research design is available in the Nature Portfolio Reporting Summary linked to this article.

## Data availability

The raw sequencing data reported in this paper have been deposited in the Genome Sequence Archive in National Genomics Data Center, China National Center for Bioinformation/Beijing Institute of Genomics, Chinese Academy of Sciences (https://ngdc.cncb.ac.cn/gsa-human). These data are accessible under the accession number 'GSA-Human: HRA003777'. These data are under controlled access by human privacy regulations and are only available for research purposes. Access to the data can be granted following approval from the Data Access Committee of the GSA-human database, as detailed at https://ngdc.cncb.ac.cn/gsa-human/document/GSA-Human_Request_Guide_for_Users_us.pdf. Data are accessible to researchers who meet the criteria for access as defined by the GSA-human database guidelines. Access requests are usually processed within approximately four weeks and data will be available for three months once access is granted. All sequencing data, including scRNA-seq and bulk RNA-seq data, are also available in NODE under the accession number OEP003829. The public datasets utilized in this study are available in the GEO database. These include scRNA-seq data from 23 healthy BM samples (accession code GSE120221 and GSE130116), the APL cohort data (GSE172057), and the PML/RARα target data generated from CUT&Tag (GSE195776). Accession links are as follows: GSE120221[72], GSE130116[73], GSE172057[53], and GSE195776[78]. Source data are provided as a Source Data file within the paper, also accessible through the project-dedicated website 'TACH' (http://www.genetictargets.com/tach). The remaining data are available within the Article, Supplementary Information, or Source Data file.

## Code availability

The single-cell RNA data were processed using cellranger (https://www.10xgenomics.com, version 6.0.2) and analyzed using the R package Seurat (https://satijalab.org/seurat, version 4.3.0). The codes used in the paper are available on GitHub (https://nrctm-bioinfo.github.io/APL_stemness) and Zenodo (https://doi.org/10.5281/zenodo.10437695)[81].

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

## Acknowledgements

This work was supported by the National Natural Science Foundation of China (81890994, 82350710226, 82370178, and 32170663) and the National Key R&D Program of China (2023YFA1800401).

## Author contributions

K.W. designed the experiments; W.J. performed the experiments; Y.D. and H.F. analyzed the data; W.J., H.H.Z., and F.D. collected clinical

samples; L.C., H.H.Z., H.M.Z., and J.L. followed up with the patients; K.W. and H.F. supervised the study and analysis; K.W., H.F., W.J., and Y.D. wrote the manuscript; Z.C., S.C., and G.M. gave conceptual advice; and all authors discussed the results and implications and reviewed the manuscript.

## Competing interests

The authors declare no competing interests.
