## [Peer Review File · Nature Communications]

Cellular hierarchy insights reveal leukemic stem-like cells and early death risk in acute promyelocytic leukemiaReviewers' Comments:

Reviewer #1:

Remarks to the Author:

Resolving single-cell intratumoral heterogeneity of acute promyelocytic leukemia identifies leukemic stem-like cells.... by Jin et al.,

The authors provide an interesting and scientifically sound study in AML-M3 or acute promyelocytic leukemia (APL) using scRNAseq in primary APL samples and an in-depth bioinformatic data elaboration. This allowed them to define several different sub-populations where PML::RARA have supposedly different effects. They here report a differentiation trajectory starting from a stem cell-like population whose biological characteristics are governed by PML::RARA. They defined a "score" for predicting early death in APL based on the stemness they found, which PML::RARA and FLT3-ITD mainly determined. Eventually, they defined the early transcriptional program induced by ATRA. As often in APL, the experimental evidence confirms things anticipated by previous findings, many but not only from the clinic. The same is here; although the authors elegantly show the hierarchy and the role of PML::RARA not only for transcription, most of the things shown here look familiar, at least since the first decade of this century. E.g., the stem cell hierarchy and the role of PML::RARA for the ATRA-induced differentiation (shown not only by the references cited by the authors). Therefore, this report lacks real novelty which could advance the APL and the leukemia field, apart from the use of the scRNAseq for the definition of cell populations and their transcriptional programs within leukemia. The authors focus on the APL diagnosis of primary APL samples and the induction therapy based on ATRA and its pro-differentiation activity. It takes a while to understand that the author investigated the effect of ATRA on "early death." This has to be stated. Furthermore, the report completely overlooked the effect of ATRA on the self-renewal of stem cells (normal and malignant) as reported already in the early 90ties of the last century and again in the first decade of this century. ATRA does not induce only differentiation; otherwise, all APL patients not suffering early death would be cured by ATRA monotherapy. Unfortunately, nearly all relapse within a short time. The next focus, at least for APL patients, is "early death" by bleeding. In an era of ATO/ATRA, you are most likely cured if you overcome this thread, as the studies showed (e.g., LoCoco et al., 2013). Unfortunately, bleeding complications are often an exclusion criterium for these studies (To this reviewer, it is not clear how many APL patients are lost on the way towards a clinical study). APL is by some still considered a medical emergency because of its bleeding problems due to a consumption coagulopathy. The authors mentioned early death in terms of prognosis but not of mechanisms. The authors defined a stemness score to distinguish high-risk from normal-risk patients. The question remains whether FLT3-ITD and high WBC would do the same job much quicker.

Major points:

Abstract.

Well written, but the study's aims have to be made clear.

Main/Introduction

The introduction contains some misleading or unclear statements, and not all are supported by references. This reviewer would re-write it, giving it a better order and clarifying things.

- line 59: Did the authors mean that the effectiveness of ATO and ATRA depends on the target cell? This depends on PML::RARA, because ATO and ATRA degrade or cleave PML::RARA, respectively, independently of the cell in which it is expressed.
- line 66: This reviewer disagrees with the authors: Next focus for a high-risk APL patient is the survival of the bleeding and the differentiation syndrome as this determines the risk of APL. The report lacked any discussion about potential mechanisms that could be responsible for the bleeding.
- Independently of the origin of the APL-initiating leukemic stem cell, PML::RARA is the major determinant efficiently targeted by low-dose ATO. Therefore, if the study aims to determine the effect of ATRA on APL-initiating leukemic stem cells in an early stage to avoid high-risk patients experiencing

early death, it must be stated more clearly.

- line 82: This reviewer disagrees with the authors. Many more APL models contributed to current knowledge of APL, and NB4 cells are one of the worst.
- line 100: that PML::RARA determines the APL initiating leukemic stem cell is known.
- line 106: that ATRA induces differentiation of primitive APL blasts has been known since the nineties of the last century. This is the basis of the use of ATRA in APL. This statement needs to be clarified.

Results

In general, many statements need references. In addition, many "likely" indicate much speculation, which is evident if data came from bioinformatics without any experimental validation.

Figure 1/Text for Figure 1

- line 127: define what is meant by early death in APL (it is slightly different in APL than in other AMLs)
- line 129: define the source of the 136,497 cells. A pool of 16 patients? The sentence "...generated from the BM at disease onset..." is misleading.
- line 133: same as above.
- How is bias avoided: age, gender, etc., during the selection of patients for different analyses? For each analysis, a different number of patients was selected. Which were the criteria for the selection? It is crucial for the clinical significance of the results.
- line 151: Published findings need the reference.
- line 162: a widely used diagnostic marker for APL - reference?
- line 175 and others: significantly - typically, this term is used if there is a p-value. Does the software used provide it?
- line 189-193: Likely?? what does it mean?

Figure 2/Text for Figure 2

Many statements need references (e.g., "...known maker genes.."). The entire interpretation of data is based on bioinformatic data without any validation.

- line 199: A stem cell is supposed to initiate a hierarchy?
- Are PML::RARA target genes only expressed in the APL-stem-like cell cluster? Then, the other cell clusters have to be tested for the correctness of the interpretation. If PML::RARA does things in a cell-type-dependent manner, it would be a breakthrough result, but it has to be proved.

Figure 3/Text for Figure 3

- Are NB4 cells the same as APL? As there were criticisms in the introduction about using NB4 as an APL model, it has to be proved.
- NB4 compared to APL-stem-like cells: Explain the apparent expression of PML::RARA target genes in stem-like cells but not in further differentiated clusters, although NB4 cells have shown their promoter occupancy.
- line 295: are TGIF1 and HIF1A really co-factors for PML::RARA?
- line 302: it is known since the 90ties of the last century. This reviewer would change the text to a better conclusion.
- line 326: do you suggest by this that FLT3-ITD precedes the t(15;17)?
- line 332: how did you select the FLT3-ITD positive APL patients; how was FLT3-ITD expression? Were these experiments performed with FLT3-ITD-negative controls?
- line 334: Speculation is for the discussion.

Figure 4 and 5/Text for Figure 4 and 5

It is not surprising that the tumor load (higher percentage of APL blasts) increases with the number of APL stem-like cells, which is correlated with the WBC. This bioinformatic analysis will not substitute the simple WBC and the check for FLT3-ITD.

- line 389: BM infiltration. Do you think that the origin of APL is outside the BM? This has to be proved!

- line 395: Explain because only experts on PML::RARA know what is meant.
- 11 gene stemness score by LASSO: Is the prognostic relevance of this score better than the one in use for OS, EFS, and DFS? It is great to have a better prognosis/prediction of the risk for early death.

Figure 6/Text for Figure 6

The analysis here lacks negative controls.

- line 438-40: Be more careful with statements. Is the high risk for bleeding of a high-risk APL patient due to the progression of the disease? What is the reason for the bleeding/early death, how long does it take for ATRA to overcome it, and how do you treat it in the meantime?
- It is not explained why day two of ATRA treatment.
- The analysis was performed without considering what ATRA is supposed to do with PML::RARA - cleavage. Therefore, control of healthy BMs would be more than helpful and interesting. Does ATRA normalize the transcription, or is it completely different?

Discussion

The discussion can be shortened. Take out all other AMLs as they further complicate the already complicated picture. The only significant parts at the moment are

1. lines 535-543
2. lines 589-610

This reviewer would discuss the following:

- How do these studies fit with other theories of the mechanisms for the bleeding in high-risk APL. Are these theories in line with the findings that high-risk patients have a more pronounced stem-like cell portion?
- How would an APL patient benefit from the findings, and how does it compare with what is already available and in use.

Methods

Patient samples

- provide a number of the Ethic approval. Each approval is supposed to have a number needed for tracking.
- It appears to this reviewer that the authors made much effort to make this study reproducible to others.

Reviewer #2:

Remarks to the Author:

In this paper, the authors use single-cell RNA-seq (scRNA-seq) data from 16 APL patients to examine the cellular composition and differentiation hierarchy of APL. They identify a stem-like cell population with projected differentiation trajectories and construct an 11-gene stemness score associated with the proportion of APL stem-like cells and survival outcomes. Finally, they examine transcriptional changes before and after ATRA treatment and identify distinct transcriptional programs in APL patients with early death.

Overall, this work provides a valuable resource for studying APL and potential mechanisms of therapeutic resistance leading to early death. As APL has a very good prognosis in the majority of patients with highly effective targeting of PML-RARA by ATRA and ATO, the biggest potential impact of this work would be in developing tools for identifying the minority of patients at high risk of induction failure and early death. While the authors do provide some data and first steps in this regard, there are some issues that if addressed would improve the overall impact of this work. In addition, some of the statements in results and discussion are not supported by the data and should be revised.

Specific comments

1. In general the methods are all using well-accepted tools and the authors provide their code in a well documented format. The authors should be careful not to use definitive language in describing inferred trajectories. As well, the language regarding the roles of RUNX1 etc and CEBP family members among the total genes listed in Figure 2f seems overstated without supporting functional analysis. The statement that "PML-RAR α was...widely expressed in all 6 branches of APL blasts, especially the stem-like cluster, further supporting the hypothesis that APL might initiate from these leukemic stem-like cells" is not supported by Figure 2h, where the expression level in the stem-like cluster looks very similar to those of the CMP/GMP clusters.

2. It would be good to provide in supplementary data some plots to show that their batch correction is sufficient – for example UMAP plots showing sample of origin, gender, age, and other confounding variables, or cumulative barplots/pie charts showing whether their clusters are over-represented for specific confounding variables.

3. In several places the authors make statements regarding the initiating role of PML-RAR α in APL based on enrichment of stemness-related genes in the APL stem-like cell cluster (e.g. p12, "Collectively, these findings demonstrate that PML-RAR α is a critical determinant in APL initiation"). In fact, the demonstration of the stemness properties of these cells does not provide evidence that PML-RAR α is the initiating event. Their work does not address the possibility that there may be an earlier initiating event that precedes PML-RAR α . Furthermore, on p20 the authors state that their results provide insight into the origin of APL cells (i.e. cell of origin), when they have not addressed this in their experimental work. For example, can they exclude that acquisition of PML-RAR α in a more mature progenitors may confer stem-like properties to generate the stem-like cells that they observed?

4. In Figure 3, the authors carry out GSEA with a very large gene set (~7,500 genes). For GSEA a maximum gene set size cutoff of 500 genes is recommended for a dataset of 10-20,000 genes. Gene sets that are too large (or small) relative to total size can affect normalization. Perhaps they could consider randomly sampling multiple permutations of (for example) 500 genes from their gene set and plotting the distribution of resulting normalized enrichment scores, or take the ~500 genes which show the strongest enrichment of cut'n'tag signal at their promoters.

Also in Figure 3a-e, the authors show high expression of APL-specific leukemia stemness genes (expressed in APL stem-like cluster and not normal HSPC cluster) that are also PML-RAR α targets suggesting that these pathways are important for stemness in this population. It would be informative to show whether these genes are also enriched in the other APL clusters (CMP-like, GMP-like, etc.) or whether this is specific to the stem-like cluster of APL.

5. Regarding the 11-gene scoring model, more detail should be provided as to the criteria used for selecting these 11 genes. Is it possible to provide a HR for the prediction of early death? It appears from Figure 5b that although a higher score was associated with a higher incidence of early death, the majority of patients with a score above their cutoff did not experience early death. As such, how do the authors envision such a tool could be used in the clinic? This should be discussed.

6. Regarding the transcriptomic changes following ATRA treatment, the differentiation changes are not surprising and are in keeping with the success of current treatments. The greatest potential impact of this work is in understanding the mechanisms underlying treatment resistance and early death. Figure 6i and 6j provide some evidence of distinct transcriptional changes after ATRA treatment in patients with early death, but the numbers are too small to draw any firm conclusions (n=2 scRNA-seq, n=9 bulk RNA-seq with 2 patients who experienced early death). They state in the Discussion that their results "provide new insights into the underlying mechanisms" but they have not in fact addressed this in any detail. It would be much more powerful to analyze a larger number of patients (including more with early death) pre and post treatment to elucidate differences in expression of TFs and identify potential pathways that could be targeted in patients with early death.

Reviewer #3:

Remarks to the Author:

In this manuscript, Jin and colleagues present a comprehensive overview of the single-cell RNA sequencing landscape of acute promyelocytic leukemia (APL) patient blasts. In particular, the authors studied: BM from 16 patients at diagnosis (23 healthy donor BM from published datasets as control); 2 scTarget to determine the presence of PML-RARa or FLT-ITD; validation of cellular composition and clinical associations by "Cybersort" on bulk RNAseq data from 323 newly diagnosed APL patients; 3+9 patients pre-/post ATRA induction therapy using scRNAseq and de-convolution analysis, respectively.

The authors define a small subset of PML-RARa+ stem-like cells, which appear at the top of the APL hierarchy (based on RNA velocity and pseudotime analysis) and are associated with higher WBC counts, lower platelet counts and the presence of FLT-ITD. An 11-gene stemness score derived by the LASSO algorithm from the bulk RNAseq cohort correlated with poorer outcome in multivariate analysis. The authors suggest that the risk of early death is linked to non-response of the stem-like compartment to ATRA, which is an intriguing hypothesis, yet not fully supported by the data.

This dataset represents an important resource for future studies, considering that many of the single cell RNA sequencing studies published so far have excluded APL cases. Hence, there is limited high resolution data available on the population hierarchies, differentiation trajectories and population dynamics during therapy, as well as the role of FLT-ITD in this specific AML entity. The study is well conducted, the scRNAseq and bioinformatic analyses appear state-of-the-art using validated, appropriate and widely used approaches in the single-cell community. The manuscript is overall well written, even though some English language revision may benefit the reader, including article and main paragraph titles where sometimes the underlying message is not easily conceived.

Limitations of the work include the descriptive nature of the data. Validation relies on correlation with external datasets and clinical outcomes, which would require future independent verification on a higher number of cases, ideally in a prospective setting. These limitations should be clearly stated in the discussion. In general, across the text, many conclusions relying on correlative data should be toned down, in the absence of data demonstrating a causal relationship.

Major points:

- The authors define APL blasts in single-cell RNA sequencing based on subjectively chosen criteria described in the methods section, which could appear arbitrary. First, bias may occur if only APL blasts that cluster with each other are selected and those that cluster with other non-malignant myeloid or progenitor clusters are disregarded. Some smaller, rare subclusters of cells, such as the cells from APL patients labeled as part of cluster 7 in Fig. 1b,c, that appear to express both MPO and CD3E (or other small erythroid marker-expressing cell clusters) and do not co-cluster with other non-APL blasts, were not included. Although rare, and unlikely to affect the results presented in the manuscript, this critical point in cell selection should be discussed in more detail. Given that the authors also performed PML-RARa-targeted scRNAseq, they could use these data for a limited number of patients to validate their approach in APL blast calling.

- APL landscape annotation: Cluster 14 was called "CMP-like". However, recent single cell studies have questioned the existence of CMPs as a specific progenitor, as they may contain a mix of progenitors already committed to erythroid and myeloid fate, respectively, rather than a common progenitor. Did the authors find erythroid/MK gene among the marker genes in Cluster 14? Based on Fig 2b, they rather seem to form a continuum with Cluster 11. Moreover, how do the cycling GMP-like cells relate to the branches? Is this a mono- or bidirectional, reversible state originating from the mature or primitive APL subsets? Validating the differences in cell cycle between normal GMP and APL GMPs in a limited

number of patients, e.g. by flow cytometry, would strengthen the scRNAseq-based predictions.

- The mechanism, by which a high APL stemness signature acts as an independent predictor of early death is not entirely clear. How much do increased WBC count, decreased platelets and presence of FLT-ITD (all associated with a higher proportion of stem-like cells) contribute to this association? Is the inferred proportion of stem-like cells in the bulk RNA seq data also predictive of early death? It would also be helpful to include a table with the frequency distribution of the clusters broken down for each single patient in the 16 patients where scRNAseq has been performed. Do the 4 early deaths in this cohort show a higher proportion of stem-like cells?

- The longitudinal samples pre-/post ATRA suggest that a subset of patients fail to downregulate stemness genes, in stark contrast to the majority of patients where stem-like cells are readily depleted, already after 2 days of ATRA. This observation is intriguing and may support an independent prognostic value of stem-like cells. However, additional data are required to consolidate this finding, first-of-all, longitudinal analysis of a higher number of poor responders (the claim is based on n=2, whereby mostly APL11 upregulates stemness CD markers). It would also be interesting to measure the content of stem-like cells at timepoints beyond day 2, and after a second therapeutic agent (e.g. ATO or Idarubicin) has been added. Is the sub-optimal response a matter of quantity (very high baseline content of stem-like cells), or quality (any transcriptional differences between stem-like cells from day2 responders vs non-responders)? Can this be predicted based on transcriptional differences in the stem-like cluster already at diagnosis?

Minor points:

- Line 176: RUNX1, MYC, JAG1: are these associated to myeloid differentiation? These factors rather regulate HSC self-renewal/differentiation. Similarly, DNMT3A prominently methylates DNA rather than regulating histone methylation.

- Fig 5a/b: the cutoff for WBC needs to be clarified: is it $10 \times 10^9/L$?

- Line 326-327: "mutations, suggesting that FLT3-ITD might be more permissive of APL initiation and essential for maintaining leukemic stemness". Increased FLT3-ITD may expand the stem/progenitor compartment, but not be necessary or essential for maintaining leukemic stemness, as seen in many FLT3-ITD negative myeloid leukemias. The claim should be toned down.

- Line 387: please indicate whether the correlation between the stem-like cells and the % of APL blasts relates to bone marrow or blood blast counts

- Line 445/6: it is not entirely clear why the cellular composition of day2 samples was "determined by deconvolution analysis of scRNAseq". If the analysis is at single cell level, then the most straightforward way of comparing cell composition is to merge the day 2 data and diagnosis data into a single object and determine the cluster distribution for the individual samples. Could you please clarify?

- Line 541 – 548: "APL LSCs have not been well defined, to a certain extent, mainly because of the CD34-negative feature of APL blast cells..." From these statements, it seems that APL LSCs are mainly CD34-. However, in their dataset, the stem-like cells are clearly CD34+ (CD34 is among the top markers of cluster 15). Please clarify.

- Line 663 Projection of APL BM cells and characterization of malignant APL blasts: It is unclear why the authors use the "projection" term to describe the integration of the two datasets with harmony. Are the authors using a projection approach (for instance using MapQuery in Seurat) to project APL BM cells into the main data structure provided by Healthy BM cells, or are they using integration of the

two datasets correcting for batches using Harmony, ending up in a final integrated dataset? Please clarify. The source code does not help to understand this point.

- Line 734 Deconvolution analysis: Why downsampling to 10000 cells? Can the profile of smaller populations, like stem-like cells, be more affected from this procedure?

Reviewer #4:

Remarks to the Author:

In the manuscript "Resolving single-cell intratumoral heterogeneity of acute promyelocytic leukemia identifies leukemia stem-like cells and their impacts on early death and all-trans retinoic acid responses in vivo," Jin et al. apply single-cell RNA-seq and state-of-the-art computational analysis methods to characterize the cellular heterogeneity of primary Acute Promyelocytic Leukemia (APL) samples and how this relates to clinical outcomes. The authors identify a relatively rare population of APL cells that have stem-like characteristics based on the expression of a variety of stem cell associated cell surface markers. Using their single cell data, the authors go on to develop a computation approach to predict the prevalence of the stem-like cells in primary APL samples from bulk RNA-sequencing data. In addition, the authors show that treatment of APL patients with all-trans retinoic acid (ATRA) causes a rapid loss of the stem-like population and induction of differentiation programs in vivo. Furthermore, in the two patients that did not respond to treatment with ATRA, the authors show that numerous markers of stemness persisted. Overall, this study advances our understanding of the heterogeneity of APL and provides a prognostic molecular signature that is relatively straight forward to implement. In addition, this study provides a conceptual advance in how single-cell RNA-seq studies can be used to assess the efficacy of differentiation therapies in the future. However, several key points related to the regulation of the leukemia-stem-like cells and the differential responses of patients with APL to ATRA could be expounded on with additional analysis to prepare this manuscript for publication.

Major Concerns:

(1) The authors state that "With regard to APL, the next focus is to address whether leukemic cells driven by the same driver, PML/RARa, can exhibit different cellular states; if so, to what extent the cellular composition and transcriptional heterogeneity could impact the targeted therapy in APL." And say that "the stemness characteristics of APL stem-like cells were determined by PML/RARa." With regards to this point, the authors compare the list of 7309 PML/RARa target genes they previously identified by CUT&Tag in NB4 cells between the APL stem-like cells and normal HSPCs. As presented the analysis is important, but does not address whether the PML/RARa oncoprotein also contributes to intratumorally heterogeneity and is perhaps driving the stem-like state specifically.

-Related to this point: In figure 3c the authors compare the expression of several PML/RARa target genes between APL stem-like cells, HSPCs and GMPs. However, a proper description of the GMP population that is analyzed here is not included in the text or the figure legend. Are these normal GMPs or GMP-like cells?

-When comparing other APL cell types and normal cell types do the authors see a consistent increase in the PML/RARa target genes in APL cells across all cell types or is the upregulation of these genes specific to the stem-like cells?

-The authors need to provide a comparative analysis of the activity of PML/RARa target genes across the 18 clusters of APL cells identified in figure 2, or at least compare the activity between the 6 major APL cell types they identify (stem-like, CMP-like, S100hiGMP-like etc.).

(2) In figure 5 the authors provide a compelling case that the APL stemness score they develop predicts the abundance of stem-like cells in primary APL samples from bulk-RNA-seq data and that an

increased APL stemness score is indicative of an elevated risk of "early death." However, in figure 6, presumably all 9 of the bulk RNA-seq samples analyzed before and after ATRA had some proportion of APL stem-like cells, and yet some patients had early death while others did not.

-Did these bulk samples have differences in their APL stemness score?

-If not do the authors see any other differences in the samples at diagnosis that went on to undergo early death versus those that did not?

-A critical question that is left unanswered is why some tumors with high stemness scores respond to ATRA while others do not. Given the wealth of data collected in this study, can the authors provide further insight on this point.

(3) How the samples were profiled is not clear from this manuscript. Were the API samples all mixed and run together on a single cell RNA-seq platform? Or were they run in parallel reaction to ensure the donor of origin for each single cell profile is known?

-Related to the last point: The authors should provide a supplementary figure showing the APL and healthy samples before batch correction.

-Related to this point: If the donor of origin is known for each single cell profile, the authors should use this information to get an idea of the intertumoral heterogeneity in the cell states of individual API samples. Can the authors quantify the fraction of cells that fall into the 6 cell types (stem-like, CMP-like, S100hiGMP-like etc.) from each donor?

-Related to this point: Figure 3f is critical to making the point that FLT3-ITD promotes a stem-like cell state. However, how the authors were able to perform this analysis is not intuitive from the manuscript. Do the authors have separate genotyping data for each sample? If so, this needs to be clearly stated.

Minor Concerns:

-Line 31: "affect it's effectiveness". Please consider rewording for clarity.

-Line 94: "Our resource and findings are significant with fourfold." Please Reword.

-Figure 1g: This plot is not clear. I understand VIPER was used to generate this plot, but simply calling it a VIPER plot is not an acceptable description in the legend. Why are there two rows for each gene?

-Line 179: "In contrast, immune response-related functions were enriched for genes that were significantly downregulated in APL blasts." Suggested correction: "In contrast, genes that were significantly downregulated in APL blasts were enriched for immune response-related functions."

-Line 186: "...found that the regulatory activity of hematopoietic transcription factors (TFs) and cofactors..." The authors performed RNA-seq and it is not possible to infer the regulatory activity of these factors from this type of data alone. Please Reword.

- The marker gene set the authors include for HSPCs and/or LSCs are mostly cell surface markers including CD200, CD44, CD99, CD2 and FAM30A. The authors should extend this list to include other types of genes that are known to be functionally required for HSPC or LSCs, e.g. TFs.

-Figure 2f: In this heatmap, genes that show the highest signal in the center are presumably enriched in the stem-like cells, but this is not clear from the labeling.

-Figure 3a: Presumably the left heatmap is the PML/RARa CUT&Tag profile and the right heatmap is

the IgG negative control but this is not clear from the labeling. Also, the heatmap says it is centered on the TSS, but the legend says it is centered on the summit. Which is true?

-Adding a FLT3-ITD-targeted scRNA-seq projection onto the APL UMAP would be helpful to show the cells are also enriched throughout.

-Figure 4a: this figure shows that you have scRNA-seq samples matched with bulk RNA-seq data, but this point should be clarified in the text. This would clarify that matched samples were used as a "ground truth" to test the model.

-Line 411: "Remarkably, we found that a high APL stemness score was significantly associated with OS and EFS." This wording is confusing and suggests that a high APL score predicts patients will likely survive, when this is the opposite of what I infer from reading the rest of the text.

-Related to the last comment: In Figure 5b many of the samples that have a DFS/EFS/OS status of "Yes", also appear to have an Early death status of "Yes". Shouldn't these two categories be mutually exclusive?

-Line 456: "almost invisible" this is colloquial, please reword.

-Line 478: "MAFB mainly upregulated..." Suggested revision: "MAFB were upregulated..."

-Line 542: "mainly because of the CD34-negative feature of the APL blast cells." Figure 2b shows that CD34 is enriched in Clusters 15 and 16, please explain.

I hope you find these comments and concerns constructive and useful in revising your manuscript.

Response to Reviewer Comments

Reviewer #1, expertise in acute promyelocytic leukemia, leukemic stem cells and therapy (Remarks to the Author):

Resolving single-cell intratumoral heterogeneity of acute promyelocytic leukemia identifies leukemic stem-like cells.... by Jin et al.,

The authors provide an interesting and scientifically sound study in AML-M3 or acute promyelocytic leukemia (APL) using scRNAseq in primary APL samples and an in-depth bioinformatic data elaboration. This allowed them to define several different sub-populations where PML::RARA have supposedly different effects. They here report a differentiation trajectory starting from a stem cell-like population whose biological characteristics are governed by PML::RARA. They defined a "score" for predicting early death in APL based on the stemness they found, which PML::RARA and FLT3-ITD mainly determined. Eventually, they defined the early transcriptional program induced by ATRA.

As often in APL, the experimental evidence confirms things anticipated by previous findings, many but not only from the clinic. The same is here; although the authors elegantly show the hierarchy and the role of PML::RARA not only for transcription, most of the things shown here look familiar, at least since the first decade of this century. E.g., the stem cell hierarchy and the role of PML::RARA for the ATRA-induced differentiation (shown not only by the references cited by the authors). Therefore, this report lacks real novelty which could advance the APL and the leukemia field, apart from the use of the scRNAseq for the definition of cell populations and their transcriptional programs within leukemia.

The authors focus on the APL diagnosis of primary APL samples and the induction therapy based on ATRA and its pro-differentiation activity. It takes a while to understand that the author investigated the effect of ATRA on "early death." This has to be stated. Furthermore, the report completely overlooked the effect of ATRA on the self-renewal of stem cells (normal and malignant) as reported already in the early 90ties of the last century and again in the first decade of this century. ATRA does not induce only differentiation; otherwise, all APL patients not suffering early death would be cured by ATRA monotherapy. Unfortunately, nearly all relapse within a short time.

The next focus, at least for APL patients, is "early death" by bleeding. In an era of ATO/ATRA, you are most likely cured if you overcome this thread, as the studies showed (e.g., LoCoco et al., 2013). Unfortunately, bleeding complications are often an exclusion criterium for these studies (To this reviewer, it is not clear how many APL patients are lost on the way towards a clinical study). APL is by some still considered a medical emergency because of its bleeding problems due to a consumption coagulopathy. The authors mentioned early death in terms of prognosis but not of mechanisms. The authors defined a stemness score to distinguish high-risk from normal-risk patients. The question remains whether FLT3-ITD and high WBC would do the same job much quicker.

Response to general comments

We sincerely appreciate the reviewer's insightful comments and recognition of the scientific contributions of our study in APL using scRNA-seq. The reviewer's suggestions have been instrumental in guiding us toward clearly describing the novelty, aims, and findings of our work. We are committed to enhancing the quality and charity of our manuscript following the reviewer's valuable suggestions, as summarized below.

- **The novelty of this study:** We thank the reviewer for pointing out the need to emphasize the novel aspects of our study. In response, we have added specific sections to highlight

the novel contributions of our research, as well as the advancements we have made in our conclusions compared to previous studies in this field (**lines 580-594**).

- **Clarity of aims:** We acknowledge the need for a more explicit delineation of the aims of our study. To enhance clarity, we have revised the manuscript to ensure that the aims are presented with precision. More specifically, our primary aims are to investigate the APL cellular composition and examine the *in vivo* effects of ATRA on these compartments, rather than directly on early death. The impetus for investigating early death in APL within this study stems from our observation that the identified APL stem-like cells are associated with survival outcomes, particularly early death. In the revision, as suggested by the reviewer, we have clarified the primary aims of our current study (**lines 97-99**) and revised the manuscript to ensure that the investigation of early death is explicitly stated and understood (**lines 489-492**).
- **Early death by bleeding in APL:** We agree with the reviewer that fatal hemorrhage is a pivotal factor influencing early death in the treatment of APL. This is a significant concern that has been the subject of numerous studies, including our previous works (Dong et al., 2022, Front Med.; Lin et al., 2021, Clin Cancer Res.). While recognizing the paramount importance of this issue, we want to clarify that it falls beyond the scope of the current study. In the revised manuscript, we have referenced relevant literature to provide context and acknowledge the broader challenges associated with APL treatment and early death due to bleeding complications (**lines 635-657**).
- **The impact of ATRA on stem cell self-renewal:** We are also intrigued by the *in vivo* effect of ATRA on the self-renewal of APL stem-like cells. Indeed, our analysis has revealed the enrichment of genes associated with self-renewal capacity in APL stem-like cells (**line 320**). However, as currently pointed out, the effects of ATRA on normal HSPCs and malignant LSCs are much more complicated than thought and can be divergent, as evident in these references (Purton, 2007, PPAR Res.; Jacobsen et al., 1994, J Exp Med.; Purton et al., 2000, Blood). To address this complexity, we propose that future efforts should focus on experimentally investigating and validating the impact of ATRA on APL stem-like cells, ideally through *in vitro* colony-forming assays and *in vivo* mouse reconstitution assays. Both *in vitro* and *in vivo* approaches will provide a more comprehensive understanding of ATRA's effects on self-renewal dynamics, considering the fact that single-cell RNA-seq technologies we are using are limited by (and not specifically designed for) accurately detecting self-renewal processes. We have added the discussions on how to address this complexity, along with the need for further experimental investigation (**lines 658-660**).
- **Discussion on the clinical implications of the APL stemness score:** We appreciate the reviewer's suggestion to provide a detailed discussion on the clinical implications of the APL stemness score. First, our constructed APL stemness score was a strong predictor for the prognosis and risk of APL, particularly early death. Further analysis of the hazard ratio and the area under the survival curve (AUC) both showed that the APL stemness score exhibited superior predictive performance for early death compared to known risk factors, such as high WBC counts and FLT3-ITD status. Incorporating our stemness score with the examination of WBC levels and aggressive mutations offers a valuable addition to the current APL risk assessment. While the detection of WBC levels enables clinicians to promptly assess risk, our stemness score may serve as a reliable measure for more intensive surveillance that is both practical and of clinical relevance. We have incorporated these discussions into the revised manuscript (**lines 626-634**), particularly emphasizing its potential benefits in risk prediction when compared to traditional markers

such as WBC count and FLT3-ITD.

Major points:

Abstract.

Well written, but the study's aims have to be made clear.

Response

We have amended our abstract in the revised manuscript to provide greater clarity regarding our objectives (**lines 34-39**) as follows:

“However, the intricate cellular hierarchies within APL, including leukemic stem cells, remain poorly understood, hampering risk assessment and therapeutic targeting strategies. Here, we report a comprehensive single-cell landscape derived from 16 APL patients, uncovering the cellular composition, the *in vivo* effects of all-trans retinoic acid (ATRA) on these compartments, and their impacts on early death.”

Main/Introduction

The introduction contains some misleading or unclear statements, and not all are supported by references. This reviewer would re-write it, giving it a better order and clarifying things.

- *line 59: Did the authors mean that the effectiveness of ATO and ATRA depends on the target cell? This depends on PML::RARA, because ATO and ATRA degrade or cleave PML::RARA, respectively, independently of the cell in which it is expressed.*
- *line 66: This reviewer disagrees with the authors: Next focus for a high-risk APL patient is the survival of the bleeding and the differentiation syndrome as this determines the risk of APL. The report lacked any discussion about potential mechanisms that could be responsible for the bleeding.*
- *Independently of the origin of the APL-initiating leukemic stem cell, PML::RARA is the major determinant efficiently targeted by low-dose ATO. Therefore, if the study aims to determine the effect of ATRA on APL-initiating leukemic stem cells in an early stage to avoid high-risk patients experiencing early death, it must be stated more clearly.*
- *line 82: This reviewer disagrees with the authors. Many more APL models contributed to current knowledge of APL, and NB4 cells are one of the worst.*
- *line 100: that PML::RARA determines the APL initiating leukemic stem cell is known.*
- *line 106: that ATRA induces differentiation of primitive APL blasts has been known since the nineties of the last century. This is the basis of the use of ATRA in APL. This statement needs to be clarified.*

Response

We appreciate the reviewer providing constructive comments and helping us enhance the accuracy in describing the Main/Introduction section. In response to the feedback, we have taken the following actions: (i) we have carefully revised and restructured the sections, particularly the statements pointed out by the reviewer, to ensure that the statements are accurate and well-referenced; and (ii) we have incorporated relevant references to bolster the statements where needed, as mentioned in **lines 62, 69, 90, 93**. Below, we address each of the points with the corresponding actions we have taken in our revision.

- **line 59:** We agree with the reviewer that the effectiveness of ATRA and ATO is indeed dependent on the presence of PML/RAR α , which they target for degradation or cleavage, respectively. We have removed the misleading sentence.

- line 66: As suggested, we have included a detailed discussion on the current challenges associated with early death due to bleeding complications in APL treatment (**lines 635-657**). We have made an effort to present a balanced view of the current understanding in this area. Also, we have toned down the previous line 66 (**current lines 72-74**) as follows:

“In the context of APL, it has now become imperative to explore whether leukemic cells driven by the same **oncogenic** driver, PML/RAR α , exhibit **diverse** cellular states.”

- Clarification of study aim at ATRA: As suggested, we have stated more clearly that our study aims to determine the *in vivo* effects of ATRA on the cellular compositions and differentiation hierarchy of APL, with a particular focus on early-stage intervention to prevent early death in high-risk patients (**lines 34-39, lines 489-492**).

- line 82: We agree with the reviewer that various APL models have contributed to the current knowledge of APL. We have revised the previous line 82 (**current lines 88-90**) to present a more inclusive overview of the research models that have been instrumental in APL studies as follows:

“Furthermore, **extensive investigations into the molecular mechanisms underlying effective targeted therapies in APL have primarily relied** on *in vitro* analyses or *in vivo* murine models^{3, 7, 8}.”

- line 100: We agree with the reviewer that the initiating role of PML/RAR α in APL leukemogenesis is supported by substantial evidence from murine models. We have provided additional background information in the Main/Introduction section (**current lines 61-62**). Our finding adds to this body of knowledge by presenting single-cell level evidence of PML/RAR α and FLT3-ITD on the stemness characteristics of APL stem-like cells. We have toned down the previous line 100 (**current lines 109-111**) as follows:

“Secondly, we showed that, at the **single-cell level**, the stemness characteristics of APL stem-like cells were **primarily** determined by PML/RAR α and **further** enhanced by FLT3-ITD.”

- line 106: The previous line 106 has been revised to accurately reflect the finding of our study (**current lines 116-118**) as follows:

“Lastly, our **single-cell level investigations into** the *in vivo* effects of ATRA **confirmed** that ATRA directly **targeted** APL primitive blasts, **leading to** their differentiation and maturation.”

We believe these revisions address the reviewer's concerns and enhance the accuracy of the Introduction section.

Results

In general, many statements need references. In addition, many "likely" indicate much speculation, which is evident if data came from bioinformatics without any experimental validation.

Figure 1/Text for Figure 1

- line 127: *define what is meant by early death in APL (it is slightly different in APL than in other AMLs)*

- line 129: *define the source of the 136,497 cells. A pool of 16 patients? The sentence "...generated from the BM at disease onset..." is misleading.*

- line 133: *same as above.*

- *How is bias avoided: age, gender, etc., during the selection of patients for different analyses? For each analysis, a different number of patients was selected. Which were the criteria for the selection? It is crucial for the clinical significance of the results.*

- line 151: *Published findings need the reference.*
- line 162: *a widely used diagnostic marker for APL - reference?*
- line 175 and others: *significantly - typically, this term is used if there is a p-value. Does the software used provide it?*
- line 189-193: *Likely?? what does it mean?*

Response

We are grateful to the reviewer for insightful comments on Figure 1 and the texts. We have diligently addressed the raised concerns, incorporating the suggested changes into the manuscript. Furthermore, we have ensured that relevant references have been added to support our statements, including revisions to **previous lines 151 and 162 (current lines 171 and 184)**.

- line 127: As suggested, we have added the definition of early death to the revised manuscript (**current lines 143-144**) as follows:

“This cohort included four patients who experienced early death, which is defined as death occurring within 30 days from diagnosis.”

- lines 129, 133: Sorry for the vagueness. We have clarified this issue in the revised manuscript (**current lines 144-147, 147-150**) as follows:

“In this endeavor, we generated a total of 136,497 cells by combining 16 separate scRNA-seq datasets of APL BM samples at disease onset, forming the foundation for a comprehensive understanding of APL cellular composition.”

“In parallel, we reanalyzed 23 separate scRNA-seq data of normal BM samples from healthy individuals (Gene Expression Omnibus with accession IDs of GSE120221 and GSE130116) to construct normal hematopoietic cell populations (totaling 102,792 cells) for use as controls.”

- Regarding the concern about potential patient selection bias, we understand the importance of this aspect for the clinical significance of our results. Patients were randomly selected to ensure that the patients in each analysis were representative of the broader APL patient demographic. The selection was also contingent upon the availability of high-quality samples, which is a prerequisite for the reliable scRNA-seq analysis that our study relies on. This ensures that the data generated is of the highest fidelity and suitable for the in-depth analyses we conducted. Furthermore, to mitigate potential biases introduced by sample origin, we have performed batch effect adjustments (detailed in the **Methods** section, **lines 749-752**). This step is crucial for minimizing non-biological variations between samples. Finally, we have included UMAP plots, which provide visual evidence supporting the unbiased nature of our sample selection and cluster identification (**new Supplementary Figure 23**). Specially, **Supplementary Figure 23a** illustrates the impact of sample origin, validating the unbiased nature of our cluster identification, and **Supplementary Figure 23b,c** confirms that no single demographic group is over-represented in our clusters.

- line 151: As suggested, the relevant references have been added to the previously published findings (**current lines 171**) as follows:

“Using 23 normal BM samples, we established the baseline of cellular diversity, which revealed seven major cell populations, consistent with previously published findings^{23,24}.”

- line 162: As suggested, the relevant reference has been added to the diagnostic power of MPO for APL (**current lines 184**) as follows:

“This cluster exhibited a high expression level of the gene *MPO*, which encodes a widely used diagnostic marker for APL²⁵,”

• line 175 and others: The P values provided by FindMarkers have been added to the revised manuscript as suggested (current lines 197 and 203):

“The analysis revealed that significantly upregulated genes (adjusted *P* value < 0.05) in APL blasts were of functional relevance to several key processes,” and “In contrast, genes that were significantly downregulated (adjusted *P* value < 0.05) in APL blasts were enriched for immune response-related functions,”

• line 189-193: Sorry for the vagueness. We have revised the corresponding statements in the revised manuscript (current lines 213-217) as follows:

“Also repressed in APL blasts were the mediators of interferon (IFN) signaling (e.g., STAT1, IRF8, and IRF1). Conversely, oncogenic TFs (e.g., RB1, HIF1A, and MAX), epigenetic regulators (e.g., SMAD4, MBD1, YY1, and SP3), and cell proliferation-associated TFs (e.g., RNF6, ELF2, and TOP2B) were activated in APL blasts compared to normal GMPs (Fig. 1g).”

New Supplementary Figure 23

New Supplementary Figure 23, UMAP plots depicting APL blasts from the 16 APL patients after batch effect adjustment. Cells are colored based on their respective sample origin (a), different age ranges (b), and genders (c), respectively. These results demonstrate that our identified clusters of APL blasts were not influenced by potential confounding variables such as age and gender.

Figure 2/Text for Figure 2

Many statements need references (e.g., "...known marker genes.."). The entire interpretation of data is based on bioinformatic data without any validation.

- line 199: A stem cell is supposed to initiate a hierarchy?
- Are *PML::RARA* target genes only expressed in the APL-stem-like cell cluster? Then, the other cell clusters have to be tested for the correctness of the interpretation. If *PML::RARA* does things in a cell-type-dependent manner, it would be a breakthrough result, but it has to be proved.

Response

We extend our gratitude for the reviewer’s continued guidance on our manuscript. In response, we have made the following improvements to enhance the accuracy and clarity of Figure 2 and the associated texts:

- As suggested, we have incorporated references that support the use of known marker genes. These references have been added to the revised manuscript at appropriate locations to

provide a strong foundation for our claims (lines 226, 228, 232, 236-238, and 242).

- Regarding the validation of bioinformatic data interpretation, we have taken multiple aspects to verify our definition of cell population. This includes validating our definition of cell population by utilizing PML/RAR α -targeted scRNA-seq data, established markers of hematopoietic populations or canonical marker genes for APL, recent scRNA-seq studies, and published gene signatures (Galen et al., 2019, Cell; Naldini et al., 2023, Nat. Commun.). These validation analyses not only strengthen the robustness of our data interpretation but also bolster the credibility of our findings.
- We have addressed the reviewer's suggestion to improve clarity by revising the statement on previous line 199, with the revised statement now reading (current line 222 in the revised manuscript):

“Characterizing the intratumoral heterogeneity of APL blasts reveals a complex cell-state transition trajectory with leukemic stem-like cells sitting at the top”.

- We appreciate the reviewer for raising the issue about the regulation of PML/RAR α in various cell branches. In line with the recommendation, we have performed an in-depth analysis of the distribution of PML/RAR α targets obtained from CUT&Tag-seq across the 6 cell branches. As shown in new Figure 2i and new Supplementary Table 5, the PML/RAR α targets exhibit branch-specific expression patterns, not limited to the stem-like cells alone. Further GO analysis has unveiled that these branch-specific targets are associated with distinct roles closely related to each branch (new Supplementary Figure 7c). Moreover, the similar expression of PML/RAR α in all 6 branches (new Supplementary Figure 7a), as addressed in response to the feedback from Reviewer #2 specific comment 1 and Reviewer #4 major concern 1, further reinforces the diverse regulatory roles of PML/RAR α beyond its involvement in the stem-like cells. These findings collectively demonstrate that PML/RAR α exerts branch-specific regulation in APL blasts.

New Figure 2i

New Figure 2i, Branch-specific expression patterns for PML/RAR α targets across the APL trajectory. The left heatmap visualizes the single-cell expression of PML/RAR α -regulated branch-specific marker genes across branches, with rows representing genes and columns for cells. For the purpose of visualization, 1,000 cells were selected from each branch. The right heatmap displays the mean gene expression across branches, accompanied by the annotations of representative marker genes on the right side.

New Supplementary Figure 7c

C

New Supplementary Figure 7c, Pathways significantly enriched for PML/RAR α -regulated branch-specific marker genes in different branches.

New Supplementary Figure 7a

New Supplementary Figure 7a, Expression levels of PML/RAR α across the 6 branches of APL blasts.

Figure 3/Text for Figure 3

- Are NB4 cells the same as APL? As there were criticisms in the introduction about using NB4 as an APL model, it has to be proved.
- NB4 compared to APL-stem-like cells: Explain the apparent expression of PML::RAR α target genes in stem-like cells but not in further differentiated clusters, although NB4 cells have shown their promoter occupancy.
- line 295: are TGIF1 and HIF1A really co-factors for PML::RAR α ?

- line 302: it is known since the 90ties of the last century. This reviewer would change the text to a better conclusion.
- line 326: do you suggest by this that FLT3-ITD precedes the t(15;17)?
- line 332: how did you select the FLT3-ITD positive APL patients; how was FLT3-ITD expression? Were these experiments performed with FLT3-ITD-negative controls?
- line 334: Speculation is for the discussion.

Response

We appreciate the reviewer's continued guidance in addressing the concerns related to Figure 3 and related content. In response, we have undertaken the following actions:

- We have revised the description of NB4 in the introduction. For more details, please refer to our response to the reviewer's comment on *line 82* for Main/Introduction.
- Regarding the utilization of the PML/RAR α targets obtained from NB4 cells, we acknowledge the limitations of using NB4 as an APL model. Nevertheless, NB4 cells have been widely utilized to examine PML/RAR α targets within the field, as evidenced by several studies (Stunneberg et al., 2010, Cancer Cell; Wang et al., 2010, Cancer Cell; Tan et al., 2021, Blood; Kamashev et al., 2004, J Exp Med.). These studies provide a valuable starting point for identifying PML/RAR α targets in our current investigation. We are actively planning future studies to delve into the diverse transcriptional regulation of PML/RAR α in distinct APL clusters (**new Figure 2i**) sorted from primary patient samples, thus corroborating our initial findings.
- line 295: We have revised the statement about TGIF1 and HIF1A (**current lines 332-335**) as follows:

“By comparing with normal HSPCs, we found that **stemness-associated TFs**, such as TGIF1 and HIF1A^{45, 46} were activated, suggesting their **potential roles** in regulating the stemness of APL cells.”
- line 302: As suggested, we have revised this conclusion regarding the role of PML/RAR α (**current lines 340-342**) as follows:

“**These findings highlighted the critical roles of PML/RAR α in regulating APL stem-like cells at the single-cell level.**”
- line 326: We have revised the description of the findings on FLT3-ITD (**current lines 367-368**) as follows:

“suggesting that **the presence of FLT3-ITD** might be essential for **enhancing the** leukemic stemness.”
- line 332: Regarding the detection of FLT3-ITD, we would like to clarify that FLT3-ITD status is part of the routine diagnostic evaluation performed at the time of APL diagnosis for every patient. The relevant mutation information for the patients in this study, including FLT3-ITD, has been incorporated into the **revised Supplementary Table 1**. This addition ensures that our data is complete and transparent.
- line 334: We have removed the speculation to the discussion.

Figure 4 and 5/Text for Figure 4 and 5

It is not surprising that the tumor load (higher percentage of APL blasts) increases with the number of APL stem-like cells, which is correlated with the WBC. This bioinformatic analysis will not substitute the simple WBC and the check for FLT3-ITD.

- *line 389: BM infiltration. Do you think that the origin of APL is outside the BM? This has to be proved!*
- *line 395: Explain because only experts on PML::RAR α know what is meant.*
- *11 gene stemness score by LASSO: Is the prognostic relevance of this score better than the one in use for OS, EFS, and DFS? It is great to have a better prognosis/prediction of the risk for early death.*

Response

We appreciate the reviewer's insightful comments on Figures 4 and 5 along with their related texts. To address the concerns raised, we have made the following revisions:

- We acknowledge the reviewer's insightful observation regarding the significant correlation between the higher percentage of APL stem-like cells within APL blasts, a higher proportion of APL blasts in bone marrow cells, and high WBC counts. We have clarified that while our data demonstrated these correlations, the specific causal relationships and the underlying biological mechanisms connecting these factors were not explicitly addressed within the scope of our study. We emphasize the need for further investigations to delve deeper into understanding the intricate biological pathways and mechanisms that establish these associations between the abundance of APL stem-like cells, APL blasts in bone marrow, and the concurrent rise in WBC counts. We have incorporated these clarifications to provide a more accurate context for the observed correlations in the revised manuscript (**lines 420-430**).
- We acknowledge the critical role of traditional assessments such as WBC counts and the detection of FLT3-ITD mutations in risk evaluation for APL patients. Our bioinformatic analysis introducing the APL stemness score is intended to complement and enhance the current risk assessment strategies, rather than replace them. We have emphasized that our APL stemness score offers independent predictive value for poor outcomes and early death, surpassing the prognostic capabilities of high WBC counts and FLT3-ITD status (**Supplementary Figure 19**). We highlight the potential of integrating our stemness score alongside these conventional measures (including WBC and FLT3-ITD status) to provide a more comprehensive risk assessment system for APL. While WBC levels are fundamental for prompt risk evaluation, our stemness score offers an additional reliable measure, enabling more intensive surveillance. This combined approach, leveraging both conventional and novel assessment methodologies, aims to provide a more robust and clinically relevant risk evaluation system for APL. For a more detailed explanation, please refer to our general response to this reviewer, specifically addressed in Point 5.
- We have clarified the statement regarding the bone marrow as the primary site for APL origin to avoid any potential confusion statement (**line 389** in the previous manuscript, **lines 430-432** in the revised manuscript) as follows:

“These results indicated that APL patients with a higher percentage of APL stem-like cells in APL blasts might have an increased tendency for blasts to circulate in peripheral blood.”
- To ensure clarity for a broader audience, we have added a brief description of the three typical isoforms of PML/RAR α in the Figure 5 legend (**lines 482-484**) as follows:

“* PML/RAR α isoforms mainly encompass three typical types: long (L), short (S), and variant (V) types, respectively defined by the breakpoint of PML on intron 6, intron 3, and exon 6.”
- We have addressed the importance of comparing the prognostic relevance of our newly constructed 11-gene stemness score with common risk factors for APL patients, such as

the WBC count and FLT3-ITD status. Our study has recognized the significance of this comparison, and our findings show that the 11-gene stemness score is a robust independent risk factor for overall survival (OS) and event-free survival (EFS), although it did not display significance for disease-free survival (DFS) (Figure 5b and current Supplementary Figure 17). Furthermore, our comprehensive assessment has revealed that the 11-gene stemness score emerges as a robust independent risk factor for early death (new Supplementary Figure 19a). We emphasize that the score exhibited superior predictive capabilities for early death when compared to common risk factors and established risk scores such as the Sanz score and a recently published risk assessment tool for early death in APL (Österroos et al., 2022, Haematologica) (new Supplementary Figure 19b). For more details, please refer to Point 5 of the general response.

New Supplementary Figure 19

New Supplementary Figure 19. Comparison of the APL stemness score against known risk factors and established risk assessment tools for APL. **a**, Forest plot of multivariate Cox analysis illustrates the APL stemness score as an independent risk factor for predicting early death in APL patients. WT, wild-type FLT3; N, the number of patients. **b**, Receiver Operating Characteristic (ROC) curve and the corresponding Area Under the Curve (AUC) values for evaluating the performance of the LASSO model, along with other models, genetic events, and clinical features in predicting early death.

Figure 6/Text for Figure 6

The analysis here lacks negative controls.

- line 438-40: Be more careful with statements. Is the high risk for bleeding of a high-risk APL patient due to the progression of the disease? What is the reason for the bleeding/early death, how long does it take for ATRA to overcome it, and how do you treat it in the meantime?
- It is not explained why day two of ATRA treatment.
- The analysis was performed without considering what ATRA is supposed to do with PML::RARA - cleavage. Therefore, control of healthy BMs would be more than helpful and interesting. Does ATRA normalize the transcription, or is it completely different?

Response

We appreciate the reviewer's comments and suggestions, which have been instrumental in refining Figure 6 and the accompanying text. We have addressed each of the points as follows:

- **Negative controls in the analysis:** We understand the reviewer’s concern regarding negative controls in our analysis, particularly concerning the effects of ATRA on healthy BMs. The primary aim of the Figure 6 section was to investigate the *in vivo* effects of ATRA treatment on cellular hierarchies of APL and associated transcriptional changes, as well as to identify distinct transcriptional programs in APL patients who experience early death. The use of bone marrow samples taken before ATRA administration served as a robust control for this investigation. However, we acknowledge that investigating the actions of ATRA on normal bone marrows is beyond the scope of this work, and it may raise potential ethical and scientific concerns. For example, administering ATRA to healthy individuals may pose health risks and would require additional ethical approval. Additionally, the biological response of normal cells to ATRA is complex and sometimes contradictory, as highlighted in previous studies (Purton, 2007, PPAR Res.; Jacobsen et al., 1994, J Exp Med.; Purton et al., 2000, Blood).
- **Inappropriate statement in previous lines 438-40:** We have revised the statements in **current lines 489-492** to ensure accuracy and careful wording. We also recognize the importance of comprehending the causes and clinical approaches to managing bleeding/early death in APL patients. However, we emphasize that this topic was not the central aim of our current study. For further information, please refer to the relevant references (Kwaan et al., 2019, Semin Thromb Hemost.; Bryan C Hambley, 2021, Front Med (Lausanne).; Tallman et al., 2004, J Thromb Haemost.). The revisions are as follows:
 “Administering ATRA as early as possible **has been proven essential in reducing** the early death rate in **APL^{48, 49}**. **We then delved further into exploring** the *in vivo* **impact** of ATRA treatment on APL cellular hierarchies, **with a particular focus on** the primitive blasts, **as their** stemness might influence treatment response.”
- **Reason for choosing day two of ATRA treatment for analysis:** We have added an explanation in the revision regarding the collection of ATRA-treated patient samples on day 2 (**lines 494-495**). This decision was based on the notable increase in CD11b expression observed on day 2 (**new Supplementary Figure 20a**).

We believe that the revisions have enhanced the clarity and quality of our manuscript. We are grateful for the reviewer’s constructive comments, which have been instrumental in refining our work.

New Supplementary Figure 20a

New Supplementary Figure 20a, Comparison of CD11b expression before (Day 0, **left panel**) and after (Day 2, **right panel**) treatment with ATRA in the APL05 patient, as detected by FCM.

Discussion

The discussion can be shortened. Take out all other AMLs as they further complicate the already complicated picture. The only significant parts at the moment are

1. lines 535-543

2. lines 589-610

This reviewer would discuss the following:

- How do these studies fit with other theories of the mechanisms for the bleeding in high-risk APL. Are these theories in line with the findings that high-risk patients have a more pronounced stem-like cell portion?*
- How would an APL patient benefit from the findings, and how does it compare with what is already available and in use.*

Response

We thank the reviewer for the constructive suggestions on the **Discussion** section. We have made the following refinements to the discussion. First, we have shortened discussions to focus more directly on the key findings of our study. Second, we have provided a comprehensive overview of how our findings fit within the broader context of existing theories on early death in APL patients (**lines 635-657**). Third, we have included a section discussing the clinical implications of our APL stemness score and its comparison with currently used tools (**lines 626-634**).

Methods

Patient samples

- provide a number of the Ethic approval. Each approval is supposed to have a number needed for tracking.*
- It appears to this reviewer that the authors made much effort to make this study reproducible to others.*

Response

We appreciate the reviewer's suggestions in our **Methods** section. As suggested, we have included the ethics approval number (2021/154) in the **Methods** section (**line 683**).

We indeed made considerable efforts to make this study reproducible for others, and we are pleased that these efforts have been recognized.

Reviewer #2, expertise in leukemia stem cells (Remarks to the Author):

In this paper, the authors use single-cell RNA-seq (scRNA-seq) data from 16 APL patients to examine the cellular composition and differentiation hierarchy of APL. They identify a stem-like cell population with projected differentiation trajectories and construct an 11-gene stemness score associated with the proportion of APL stem-like cells and survival outcomes. Finally, they examine transcriptional changes before and after ATRA treatment and identify distinct transcriptional programs in APL patients with early death.

Overall, this work provides a valuable resource for studying APL and potential mechanisms of therapeutic resistance leading to early death. As APL has a very good prognosis in the majority of patients with highly effective targeting of PML-RAR α by ATRA and ATO, the biggest potential impact of this work would be in developing tools for identifying the minority of patients at high risk of induction failure and early death. While the authors do provide some data and first steps in this regard, there are some issues that if addressed would improve the overall impact of this work. In addition, some of the statements in results and discussion are not supported by the data and should be revised.

Response

We greatly appreciate the reviewer's affirmation of our work and the insightful comments for improving the quality of our work.

Specific comments

1. In general the methods are all using well-accepted tools and the authors provide their code in a well documented format. The authors should be careful not to use definitive language in describing inferred trajectories. As well, the language regarding the roles of RUNX1 etc and CEBP family members among the total genes listed in Figure 2f seems overstated without supporting functional analysis. The statement that "PML-RAR α was...widely expressed in all 6 branches of APL blasts, especially the stem-like cluster, further supporting the hypothesis that APL might initiate from these leukemic stem-like cells" is not supported by Figure 2h, where the expression level in the stem-like cluster looks very similar to those of the CMP/GMP clusters.

Response

We are grateful for the reviewer's acknowledgment of the methodological rigor and the transparency of our code documentation in this study. We also appreciate the reviewer's suggestions regarding the language used to describe the bioinformatic results. As suggested, we have toned down the statements to describe the inferred trajectories (**lines 256-260**) and the roles of genes such as RUNX1 and CEBP family members (**lines 260-265**) in our revised manuscript.

Regarding the expression of PML-RAR α , we realize that our initial wording may have been misleading. Our original intention was to convey that PML-RAR α is expressed across all six identified branches of APL blasts, including the stem-like cells, without suggesting a higher prevalence in any particular branch. To clarify this, we have included an additional figure (**new Supplementary Figure 7a**) to accompany **Figure 2h** to demonstrate the uniform expression of PML-RAR α across all branches and revised the statement accordingly. The revised text now clearly states: "PML/RAR α -targeted scRNA-seq confirmed that PML/RAR α was uniformly expressed across all six branches of APL blasts, with notable expression in the stem-like cell cluster (**Fig. 2g,h** and **Supplementary Fig. 7a**).".

Furthermore, we have expanded our analysis in response to comments from Reviewer #1 and Reviewer #4 and revealed branch-specific expression patterns for PML/RAR α targets and associated distinct biological processes across the 6 branches of APL blasts (**new Figure 2i** and **new Supplementary Figure 7c, lines 266-283**). We think that these revisions and added results have enhanced the clarity and accuracy of our findings.

New Supplementary Figure 7a

New Supplementary Figure 7a, Expression levels of PML/RAR α across the 6 branches of APL blasts.

New Figure 2i

New Figure 2i, Branch-specific expression patterns for PML/RAR α targets across the APL trajectory. The left heatmap visualizes the single-cell expression of PML/RAR α -regulated branch-specific marker genes across branches, with rows representing genes and columns for cells. For the purpose of visualization, 1,000 cells were selected from each branch. The right heatmap displays the mean gene expression across branches, accompanied by the annotations of representative marker genes on the right side.

New Supplementary Figure 7c

C
New Supplementary Figure 7c, Pathways significantly enriched for PML/RAR α -regulated branch-specific marker genes in different branches.

2. It would be good to provide in supplementary data some plots to show that their batch correction is sufficient – for example UMAP plots showing sample of origin, gender, age, and other confounding variables, or cumulative barplots/pie charts showing whether their clusters are over-represented for specific confounding variables.

Response

We agree with the reviewer regarding the importance of ensuring that our results are not influenced by variables such as sample origin, gender, and age. As suggested, we have included new supplementary data to visually demonstrate the effectiveness of our batch correction method. As illustrated in the **new Supplementary Figure 23a**, the UMAP plot confirms that our batch correction method has effectively removed the potential impact of sample origin on the cellular composition of APL blasts. Additionally, the results in the **new Supplementary Figure 23b and c** indicate that our identified clusters were not over-represented by any specific age or gender group. These additional visualizations support the sufficiency of our batch correction approach and further enhance the reliability of our results. Accordingly, a detailed description of our batch effect adjustment has been added in the **Methods** section (**lines 749-752**).

New Supplementary Figure 23

New Supplementary Figure 23, UMAP plots depicting APL blasts from the 16 APL patients after batch effect adjustment. Cells are colored based on their respective sample origin (a), different age ranges (b), and genders (c), respectively. These results demonstrate that our identified clusters of APL blasts were not influenced by potential confounding variables such as age and gender.

3. In several places the authors make statements regarding the initiating role of PML-RAR α in APL based on enrichment of stemness-related genes in the APL stem-like cell cluster (e.g. p12, “Collectively, these findings demonstrate that PML-RAR α is a critical determinant in APL initiation”). In fact, the demonstration of the stemness properties of these cells does not provide evidence that PML-RAR α is the initiating event. Their work does not address the possibility that there may be an earlier initiating event that precedes PML-RAR α . Furthermore, on p20 the authors state that their results provide insight into the origin of APL cells (i.e. cell of origin), when they have not addressed this in their experimental work. For example, can they exclude that acquisition of PML-RAR α in a more mature progenitors may confer stem-like properties to generate the stem-like cells that they observed?

Response

We agree with the reviewer’s comment that demonstrating the stemness properties of the APL stem-like cell cluster does not directly provide evidence that PML-RAR α is the initiating event in APL. The initiating role of PML/RAR α in APL leukemogenesis is, however, well-established and supported by an abundance of evidence, including evidence from murine models (Brown et al., 1997, Proc Natl Acad Sci U S A.; He et al., 1997, Proc Natl Acad Sci U S A.). Correspondingly, we have added the background information in the Main/Introduction section (lines 61-62) and toned down the descriptions in the **Results** and **Discussion** section to ensure that our statements are accurately reflective of our findings, including those on previous p12 and p20 (current p13 and p23). As part of our future research endeavors, we plan to verify the initiating role of APL stem-like cells in animal models, which we believe will provide a more comprehensive understanding of APL initiation and progression. We are grateful for the reviewer’s constructive comments and we think that the revisions made have enhanced the clarity and accuracy of our work.

4. In Figure 3, the authors carry out GSEA with a very large gene set (~7,500 genes). For GSEA a maximum gene set size cutoff of 500 genes is recommended for a dataset of 10-20,000 genes. Gene sets that are too large (or small) relative to total size can affect normalization. Perhaps they could consider randomly sampling multiple permutations of (for example) 500 genes from their gene set and plotting the distribution of resulting normalized enrichment scores, or take the ~500 genes which show the strongest enrichment of cut’n’tag signal at their promoters.

Response

We thank the reviewer for pointing out this. As suggested, we performed randomly sampling of 500 PML/RAR α targets, iterated this process for 100 times using R random seeds from 1 to 100, and analyzed the resulting distribution of NES values (**new Supplementary Figure 24**). The results revealed that the median NES from this random sampling was 1.466, which closely mirrored the NES value of 1.562 obtained in our initial GSEA analysis (**revised Figure 3a**). Again, we thank the reviewer for the expertise, which prompted us to implement a more rigorous analysis approach to substantiate our finding. Accordingly, a detailed description of the random sampling process for 500 PML/RAR α targets has been added to the **Methods** section (**lines 854-860**).

New Supplementary Figure 24

New Supplementary Figure 24, Distribution of NES calculated for PML/RAR α targets between the APL stem-like population and normal HSPCs after 100 iterations. NES, normalized enrichment score.

Revised Figure 3a

Revised Figure 3a, Gene set enrichment analysis (GSEA) enrichment plot of top 500 PML/RAR α . The gene set for GSEA analysis was defined based on the top 500 PML/RAR α targets according to adjusted *P*-values derived from CUT&Tag. Genes were ranked by the fold change between APL stem-like cells and HSPCs at the mRNA level. NES, normalized enrichment score.

Also in Figure 3a-e, the authors show high expression of APL-specific leukemia stemness genes (expressed in APL stem-like cluster and not normal HSPC cluster) that are also PML-

RARα targets suggesting that these pathways are important for stemness in this population. It would be informative to show whether these genes are also enriched in the other APL clusters (CMP-like, GMP-like, etc.) or whether this is specific to the stem-like cluster of APL.

Response

We appreciate the reviewer for raising this issue. As suggested, we have conducted GSEA analysis to examine the distribution of PML/RAR α -regulated leukemia stemness genes across the 6 identified APL cell branches. This analysis revealed that these genes were most significantly enriched in stem-like cells, with the highest NES among the 6 cell branches (**new Supplementary Figure 8**). This finding underscores the specificity of high expression levels of these genes in the APL stem-like cells.

New Supplementary Figure 8

New Supplementary Figure 8. Enrichment analysis of PML/RAR α -regulated leukemic stemness genes illustrating the highest significant enrichment in APL stem-like cells. The left dot plot shows the NES obtained by comparing cells in each branch with other APL cell branches. The right panel shows a GSEA enrichment plot of PML/RAR α -regulated leukemic stemness genes in the stem-like cells compared to the other APL cell branches. Genes were ranked based on the fold change between APL stem-like cells and other APL cell branches.

5. *Regarding the 11-gene scoring model, more detail should be provided as to the criteria used for selecting these 11 genes. Is it possible to provide a HR for the prediction of early death? It appears from Figure 5b that although a higher score was associated with a higher incidence of early death, the majority of patients with a score above their cutoff did not experience early death. As such, how do the authors envision such a tool could be used in the clinic? This should be discussed.*

Response

We appreciate the reviewer's insightful comments and valuable suggestions. To address these concerns, we have conducted additional analyses and provided clarifications to ensure a more comprehensive and clear explanation of our methodology and findings.

Regarding the selection criteria for our 11-gene scoring model, we have included a detailed explanation of the selection process. This process was based on the LASSO coefficient selection and variable screening method, with a minimum lambda value of 0.03548, resulting in the identification of 11 genes with non-zero estimated coefficients (**new Supplementary Figure 16a,b**). To assess the robustness of these 11 genes, we re-established the LASSO model using different random seeds. After 100 iterations, the occurrence ratio for all 11 genes

was consistently above 50%, indicating their robustness and importance (**new Supplementary Figure 16c**).

To calculate **the hazard ratio (HR)** of the APL stemness score **for predicting early death**, we have conducted a multivariate Cox proportional hazards regression analysis. The analysis reveals that our APL stemness score is an independent risk factor with a significant HR (HR = 5.627; 95% CI, 1.981-15.980; P = 0.001), when compared to the known risk factors, such as WBC, platelet count, and FLT3-ITD (**new Supplementary Figure 19a**). This HR has been added to the **Results** section (**lines 466-468**).

We understand the reviewer's concern about why some patients with a higher APL stemness score did not experience early death. First, a significant HR of 5.627 indicates a substantially higher risk (a 5.627 times higher likelihood) and does not imply that all patients with a score above the cutoff will inevitably experience early death. Second, early death is a complex event influenced by many factors beyond the stemness score, including the tumor microenvironment, cooperating mutations, epigenetic modifications, and various clinical variables. In response to the comment from Reviewer #4, we have performed a preliminary comparative analysis using our cohort data to explore differences between patients with higher stemness scores who did and did not experience early death. This analysis revealed that the IL8-related pathway, as well as abnormal metabolic processes, could potentially play a role in influencing patient survival (**new Supplementary Figure 21**). The corresponding discussions have been added to the revised manuscript (**lines 647-654**).

Furthermore, in response to the clinical implications of the APL stemness score, additional comparative analyses with known risk factors and scores have shown that our APL stemness score provides superior prognostic value. The AUROC for early death was highest for our stemness score (0.672) compared to other factors, including WBC levels (0.571), the Sanz score (0.569), and a recently published risk score (0.628, Österroos et al., 2022, Haematologica) (**new Supplementary Figure 19b**). Therefore, our stemness score offers a valuable addition to current risk assessment practices in APL, providing a reliable measure for more intensive surveillance and complementing the evaluation of WBC levels and the presence of aggressive mutations. While measuring WBC levels allows clinicians to promptly evaluate risk, our stemness score may serve as a reliable measure for more intensive surveillance, which is both practical and of clinical relevance. These results and implications have been added to the **Discussion** section (**lines 626-634**) in the revision.

Again, we express our gratitude to the reviewer for raising these issues. These additional analyses and clarifications significantly strengthen our manuscript and provide a clear and comprehensive understanding of the prognostic value of our APL stemness score.

New Supplementary Figure 16

New Supplementary Figure 16. The selection criteria for the APL stemness score. **a**, Cross-validation in the LASSO model to select the tuning parameter. The X-axis represents the log (λ) value, and the Y-axis for the partial likelihood deviance. The minimum mean cross-validated error of λ is selected. The lower X-axis represents the lambda value, and the upper X-axis scale for the number of genes in the LASSO model. **b**, Coefficient selection and variable screening of LASSO. **c**, Occurrence ratio of LASSO genes after repeated 100 times using different random seeds.

New Supplementary Figure 19

New Supplementary Figure 19. Comparison of the APL stemness score against known risk factors and established risk assessment tools for APL. **a**, Forest plot of multivariate Cox analysis illustrates the APL stemness score as an independent risk factor for predicting early death in APL patients. WT, wild-type FLT3; N, the number of patients. **b**, Receiver Operating Characteristic (ROC) curve and the corresponding Area Under the Curve (AUC) values for evaluating the performance of the LASSO model, along with other models, genetic events, and clinical features in predicting early death.

New Supplementary Figure 21

New Supplementary Figure 21, Differentially upregulated (**a**) and downregulated pathways (**b**) in patients who experienced early death (ED) compared to patients who achieved complete remission (CR) among patients with a higher stemness score (ED vs. CR).

6. Regarding the transcriptomic changes following ATRA treatment, the differentiation changes are not surprising and are in keeping with the success of current treatments. The

greatest potential impact of this work is in understanding the mechanisms underlying treatment resistance and early death. Figure 6i and 6j provide some evidence of distinct transcriptional changes after ATRA treatment in patients with early death, but the numbers are too small to draw any firm conclusions ($n=2$ scRNA-seq, $n=9$ bulk RNA-seq with 2 patients who experienced early death). They state in the Discussion that their results “provide new insights into the underlying mechanisms” but they have not in fact addressed this in any detail. It would be much more powerful to analyze a larger number of patients (including more with early death) pre and post treatment to elucidate differences in expression of TFs and identify potential pathways that could be targeted in patients with early death.

Response

We acknowledge the limitation due to the notable low incidence of early death in APL, which restricts the sample size for our analysis. Despite this challenge, we have made every effort to incorporate additional data as suggested. We have expanded our dataset to include pre- and post-ATRA treatment samples from an additional patient with early death. The results from this patient are consistent with our previous data from the other two patients with early death, providing further support for our findings. These updates have been incorporated into **Figure 6i,j** of the revised manuscript. Furthermore, to provide a deeper understanding of the mechanisms underlying early death, we have performed GO analysis on genes that were differentially expressed post-ATRA treatment in patients who achieved complete remission but not in patients who experienced early death. In addition to changes in the induction of differentiation, this analysis revealed additional insights, particularly the downregulation of stemness-associated pathways, such as the MAPK cascade, not observed in patients with early death after ATRA treatment (**new Supplementary Figure 20c,d**). We have incorporated these findings into our revision (**lines 543-546**). We agree that more work is needed to fully elucidate these underlying mechanisms, and we are planning such future studies.

Revised Figure 6i,j

Revised Figure 6i,j. **i**, Heatmap illustrating the log₂(fold changes) (log₂FC) of gene expression levels for stemness-associated CD markers and TFs between APL samples on Day 0 and Day 2 (Day 2 vs. Day 0) in different APL patients. **j**, Heatmap illustrating the log₂FC of gene expression levels for differentiation markers and hematopoietic TFs between APL samples on Day 0 and Day 2 (Day 2 vs. Day 0) in different APL patients. Gray represents patients alive after induction therapy, and black for patients with early death.

New Supplementary Figure 20c,d

c**Downregulated pathways after ATRA treatment in patients with CR rather than ED****d****Upregulated pathways after ATRA treatment in patients with CR rather than ED**
New Supplementary Figure 20c,d, Pathways significantly enriched for genes that were downregulated (c) or upregulated (d) post-ATRA treatment in patients who achieved CR but not in those who experienced ED.

Reviewer #3, expertise in hematopoietic stem cells, leukemia, therapy and scRNAseq (Remarks to the Author):

In this manuscript, Jin and colleagues present a comprehensive overview of the single-cell RNA sequencing landscape of acute promyelocytic leukemia (APL) patient blasts. In particular, the authors studied: BM from 16 patients at diagnosis (23 healthy donor BM from published datasets as control); 2 scTarget to determine the presence of PML-RAR α or FLT-ITD; validation of cellular composition and clinical associations by “Cybersort” on bulk RNAseq data from 323 newly diagnosed APL patients; 3+9 patients pre-/post ATRA induction therapy using scRNAseq and de-convolution analysis, respectively.

The authors define a small subset of PML-RAR α + stem-like cells, which appear at the top of the APL hierarchy (based on RNA velocity and pseudotime analysis) and are associated with higher WBC counts, lower platelet counts and the presence of FLT-ITD. An 11-gene stemness score derived by the LASSO algorithm from the bulk RNAseq cohort correlated with poorer outcome in multivariate analysis. The authors suggest that the risk of early death is linked to non-response of the stem-like compartment to ATRA, which is an intriguing hypothesis, yet not fully supported by the data.

This dataset represents an important resource for future studies, considering that many of the single cell RNA sequencing studies published so far have excluded APL cases. Hence, there is limited high resolution data available on the population hierarchies, differentiation trajectories and population dynamics during therapy, as well as the role of FLT-ITD in this specific AML entity. The study is well conducted, the scRNAseq and bioinformatic analyses appear state-of-the-art using validated, appropriate and widely used approaches in the single-cell community. The manuscript is overall well written, even though some English language revision may benefit the reader, including article and main paragraph titles where sometimes the underlying message is not easily conceived.

Limitations of the work include the descriptive nature of the data. Validation relies on correlation with external datasets and clinical outcomes, which would require future independent verification on a higher number of cases, ideally in a prospective setting. These limitations should be clearly stated in the discussion. In general, across the text, many conclusions relying on correlative data should be toned down, in the absence of data demonstrating a causal relationship.

Response

We greatly appreciate the reviewer’s affirmation of our work and constructive suggestions for improving our manuscript. As suggested, we have thoroughly revised the manuscript and toned down the statements regarding the corresponding conclusions. We believe that these revisions have enhanced the clarity and precision of our findings.

Major points:

- The authors define APL blasts in single-cell RNA sequencing based on subjectively chosen criteria described in the methods section, which could appear arbitrary. First, bias may occur if only APL blasts that cluster with each other are selected and those that cluster with other non-malignant myeloid or progenitor clusters are disregarded. Some smaller, rare subclusters of cells, such as the cells from APL patients labeled as part of cluster 7 in Fig. 1b,c, that appear to express both MPO and CD3E (or other small erythroid marker-expressing cell clusters) and do not co-cluster with other non-APL blasts, were not included. Although rare, and unlikely to affect the results presented in the manuscript, this critical

point in cell selection should be discussed in more detail. Given that the authors also performed PML/RAR α -targeted scRNA-seq, they could use these data for a limited number of patients to validate their approach in APL blast calling.

Response

We appreciate the reviewer's constructive comments regarding the definition of APL blasts in our scRNA-seq. To address the potential for bias, we have conducted a validation using PML/RAR α -targeted scRNA-seq data. By applying the K-nearest neighbors (KNN) algorithm, we projected the PML/RAR α -targeted scRNA-seq data onto cell populations annotated using the scRNA-seq data of 16 APL and 23 normal samples (**revised Figure 1b**). This approach confirmed the almost exclusive expression of PML/RAR α in our defined APL blasts (**new Supplementary Figure 2**), validating the accuracy of our APL blast identification method.

We also observed the presence of a few PML/RAR α -positive cells in B cells, T/NK cells, and erythroid cells. However, these were extremely rare (only 6, 15, and 2 cells, respectively, **new Supplementary Figure 2**), and we think that this is largely due to the inherent limitations of the scTarget technology rather than the misidentification of APL blasts. We have included these validation results in the revised **Results (lines 179-182)** and discussed the limitations of the scTarget technology in the **Methods** section (**lines 793-797**). We are grateful for the reviewer's constructive suggestion again, which has strengthened the clarity and robustness of APL blast calling.

New Supplementary Figure 2

New Supplementary Figure 2, Comparison of the numbers of PML/RAR α -positive cells detected by scTarget in APL blasts, B cells, T/NK cells, and erythroid cells from two APL patient samples. Cells detected with more than three PML/RAR α fusion reads are calculated.

Revised Figure 1b

Revised Figure 1b, UMAP plots of APL and normal BM cells, with color-coding indicating sample types (left panel), inferred cell populations (middle panel), and PML/RAR α -positive cells detected by scTarget in two APL samples (right panel). Cells detected with more than three PML/RAR α fusion reads are illustrated.

- *APL landscape annotation: Cluster 14 was called “CMP-like”. However, recent single cell studies have questioned the existence of CMPs as a specific progenitor, as they may contain a mix of progenitors already committed to erythroid and myeloid fate, respectively, rather than a common progenitor. Did the authors find erythroid/MK gene among the marker genes in Cluster 14? Based on Fig 2b, they rather seem to form a continuum with Cluster 11.*

Moreover, how do the cycling GMP-like cells relate to the branches? Is this a mono- or bidirectional, reversible state originating from the mature or primitive APL subsets? Validating the differences in cell cycle between normal GMP and APL GMPs in a limited number of patients, e.g. by flow cytometry, would strengthen the scRNAseq-based predictions.

Response

We appreciate the reviewer’s valuable comments on the APL landscape annotation.

First, we understand the recent discussions and evolving understanding regarding the existence and identity of CMPs. We concur that the designation “CMP-like” may not accurately capture the characteristics of Cluster 14. Based on the literature and the main markers of Cluster 14, we revised the annotation of Cluster 14 from “CMP-like” to “Prog-like”. This decision is supported by recent studies revealing that LSC-like cells typically transition into GMP-like blasts, often going through progenitor-like cells (Galen et al., 2019, Cell). We believe that this updated annotation provides a more accurate representation of the cellular state of Cluster 14 and aligns with the current understanding of hematopoietic differentiation and leukemia progression.

Second, to address the reviewer’s concern about the relationship of cycling GMP-like cells to other branches, we performed the RNA velocity trajectory analysis of the four main clusters related to cycling GMP-like cells, i.e., stem-like, Prog-like, cycling GMP-like 1, and GMP-like 1. As shown in the **new Supplementary Figure 6**, the cycling GMP-like cells originated from primitive APL subsets (i.e., stem-like and Prog-like) and progressed toward mature GMP-like subsets, suggesting that cell cycle entry was also a part of the continuum of cell differentiation. Furthermore, this finding was in line with a recent study on the trajectory of the cycling cluster, which reveals that the actively cycling cluster initiates from early progenitors and proceeds to the differentiated myeloid-like cells based on pseudotime analysis (Naldini et al., 2023, Nat. Commun.).

Third, we agree the importance of validating differences in cell cycle between normal GMPs and APL GMPs. It is regrettable that our ethical approval exclusively authorized us to work with samples from APL patients, precluding us from obtaining bone marrow samples from healthy individuals for direct comparison. This issue would be addressed in the future.

Once again, we appreciate the reviewer for raising these constructive comments, as these revisions make our findings more precise and clearer.

New Supplementary Figure 6

New Supplementary Figure 6, RNA velocity-based differentiation trajectory analysis of four main clusters, including Stem-like, Prog-like, Cycling GMP-like 1, and GMP-like 1. The analysis reveals that cycling GMP-like cells originate from primitive APL subsets and transition towards mature GMP-like subsets. This finding aligns with recent research, emphasizing that cell cycle entry is an integral part of the continuum of cell differentiation.

- The mechanism, by which a high APL stemness signature acts as an independent predictor of early death is not entirely clear. How much do increased WBC count, decreased platelets and presence of FLT-ITD (all associated with a higher proportion of stem-like cells) contribute to this association? Is the inferred proportion of stem-like cells in the bulk RNA seq data also predictive of early death? It would also be helpful to include a table with the frequency distribution of the clusters broken down for each single patient in the 16 patients where scRNAseq has been performed. Do the 4 early deaths in this cohort show a higher proportion of stem-like cells?

Response

We appreciate the reviewer's comment regarding the predictive mechanisms of our APL stemness score for early death.

First, we conducted a multivariate Cox regression analysis, which confirmed our APL stemness score as an independent predictor of early death (HR = 5.627; 95% CI, 1.981-15.980; P = 0.001), even after adjusting for WBC count, platelet count, and FLT3-ITD status (**new Supplementary Figure 19a**).

Second, as suggested, we investigated whether the inferred proportion of stem-like cells in the bulk RNA-seq data could predict early death. Our preliminary analysis revealed that the inferred proportions of stem-like cells were relatively higher in patients who experienced early death (median 1.05%) compared to those who achieved complete remission (median 0.43%).

Third, we have included a **new Supplementary Table 9** in the revised manuscript, which details the frequency distribution of the six identified cell branches for each of the 16 patients in our scRNA-seq dataset. Preliminary analysis also revealed a relatively higher proportion of

stem-like cells in the 4 patients who experienced early death (median 1.77%) compared to the 12 patients who achieved complete remission (median 0.45%).

These additional analyses and results have been added in the revised manuscript (**lines 462-468**), enhancing our understanding of the APL stemness score as an independent predictor of early death.

New Supplementary Figure 19a

New Supplementary Figure 19a, Forest plot of multivariate Cox analysis illustrates the APL stemness score as an independent risk factor for predicting early death in APL patients. WT, wild-type FLT3; N, the number of patients.

- The longitudinal samples pre-/post ATRA suggest that a subset of patients fail to downregulate stemness genes, in stark contrast to the majority of patients where stem-like cells are readily depleted, already after 2 days of ATRA. This observation is intriguing and may support an independent prognostic value of stem-like cells. However, additional data are required to consolidate this finding, first-of-all, longitudinal analysis of a higher number of poor responders (the claim is based on $n=2$, whereby mostly APL11 upregulates stemness CD markers). It would also be interesting to measure the content of stem-like cells at timepoints beyond day 2, and after a second therapeutic agent (e.g. ATO or Idarubicin) has been added. Is the sub-optimal response a matter of quantity (very high baseline content of stem-like cells), or quality (any transcriptional differences between stem-like cells from day2 responders vs non-responders)? Can this be predicted based on transcriptional differences in the stem-like cluster already at diagnosis?

Response

We are grateful for the reviewer's constructive suggestions and interest in our findings on the differential response to ATRA treatment in APL patients. We have made efforts to address the reviewer's points as follows.

We acknowledge the limitation due to the notable low incidence of early death in APL, which restricts the sample size for our analysis. Despite this challenge, we have made every effort to incorporate additional data as suggested. We have expanded our dataset to include pre- and post-ATRA treatment samples from an additional patient with early death. The results from this patient are consistent with our previous data from the other two patients with

early death, providing further support for our findings. These updates have been incorporated into the current **Figure 6i, j** of the revised manuscript.

Furthermore, we agree that extending the analysis to include timepoints beyond day 2 and after the introduction of other therapeutic agents would be of interest. While our current study provides initial insights, we are actively working to expand our dataset to include additional timepoints and post-treatment scenarios in future research.

Regarding the reviewer’s question regarding the quantity or quality of stem-like cells, our study has indeed indicated that both the quantity and the transcriptional characteristics of stem-like cells are critical in determining the treatment outcomes. Specifically:

- Our scRNA-seq data and bulk RNA-seq data both revealed a relatively higher baseline proportion of stem-like cells in patients who experienced early death (**Figure 5b**).
- Longitudinal data revealed that the stemness program persisted post-ATRA treatment in patients with early death (**Figure 6i**).
- Furthermore, as suggested, we used scRNA-seq data to compare transcriptional differences in the stem-like cells at diagnosis between patients with and without early death. We found that the stemness program of the stem-like cells at diagnosis was higher in patients with early death compared to those without early death. This was evidenced by the higher expression of stemness-associated genes, such as *FCGR2A*, *IL1RAP*, *HOPX*, *MAP2K1* and *KLF9* (**new Supplementary Figure 18**). Together, our findings underscore the multifaceted nature of stem-like cells in APL and their impacts on patient outcomes.

We are thankful to the reviewer for their constructive comments, which has significantly contributed to the depth of our study.

Revised Figure 6i,j

Revised Figure 6i,j. **i**, Heatmaps illustrating the log₂(fold changes) (log₂FC) of gene expression levels for stemness-associated CD markers and TFs between APL samples on Day 0 and Day 2 (Day 2 vs. Day 0) in different APL patients. **j**, Heatmap illustrating the log₂FC of gene expression levels for differentiation markers and hematopoietic TFs between APL samples on Day 0 and Day 2 (Day 2 vs. Day 0) in different APL patients. Gray represents patients alive after induction therapy, and black for patients with early death.

New Supplementary Figure 18

Differentially expressed genes in stem-like cells between the patients with and without ED

New Supplementary Figure 18, Volcano plot illustrating differentially expressed genes in stem-like cells between patients with and without ED. Red dots represent upregulated genes, and blue dots for downregulated genes.

Minor points:

- Line 176: *RUNX1, MYC, JAG1*: are these associated to myeloid differentiation? These factors rather regulate HSC self-renewal/differentiation. Similarly, *DNMT3A* prominently methylates DNA rather than regulating histone methylation.

Response

The inappropriate descriptions have been corrected in the revised manuscript (**current lines 198-200**) as follows:

“The analysis revealed that significantly upregulated genes (adjusted *P* value < 0.05) in APL blasts were of functional relevance to several key processes, including HSC self-renewal/differentiation (*RUNX1, MYC, and JAG1*), histone modification (*EP300*) and DNA methylation (*DNMT3A* and *MBD1*), cell cycle arrest and cell growth (*CDK6, CCNA1, and WTI*), as well as the response to endoplasmic reticulum stress and unfolded protein (*XBP1, ATF6, and USP14*) (**Fig. 1e** and **Supplementary Fig. 3a**).”

- Fig 5a/b: the cutoff for WBC needs to be clarified: is it $10 \times 10^9/L$?

Response

This typographical error has been corrected to $10 \times 10^9/L$.

- Line 326-327: “mutations, suggesting that *FLT3-ITD* might be more permissive of APL initiation and essential for maintaining leukemic stemness”. Increased *FLT3-ITD* may expand the stem/progenitor compartment, but not be necessary or essential for maintaining leukemic stemness, as seen in many *FLT3-ITD* negative myeloid leukemias. The claim should be toned down.

Response

We have toned down the statement (**current lines 367-368**) as follows: “suggesting that the presence of *FLT3-ITD* might be essential for enhancing the leukemic stemness”.

- Line 387: please indicate whether the correlation between the stem-like cells and the % of APL blasts relates to bone marrow or blood blast counts

Response

As suggested, our additional analysis revealed a significant association between the proportion of stem-like cells in APL blasts and the blood blast counts ($R = 0.35$; $P < 0.0001$; **new Supplementary Figure 13b**). The result has been added to the revised manuscript (**current lines 428-429**).

New Supplementary Figure 13b

b

New Supplementary Figure 13b, Correlation between the proportion of stem-like cells in APL blasts and the blood blast count.

- Line 445/6: it is not entirely clear why the cellular composition of day2 samples was “determined by deconvolution analysis of scRNAseq”. If the analysis is at single cell level, then the most straightforward way of comparing cell composition is to merge the day 2 data and diagnosis data into a single object and determine the cluster distribution for the individual samples. Could you please clarify?

Response

We appreciate the reviewer’s careful examination of our manuscript and sincerely apologize for any confusion caused by our previous description of the methodology. In our analysis, we indeed merged cells from both Day 0 and Day 2 to investigate changes in cell composition after ATRA treatment. Specifically, for the scRNA-seq data on Day 2, we employed the KNN algorithm to define cell types based on their nearest counterparts in the Day 0 dataset. This approach allowed us to accurately determine the cellular composition on Day 2 relative to the pre-defined six branches of APL blasts at diagnosis (Day 0). To ensure the clarity of our methodology, we have revised the sentence as follows (**current lines 495-498**):

“Using the pre-defined six branches of APL blasts at diagnosis served as the reference, the cell types of APL cells on Day 2 after ATRA treatment were determined by employing the KNN algorithm in a merged dataset that included cells from both Day 0 and Day 2.”

- Line 541 – 548: “APL LSCs have not been well defined, to a certain extent, mainly because of the CD34-negative feature of APL blast cells....” From these statements, it seems that APL LSCs are mainly CD34-. However, in their dataset, the stem-like cells are clearly CD34+ (CD34 is among the top markers of cluster 15). Please clarify.

Response

We sincerely thank the reviewer's careful observation and insightful comments regarding CD34 among the top markers in the APL stem-like cells. We acknowledge that our previous statement may have inadvertently implied that APL LSCs are predominantly CD34-negative, which could lead to a misunderstanding of the complexity of CD34 expression in APL. Defining APL leukemic stem cells is indeed a challenging task, partly due to the lower density of CD34 expression on APL cells when compared to other forms of AML (Paietta et al., 2004, Cytometry B Clin Cytom.). It is crucial to note, however, that CD34 expression in APL is variable and can be detected in a subset of cases. Studies have reported CD34 expression in 20 to 31% of APL patients (Foley et al., 2001, Am J Hematol.; Lee et al., 2003, Am J Hematol.; Albano et al., 2006, Haematologica.), with some studies reporting up to 43% when a lower cutoff threshold for CD34 positivity is applied (Massimo Breccia et al., 2014, Ann Hematol.).

In our study, the identification of CD34-positive cells within the APL stem-like cells predominantly originated from three patients who were CD34-positive based on their immunotyping results. This observation aligns with the documented variability of CD34 expression in APL patients. It is noteworthy that LSCs can be found in both CD34-positive and CD34-negative populations. Our single-cell analysis grouped cells with stemness characteristics into the stem-like cluster, which included CD34-positive stem-like cells and other stem-like cells with lower CD34 expression. The presence of CD34 as a marker in this cluster likely reflects its role as a stemness marker, rather than indicating uniform high expression across all cells within the cluster.

We revised our descriptions to more accurately reflect these nuances and to avoid any overstatement regarding CD34 expression in APL LSCs (**current lines 599-608**). The revised text now includes a more balanced view and acknowledges the variability of CD34 expression in APL. Once again, we extend our gratitude to the reviewer for bringing this to our attention, which has allowed us to clarify this important aspect of our study more clearly and accurately.

- Line 663 Projection of APL BM cells and characterization of malignant APL blasts: It is unclear why the authors use the "projection" term to describe the integration of the two datasets with harmony. Are the authors using a projection approach (for instance using MapQuery in Seurat) to project APL BM cells into the main data structure provided by Healthy BM cells, or are they using integration of the two datasets correcting for batches using Harmony, ending up in a final integrated dataset? Please clarify. The source code does not help to understand this point.

Response

We appreciate the reviewer's comment and understand the need for clarification on the method for characterizing malignant APL blasts. In the initial manuscript, we used the term "projection" in a general context, which may have led to some confusion. To be more precise, our approach entailed the integration of APL BM and healthy BM datasets, with the Harmony algorithm applied for the correction of batch effects. The purpose of this integration was to create a unified and harmonized dataset that combines both APL and healthy BM cells. This process involves batch correction and ensures that the two datasets can be analyzed together without the influence of technical batch effects.

We have revised the statement in the revised manuscript to explicitly mention that Harmony was used for batch effect adjustment (**current lines 724-725**):

“Characterizing malignant APL blasts through integration with healthy BM cells and batch effect correction using Harmony”

- Line 734 Deconvolution analysis: Why downsampling to 10000 cells? Can the profile of smaller populations, like stem-like cells, be more affected from this procedure?

Response

The purpose of downsampling to 10,000 cells was to find a balance between computational efficiency and preserving the diversity and representation of cell populations in the dataset. This practice is in line with common procedures in scRNA-seq analysis (Chung et al., 2022, Hepatol Commun.). In our analysis, when downsampling to 10,000 cells, we ensured that each branch of APL blasts was adequately represented, with each branch containing well above the recommended minimum of 5 cells, as per the guidelines provided by CIBERSORTx.

We acknowledge the reviewer’s concern about the potential impact of downsampling on smaller populations, such as stem-like cells. To address this, we took additional steps to ensure that each branch of APL blasts, including the stem-like cells, is well represented in the downsampled dataset. Specifically, we included two additional reference datasets: one with 100 cells per branch, and the other with 1,000 cells per branch. We applied the same deconvolution procedures to these three references and compared the deconvoluted percentages with the observed percentages from 12 APL samples that had both RNA-seq and scRNA-seq data available. Our analysis consistently showed no significant differences in the deconvoluted percentages obtained using these various reference approaches (ANOVA P-value > 0.05), affirming the reliability and accuracy of our downsampling.

Figure for Reviewer

Figure for Reviewer. Comparison of cell percentages using different deconvolution references. The red bars represent the cell percentages observed from scRNA-seq analysis. The yellow bars represent the deconvoluted percentages calculated using the default workflow of 10,000 cells from D0 through downsampling. The green and blue bars represent the references built from 100 cells per branch in D0, and 1,000 cells per branch in D0, respectively. P-values were calculated using ANOVA, and no statistically significant differences (ns) were observed.

Reviewer #4, expertise in CUT&Tag analysis (Remarks to the Author):

In the manuscript “Resolving single-cell intratumoral heterogeneity of acute promyelocytic leukemia identifies leukemia stem-like cells and their impacts on early death and all-trans retinoic acid responses in vivo,” Jin et al. apply single-cell RNA-seq and state-of-the-art computational analysis methods to characterize the cellular heterogeneity of primary Acute Promyelocytic Leukemia (APL) samples and how this relates to clinical outcomes. The authors identify a relatively rare population of APL cells that have stem-like characteristics based on the expression of a variety of stem cell associated cell surface markers. Using their single cell data, the authors go on to develop a computation approach to predict the prevalence of the stem-like cells in primary APL samples from bulk RNA-sequencing data. In addition, the authors show that treatment of APL patients with all-trans retinoic acid (ATRA) causes a rapid loss of the stem-like population and induction of differentiation programs in vivo. Furthermore, in the two patients that did not respond to treatment with ATRA, the authors show that numerous markers of stemness persisted. Overall, this study advances our understanding of the heterogeneity of APL and provides a prognostic molecular signature that is relatively straight forward to implement. In addition, this study provides a conceptual advance in how single-cell RNA-seq studies can be used to assess the efficacy of differentiation therapies in the future. However, several key points related to the regulation of the leukemia-stem-like cells and the differential responses of patients with APL to ATRA could be expounded on with additional analysis to prepare this manuscript for publication.

Response

We greatly appreciate the reviewer’s thoughtful comments and recognition of the significance of our study in unraveling the cellular heterogeneity within APL and its clinical implications. We have taken these suggestions into account and made substantial revisions to our manuscript to address key points raised by the reviewer, particularly regarding the contribution of PML/RAR α to APL intratumoral heterogeneity, with a focus on its critical roles in stem-like cells (see **our response to major concern 1**) and the differential response of ATRA in patients with a higher stemness score (see **our response to major concern 2**). Additionally, we have addressed the other valuable comments related to batch correction, intertumoral heterogeneity, and provided further clarifications in the manuscript. These improvements enhance the depth, clarity, and transparency of our research.

Major Concerns:

(1) The authors state that “With regard to APL, the next focus is to address whether leukemic cells driven by the same driver, PML/RAR α , can exhibit different cellular states; if so, to what extent the cellular composition and transcriptional heterogeneity could impact the targeted therapy in APL.” And say that “the stemness characteristics of APL stem-like cells were determined by PML/RAR α .” With regards to this point, the authors compare the list of 7309 PML/RAR α target genes they previously identified by CUT&Tag in NB4 cells between the APL stem-like cells and normal HSPCs. As presented the analysis is important, but does not address whether the PML/RAR α oncoprotein also contributes to intratumorally heterogeneity and is perhaps driving the stem-like state specifically.

-Related to this point: In figure 3c the authors compare the expression of several PML/RAR α target genes between APL stem-like cells, HSPCs and GMPs. However, a proper description of the GMP population that is analyzed here is not included in the text or the figure legend. Are these normal GMPs or GMP-like cells?

-When comparing other APL cell types and normal cell types do the authors see a consistent increase in the PML/RAR α target genes in APL cells across all cell types or is the upregulation of these genes specific to the stem-like cells?

-The authors need to provide a comparative analysis of the activity of PML/RAR α target genes across the 18 clusters of APL cells identified in figure 2, or at least compare the activity between the 6 major APL cell types they identify (stem-like, CMP-like, S100hiGMP-like etc.).

Response

We appreciate the reviewer for raising this important issue about the role of PML/RAR α in the intratumoral heterogeneity of APL. In response to the reviewer's valuable suggestions, we have undertaken a more comprehensive analysis and have made the following revisions to our manuscript:

- **Contribution of PML/RAR α to APL intratumoral heterogeneity:** We acknowledge that our initial analysis did not fully address the role of PML/RAR α in contributing to intratumoral heterogeneity. As suggested by the reviewer in the last subpoint of this point, we have extended our analysis to include a comparison of PML/RAR α targets across the six identified cell branches. This revealed that each branch possessed a considerable number of distinct PML/RAR α targets (**new Figure 2i**), suggesting the presence of branch-specific expression patterns for PML/RAR α targets across the APL trajectory. Additionally, GO analysis illustrated that PML/RAR α targets were involved in distinct functional pathways closely related to each branch's characteristic, such as the pathways involved in stem cell maintenance enriched in the APL stem-like branch and pathways linked to ribosomal functions enriched in the GMP-like branch (**new Supplementary Figure 7c**). These findings emphasize the critical involvement of the PML/RAR α oncoprotein in the intratumoral heterogeneity of APL, including in regulating the stem-like cells. Together with our responses to Reviewer #1's comment on Figure 2 and Reviewer #2's specific comment 1, we have incorporated this information into our manuscript (**lines 266-283**).
- **Proper description of GMPs on Figure 3c:** We have updated the labeling in the previous Figure 3c (**now Figure 3b**) to clearly indicate that the GMPs included in our comparison are normal GMPs.
- **Comparison of other APL cell types with normal cell types:** It is challenging to compare other APL cell types with their normal counterparts due to the absence of matched normal cell types in our dataset. Instead, we have conducted a direct comparison of PML/RAR α target genes across the identified 6 major APL cell branches. This analysis has revealed that distinct PML/RAR α targets are expressed in each branch, suggesting branch-specific expression patterns for PML/RAR α targets across the APL trajectory (detailed in the response to the first subpoint above).

We greatly appreciate the reviewer's guidance, which is instrumental in enhancing the depth and clarity of our study.

New Figure 2i

New Figure 2i, Branch-specific expression patterns for PML/RAR α targets across the APL trajectory. The left heatmap visualizes the single-cell expression of PML/RAR α -regulated branch-specific marker genes across branches, with rows representing genes and columns for cells. For the purpose of visualization, 1,000 cells were selected from each branch. The right heatmap displays the mean gene expression across branches, accompanied by the annotations of representative marker genes on the right side.

New Supplementary Figure 7c

New Supplementary Figure 7c, Pathways significantly enriched for PML/RAR α -regulated branch-specific marker genes in different branches.

Revised Figure 3b

Revised Figure 3b, Violin plots illustrating representative genes highly expressed in APL stem-like cells compared with HSPCs.

(2) In figure 5 the authors provide a compelling case that the APL stemness score they develop predicts the abundance of stem-like cells in primary APL samples from bulk-RNA-seq data and that an increased APL stemness score is indicative of an elevated risk of “early death.” However, in figure 6, presumably all 9 of the bulk RNA-seq samples analyzed before and after ATRA had some proportion of APL stem-like cells, and yet some patients had early death while others did not.

-Did these bulk samples have differences in their APL stemness score?

-If not do the authors see any other differences in the samples at diagnosis that went on to undergo early death versus those that did not?

-A critical question that is left unanswered is why some tumors with high stemness scores respond to ATRA while others do not. Given the wealth of data collected in this study, can the authors provide further insight on this point.

Response

We appreciate the reviewer’s constructive suggestions regarding our findings in Figures 5 and 6. We have conducted additional analyses and discussions to strengthen our manuscript.

- **Differences in the APL stemness score in bulk RNA-seq samples:** As suggested, we have analyzed the bulk RNA-seq data to examine any differences in the APL stemness score, including the difference among samples at diagnosis and the difference between samples before and after ATRA treatment. First, APL patients who experienced early death generally exhibit higher APL stemness scores at diagnosis compared to those who achieved complete remission (**new Supplementary Figure 20b**). This observation is consistent with our findings from a large cohort of APL patients (**Figure 5b**), reinforcing the prognostic value of the APL stem score. Second, we have observed a significant decrease in the APL stemness scores after ATRA treatment in patients who achieved complete remission, but this decrease is not observed in patients who experienced early death (**new Supplementary Figure 20b**). These results are in line with the transcriptional changes we observed in the stemness program (**Figure 6i,j**). These data have been added to the revised manuscript (**lines 504-507, 546-549**).

- **Why some APL patients with high stemness scores respond to ATRA while others do not:** This is indeed a critical question. As suggested, we have conducted further analyses on samples with high APL stemness scores in our previous large cohort data. These analyses reveal the IL8-related pathway as well as abnormal metabolic processes that could potentially influence the treatment outcome (**new Supplementary Figure 21**). We acknowledge that the treatment outcome might be influenced by a complex interplay of multiple factors, including the tumor microenvironment, cooperating mutations, epigenetic modifications, and various clinical variables. We plan to delve deeper into these aspects to provide a more comprehensive understanding in the future. Accordingly, we have discussed these aspects in the revised manuscript (**lines 647-654**).

New Supplementary Figure 20b

New Supplementary Figure 20b, Comparison of the APL stemness score between patients who achieved complete remission (CR) and those who experienced early death (ED) on Day 0 (D0) and Day 2 (D2) after ATRA treatment.

New Supplementary Figure 21

New Supplementary Figure 21, Differentially upregulated (**a**) and downregulated pathways (**b**) in patients who experienced early death (ED) compared to patients who achieved complete remission (CR) among patients with a higher stemness score (ED vs. CR).

(3) How the samples were profiled is not clear from this manuscript. Were the APL samples all mixed and run together on a single cell RNA-seq platform? Or were they run in parallel reaction to ensure the donor of origin for each single cell profile is known?

-Related to the last point: The authors should provide a supplementary figure showing the APL and healthy samples before batch correction.

-Related to this point: If the donor of origin is known for each single cell profile, the authors should use this information to get an idea of the intertumoral heterogeneity in the cell states of individual APL samples. Can the authors quantify the fraction of cells that fall into the 6 cell types (stem-like, CMP-like, S100hiGMP-like etc.) from each donor?

-Related to this point: Figure 3f is critical to making the point that FLT3-ITD promotes a stem-like cell state. However, how the authors were able to perform this analysis is not intuitive from the manuscript. Do the authors have separate genotyping data for each sample? If so, this needs to be clearly stated.

Response

We understand the importance of clarity in describing the experimental procedures and data analysis. We address each of the points in detail.

- **Single-cell profiling of APL samples:** We have clarified this issue in the **Methods** section of the revised manuscript (**lines 689-691**). We have made it explicit that each APL sample was individually subjected to a separate scRNA-seq experiment, preserving the donor of origin for each patient throughout the analysis. These independently generated datasets were merged to investigate the cellular heterogeneity of APL, while also allowing for accurate attribution of cellular states to individual patients.
- **Additional figure for samples before batch correction:** As suggested, we have included a **new Supplementary Figure 22** that displays the APL and healthy normal samples before batch correction. This addition enhances the transparency of our data processing and analysis methods. Accordingly, a detailed description has been added to the **Methods** section (**lines 729-730**).
- **Intertumoral heterogeneity in cell states:** As suggested, we have quantified the fraction of cells from each patient that fall into the six identified cell branches and presented this information in a **new Supplementary Table 9** and **Supplementary Figure 11 (line 425)**. This analysis provides insights into the intertumoral heterogeneity in cell states among individual APL samples.
- **Analysis of FLT3-ITD:** FLT3-ITD is routinely detected at the diagnosis of APL in the clinic. We have added this genotyping information in the **revised Supplementary Table 1** to provide a comprehensive description of the methods and data used in our study.

These updates and clarifications significantly improve the clarity and transparency of our manuscript, and we appreciate the reviewer's guidance in this regard.

New Supplementary Figure 22

New Supplementary Figure 22, UMAP plots of APL and normal BM cells, colored by sample types (**a**) and inferred cell populations (**b**), respectively. No batch adjustment was performed.

New Supplementary Figure 11

New Supplementary Figure 11, Proportions of the 6 cell branches for each APL patient.

Minor Concerns:

-Line 31: “affect it’s effectiveness”. Please consider rewording for clarity.

Response

As suggested, we have reworded this statement (**lines 34-36**) as follows:

“However, the intricate cellular hierarchies within APL, including leukemic stem cells, remain poorly understood, hampering risk assessment and therapeutic targeting strategies.”

-Line 94: “Our resource and findings are significant with fourfold.” Please Reword.

Response

Thanks for the suggestion. We have rephrased the sentence (**lines 104-105**) as follows:

“The resource and findings presented in this study hold significant implications in four aspects.”

-Figure 1g: This plot is not clear. I understand VIPER was used to generate this plot, but simply calling it a VIPER plot is not an acceptable description in the legend. Why are there two rows for each gene?

Response

We appreciate the reviewer’s comment regarding Figure 1g and understand the need for a clearer explanation of the data generated by VIPER. We have revised the figure legend for clarity. The two rows for each gene represent the distribution of activated (depicted in red) and repressed (depicted in blue) targets of a TF, with positions ranked according to the differential expression between APL blasts and normal GMPs (leftmost: the most downregulated in APL blasts, rightmost: the most upregulated in APL blasts). Accordingly, we have revised the figure legend to enhance clarity in the revised manuscript (**lines 130-135**).

-Line 179: “In contrast, immune response-related functions were enriched for genes that were significantly downregulated in APL blasts.” Suggested correction: “In contrast, genes that were significantly downregulated in APL blasts were enriched for immune response-related functions.”

Response

Thanks for the suggestion. We have corrected the sentence (**lines 202-204**) as follows: "In contrast, genes that were significantly downregulated (adjusted P value < 0.05) in APL blasts were enriched for immune response-related functions".

-Line 186: “...found that the regulatory activity of hematopoietic transcription factors (TFs) and cofactors...” The authors performed RNA-seq and it is not possible to infer the regulatory activity of these factors from this type of data alone. Please Reword.

Response

Thanks for the suggestion. We have revised the statement (**lines 210-212**) as follows:

“Our analysis suggested that the hematopoietic TFs and cofactors (such as SPI1, ERG, FOS, and RXRA) were repressed in APL blasts (**Fig. 1g**)”

- The marker gene set the authors include for HSPCs and/or LSCs are mostly cell surface markers including CD200, CD44, CD99, CD2 and FAM30A. The authors should extend this list to include other types of genes that are known to be functionally required for HSPC or LSCs, e.g. TFs.

Response

Thanks for the suggestion. We have incorporated master transcription factor genes for HSPC or LSC functions, such as *SOX4* and *MYC*, to the revised main text (**lines 237-238**) as follows:

“Of striking interest, clusters C14-C16 exhibited stemness-like characteristics with high expression of marker genes specific to early HSPCs and/or LSCs (such as *CD200*³⁰, *CD44*³¹, *CD99*³², *CD233*, and *FAM30A*³⁴) (**Fig. 2b**), master stemness-related TF genes (such as *SOX4*³⁵ and *MYC*³⁶), as well as APL characteristic genes (such as *MPO*) (**Supplementary Fig. 5**).”

-Figure 2f: In this heatmap, genes that show the highest signal in the center are presumably enriched in the stem-like cells, but this is not clear from the labeling.

Response

As suggested, we have clarified labeling of stem-like cells in the revised **Figure 2f** as follows:

Revised Figure 2f, Heatmap showing the dynamic changes in gene expression along the pseudotime. Cell branches are labeled by colors (upper panel), including stem-like cells (center), S100^{hi}GMP-like cells (left), and GMP-like (right). Characteristic TFs are listed on the right.

-Figure 3a: Presumably the left heatmap is the PML/RAR α CUT&Tag profile and the right heatmap is the IgG negative control but this is not clear from the labeling. Also, the heatmap says it is centered on the TSS, but the legend says it is centered on the summit. Which is true?

Response

We appreciate the reviewer's careful observations regarding the previous Figure 3a. As suggested, we have added the missing labels of the PML/RAR α CUT&Tag profile (left heatmap) and the IgG negative control (right heatmap) (**current Supplementary Figure 7b**) and removed the incorrect label of "TSS" in the revised figure as follows:

Revised Supplementary Figure 7b, Density plot showing the enrichment of endogenous PML/RAR α surrounding the summit of 7,309 PML/RAR α peaks in NB4 cells, ranked by PML/RAR α peak intensity.

-Adding a FLT3-ITD-targeted scRNA-seq projection onto the APL UMAP would be helpful to show the cells are also enriched throughout.

Response

As suggested, we have added the UMAP plot of the FLT3-ITD-targeted scRNA-seq data projection onto the APL UMAP in the **new Figure 3g**.

New Figure 3g, Visualization of FLT3 expression through projection onto the UMAP of APL blasts using the scTarget data from two patients. Cells detected with more than three FLT3-ITD mutated reads were color-coded according to the different branches.

-Figure 4a: this figure shows that you have scRNA-seq samples matched with bulk RNA-seq data, but this point should be clarified in the text. This would clarify that matched samples were used as a “ground truth” to test the model.

Response

As suggested, we have now included a detailed description of how we constructed the deconvolution model in the **Methods** section of our revised manuscript (**lines 824-826**). Our model was indeed developed using matched scRNA-seq and bulk RNA-seq data, which served as a “ground truth” to validate the accuracy and reliability of our model.

-Line 411: “Remarkably, we found that a high APL stemness score was significantly associated with OS and EFS.” This wording is confusing and suggests that a high APL score predicts patients will likely survive, when this is the opposite of what I infer from reading the rest of the text.

Response

We apology for any confusion. We have reworded this sentence (**line 454**) as follows:

“Remarkably, a high APL stemness score was significantly associated with a poorer OS ($P = 5.7e-3$) and EFS ($P = 0.0342$),”

-Related to the last comment: In Figure 5b many of the samples that have a DFS/EFS/OS status of “Yes”, also appear to have and Early death status of “Yes”. Shouldn’t these two categories be mutually exclusive?

Response

We have corrected the labeling mistakes on the DFS/EFS/OS status (details in the **revised Figure 5b**). We apologize for the confusion caused by such mistakes and appreciate the reviewer’s attention to detail which has helped us improve the accuracy of the data presentation.

Revised Figure 5b

-Line 456: “almost invisible” this is colloquial, please reword.

Response

Thanks for the suggestion., we have reworded it (**line 511**) as follows: “almost undetectable”.

-Line 478: “MAFB mainly upregulated...” Suggested revision: “MAFB were upregulated...”

Response

Thanks for the suggestion. We have revised it (**line 534**) as follows: “MAFB were upregulated” .

-Line 542: “mainly because of the CD34-negative feature of the APL blast cells.” Figure 2b shows that CD34 is enriched in Clusters 15 and 16, please explain.

Response

We thank the reviewer for the careful observation and insightful comment regarding CD34 expression in Clusters 15 and 16 identified in our study. APL leukemic stem cells have been challenging to define, particularly due to a lower density of CD34 expression on APL cells in comparison to other forms of AML. However, it is crucial to note that CD34 expression in APL is variable and can be detected in a subset of cases.

In our study, scRNA-seq analysis did reveal a subset of cells within the APL stem-like cluster that expressed CD34 (Figure 2b). This observation is consistent with previous literature reports, which have documented CD34 expression in a proportion of APL cases. The frequency of CD34 expression in APL has been reported to range from 20 to 31% (Foley et al., 2001, Am J Hematol.; Lee et al., 2003, Am J Hematol.; Albano et al., 2006, Haematologica.), and in some studies, it can be as high as 43% when considering a low cutoff level (Massimo Breccia et al., 2014, Ann Hematol.).

In our dataset, the presence of CD34-positive cells within Clusters 15 and 16 could be attributed to several factors. Specifically, 3 of the 16 patients included in our scRNA-seq

analysis exhibited CD34-positive characteristics based on their immunotyping results. The CD34-positive cells within these clusters were predominantly derived from these CD34-positive patients.

We acknowledge that the presence of CD34-positive cells in APL is a complex and nuanced topic. In the revision, we have included additional clarifications to ensure that our findings are presented accurately and to avoid any overstatement (**lines 599-608**).

Once again, we thank the reviewer for bringing this to our attention and providing us with the opportunity to clarify this important aspect of our study.

Reviewers' Comments:

Reviewer #1:
Remarks to the Author:

.

Reviewer #2:
Remarks to the Author:

Jin et al Nature Comm 2023 revised manuscript

The authors have addressed all my concerns and the revised manuscript is much improved and acceptable for publication.

Some minor additional comments:

In the New Figure 2i, 1,000 cells were selected from each branch. It would be good to provide a statement explaining how these 1,000 cells were picked.

In the New Supp Fig 20 c,d and New Supp Fig 21, are the up- and down-regulated pathways shown selected from the total? Presumably these were not the only pathways that were differentially regulated – if that is the case, it would be good to acknowledge this in the text somewhere and/or provide a full supplementary table of all differentially regulated pathways.

Reviewer #3:
Remarks to the Author:

The authors have done in in-depth revision of their work, which has resulted in a strengthened and clearly-articulated manuscript. All my comments have been sufficiently addressed, I fully support publication.

Reviewer #4:
Remarks to the Author:

I appreciate the authors thorough response to my comments as well as the other reviewers and I think the revised manuscript has benefitted tremendously. All my major concerns have been addressed, and I only have a few remaining cosmetic concerns that should be addressed to prepare the manuscript for publication.

(1) On line 110 the authors state "APL stem-like cells were primarily determined by the PML/RARa and further enhanced by FLT3-ITD." This is unclear, and not entirely consistent with their results. The revised Figure 3a shows the genes that are most different between the APL stem-like cells and HSPCs are enriched in the direct targets of the PML/RARa oncoprotein. Can the authors revise this statement to reflect their results more accurately?

(2) The authors have tried to improve the description of the VIPER plot in Figure 1g legend, but unfortunately, I am still missing something. They state, "the central two-row graph illustrates the distribution of the activated targets (depicted in red) and the repressed targets (depicted in blue) of a TF, with positions ranked according to the differential expression between APL blasts and normal GMPs (leftmost: most downregulated in APL blasts, rightmost: most upregulated in APL blasts)." So, in this case each column in the Viper plot is a target gene of the TF, listed to the right of the paired red a blue

row, and these targets were inferred from a previous study or collection of studies? Why does each TF have the same number of target genes? or how do you decide on the group of target genes that are shown? Whatever the case, this information needs to be succinctly summarized in the main text or the legend to make this plot interpretable.

(3) The addition of Figure 2i convincingly shows that the PML/RXRa target genes show differential expression across the six clusters. But the labels on the plot could be revised for clarity. Specifically, "Marker genes (PR targets)" everywhere else this is "PML/RARa targets" why here is it PR targets?

(4) On line 367 the authors state that "FLT3-ITD might be essential for enhancing the leukemic stemness." Is the word "essential" overstated? Aren't there many instances of APL leukemia, including those profiled in this study, with a high stemness score that lack FLT3-ITD mutations?

(5) Lines 487-488: The title of this section "The ability of ATRA in inducing differentiation of APL primitive blasts, including the stem-like cells, towards more mature cells" should be revised for clarity. What is the point of the section? What are the findings?

(6) Line 540: in all other cases the two example genes are the ones that show the pattern most convincingly, however in the pair "(HHEX and KLF9)" NFATC2 is much more convincing than KLF9 as an example of a TF that is not downregulated in APL patients with early death. Perhaps replace KLF9 with NFATC2 here.

I hope you find these comments and concerns constructive and useful in revising your manuscript.

REVIEWERS' COMMENTS

Reviewer #2 (Remarks to the Author):

Jin et al Nature Comm 2023 revised manuscript

The authors have addressed all my concerns and the revised manuscript is much improved and acceptable for publication.

Response

We are grateful for the reviewer's recommendation for publication.

Some minor additional comments:

In the New Figure 2i, 1,000 cells were selected from each branch. It would be good to provide a statement explaining how these 1,000 cells were picked.

Response

The decision to select 1,000 cells from each branch was made, with the intention of providing a clear and representative visualization of distinct branch-specific expression patterns for PML/RAR α targets. This choice takes into consideration the varying cell counts across branches, particularly the limited number of stem-like cells (2,344 cells). By opting for this specific number, our aim was to strike a balance between robust sample representation within each branch, achieving visual clarity. In revision, we have added a more detailed explanation (**lines 1020-1022**) as follows: "To offer a clear and representative depiction of the branch-specific expression patterns for PML/RAR α targets, we selected 1,000 cells from each branch for interpretation."

In the New Supp Fig 20 c,d and New Supp Fig 21, are the up- and down-regulated pathways shown selected from the total? Presumably these were not the only pathways that were differentially regulated – if that is the case, it would be good to acknowledge this in the text somewhere and/or provide a full supplementary table of all differentially regulated pathways.

Response

We value the reviewer's input regarding the presentation of up- and down-regulated pathways in **New Supp Fig. 20 c,d** and **New Supp Fig. 21**. Our study focused on highlighting the most pertinent differentially expressed pathways related to hematological malignancies or stemness, aligning with the objectives of our study. The selection criterion was based on their direct relevance to the research questions we aimed to address. We acknowledge that there are additional pathways differentially regulated in our analysis that were not included in these figures. To ensure clarity, we have included a detailed explanation in the **Methods** section (**lines 730-731**), stating: "The differentially expressed pathways depicted were the most relevant to hematological malignancies or stemness."

Reviewer #3 (Remarks to the Author):

The authors have done in in-depth revision of their work, which has resulted in a strengthened and clearly-articulated manuscript. All my comments have been sufficiently addressed, I fully support publication.

Response

We greatly appreciate the reviewer's encouraging remarks regarding the revisions made to our manuscript. We are grateful for the support and endorsement for its publication.

Reviewer #4 (Remarks to the Author):

I appreciate the authors thorough response to my comments as well as the other reviewers and I think the revised manuscript has benefitted tremendously. All my major concerns have been addressed, and I only have a few remaining cosmetic concerns that should be addressed to prepare the manuscript for publication.

Response

We thank the reviewer's encouraging comments regarding our revised manuscript. We are committed to ensuring the highest quality for our manuscript, and we take your remaining cosmetic concerns seriously, with all addressed as described in our point-by-point response below.

(1) On line 110 the authors state "APL stem-like cells were primarily determined by the PML/RAR α and further enhanced by FLT3-ITD." This is unclear, and not entirely consistent with their results. The revised Figure 3a shows the genes that are most different between the APL stem-like cells and HSPCs are enriched in the direct targets of the PML/RAR α oncoprotein. Can the authors revise this statement to reflect their results more accurately?

Response

The corresponding sentence has been revised as follows (**lines 97-100**):

"Secondly, at the single-cell level, we showed that the stemness characteristics of APL stem-like cells were determined by PML/RAR α target genes and could be further enhanced in the presence of FLT3-ITD."

(2) The authors have tried to improve the description of the VIPER plot in Figure 1g legend, but unfortunately, I am still missing something. They state, "the central two-row graph illustrates the distribution of the activated targets (depicted in red) and the repressed targets (depicted in blue) of a TF, with positions ranked according to the differential expression between APL blasts and normal GMPs (leftmost: most downregulated in APL blasts, rightmost: most upregulated in APL blasts)." So, in this case each column in the Viper plot is a target gene of the TF, listed to the right of the paired red a blue row, and these targets were inferred from a previous study or collection of studies? Why does each TF have the same number of target genes? or how do you decide on the group of target genes that are shown? Whatever the case, this information needs to be succinctly summarized in the main text or the legend to make this plot interpretable.

Response

We thank the reviewer for bringing this to our attention. For the VIPER analysis, we followed the recommended workflow using default parameters, as outlined in the tutorial available at <https://bioconductor.org/packages/viper/>. In our analysis, the regulatory model was sourced from the built-in database of the VIPER package. This model was derived from an ARACNe-inferred interactome (Alvarez et al., 2016, Nat Genet.; Basso et al., 2005, Nat Genet.), specifically designed to depict the relationships between TFs and their corresponding target genes. The regulatory model comprises 621 TFs, 6,249 targets, and a network of 172,240 interactions, collated, expanded, and evaluated through a series of studies (Guan et al., 2022, Nature; Wu et al., 2020, J Hematol Oncol.; Lei et al., 2023, Acta Pharmacol Sin.). Moreover, the target genes regulated by different TFs exhibited variability and were visualized using the default functionality provided within the VIPER tool. TFs inferred as differentially active (P value < 0.05) and previously reported to be associated with APL were selected for visualization. The corresponding legend has been revised as follows:

"**g**, Inferred activated (red) and repressed (blue) TFs in APL blasts compared to normal GMPs. The central two-row graph illustrates the distribution of activated targets (depicted in red) and

repressed targets (depicted in blue) of different TFs, with positions ranked according to the differential expression between APL blasts and normal GMPs (leftmost: most downregulated in APL blasts, rightmost: most upregulated in APL blasts). The regulatory model was based on the ARACNe-inferred interactome, provided in the build-in function of the VIPER R package. The *P*-value is shown on the left of the column, and the inferred differential activity level is shown on the right. The *P*-values were calculated using the `msvip` function in the VIPER R package. Two-sided *P*-values were calculated.”

(3) *The addition of Figure 2i convincingly shows that the PML/RXR α target genes show differential expression across the six clusters. But the labels on the plot could be revised for clarity. Specifically, “Marker genes (PR targets)” everywhere else this is “PML/RAR α targets” why here is it PR targets?*

Response

As suggested, we have revised the label from “PR targets” to “PML/RAR α targets” in the **revised Figure 2i**.

(4) *On line 367 the authors state that “FLT3-ITD might be essential for enhancing the leukemic stemness.” Is the word “essential” overstated? Aren’t there many instances of APL leukemia, including those profiled in this study, with a high stemness score that lack FLT3-ITD mutations?*

Response

As suggested, we have revised the statement (**current line 292**) as follows: “FLT3-ITD might play a significant role in enhancing the leukemic stemness.”

(5) *Lines 487-488: The title of this section “The ability of ATRA in inducing differentiation of APL primitive blasts, including the stem-like cells, towards more mature cells” should be revised for clarity. What is the point of the section? What are the findings?*

Response

As suggested, we have revised the title as follows: “*In vivo* effect of ATRA on differentiation of primitive APL blasts and its influence on early death risk”. This revised title captures the key findings of this section: the *in vivo* effects of ATRA treatment, particularly its role in the differentiation of primitive APL blasts and the stemness program, along with the implications of these factors to the risk of early death in APL patients. We appreciate the reviewer’s guidance in improving the clarity of our manuscript.

(6) *Line 540: in all other cases the two example genes are the ones that show the pattern most convincingly, however in the pair “(HHEX and KLF9)” NFATC2 is much more convincing than KLF9 as an example of a TF that is not downregulated in APL patients with early death. Perhaps replace KLF9 with NFATC2 here.*

Response

We appreciate the reviewer’s detailed observation and suggestion. As suggested, we have replaced KLF9 with NFATC2 (**current line 430**).